# Explain-then-Translate: An Analysis on Improving Program Translation with Self-generated Explanations

**Zilu Tang**[1], **Mayank Agarwal**[2], **Alex Shypula**[3], **Bailin Wang**[4],
**Derry Wijaya**[1,5], **Jie Chen**[2], **Yoon Kim**[4]
[1]Boston University   [2]MIT-IBM Watson AI Lab, IBM Research
[3]UPenn   [4]MIT   [5]Monash University Indonesia
zilutang@bu.edu

## Abstract

This work explores the use of self-generated natural language explanations as an intermediate step for code-to-code translation with language models. Across three types of explanations and 19 programming languages constructed from the MultiPL-E dataset (Cassano et al., 2022), we find the explanations to be particularly effective in the zero-shot case, improving performance by 12% on average. Improvements with natural language explanations are particularly pronounced on difficult programs. We release our dataset, code, and canonical solutions in all 19 languages.[1]

## 1 Introduction

Program translation (i.e., translating code from one language to another) has significant value in real-life applications, including in legacy software modernization and enabling programmers to quickly adapt to new languages. Within prompt-based approaches to code translation, Chen et al. (2023b) recently found that simply prompting an LLM to generate explanations of the source program before generating the target program can improve performance. However, this conclusion is drawn on a single translation direction from C++ to Python (Lachaux et al., 2020), and lacks evaluation on a broader set of programming languages including low-resource languages—a key component of code-to-code translation tasks in a software modernization setting.

This paper systemtically evaluates this "Explain-then-Translate" approach to code translation through the MultiPL-E dataset (Cassano et al., 2022). As the original dataset was constructed for the NL-to-code setting, we repurpose this dataset into a code-to-code, "MultiPL-C2C" dataset. We analyze our results in 36 different translation directions over different types of explanations. We find

that Explain-then-Translate improves zero-shot performance consistently in 18 Python-to-X translation directions, but much less so in the few-shot setting. We observe detailed explanations to be more useful when translating into high-resource PLs and from low-resource into other low-resource PLs. In contrast, translating from high- to low-resource PL's benefits from more abstract explanations. To aid future research in code-to-code translation across diverse language, we release our evaluation system, as well as canonical solutions in all languages, providing a 19-way parallel program translation evaluation set.

## 2 Explain-then-Translate for Code Translation

In code translation, we are given code $x$ in a source language and must generate a program $y$ in a target language that is functionally equivalent to $x$. In this paper we are interested in whether a self-generated natural language explanation $z$ can be used to improve this translation process.[2]

### 2.1 Prompt Variations

Fig 1 shows an example of our prompts for program translation. In addition to the **direct** translation baseline (Fig 1, left), we experiment with 3 types of explanations (full prompts are given in Apx C):

1. **exp**: We ask the model to **exp**lain the source program in a few sentences (Fig 1, right).
2. **exp-lbl**: We ask the model to **exp**lain the source program **l**ine **b**y **l**ine. This roughly mirrors the setup in Chen et al. (2023b).
3. **exp-lbl-d**: We ask the model to **exp**lain the source program **l**ine **b**y **l**ine in additional **d**etail. In particular if an individual line is complicated, we ask it to break it down, ex-

---

[1]https://github.com/PootieT/
explain-then-translate

[2]While we focus on natural language explanations in our main experiments, in Apx J we show results with other types of "explanations", such as another pivot language, pseudocode, etc.

```
Standard Translation (baseline): direct

Please translate the following Python code
to R?

### Python version

def largest_divisor(n: int) -> int:
    for i in reversed(range(n)):
        if n % i == 0:
            return i

### R version

largest_divisor <- function(n) {
    for (i in rev(seq(n))) {
        if (n %% i == 0) {
            return(i)
        }
    }
}    // Translation is incorrect!
```

```
Explain-then-Translate (ours): exp

Can you explain what this Python program
does with a couple of sentences? The goal
with the explanation, is so that a reader
can easily rewrite the program in R.
### Python version
def largest_divisor(n: int) -> int:
    for i in reversed(range(n)):
        if n % i == 0:
            return i
### Explanation
This function takes an integer 'n' and
returns the largest divisor of 'n'. It does
this by iterating over all integers from
'n-1' down to '1' and checking if 'n' is
divisible by each integer. If it is, the
function returns that integer. If no divisor
is found, the function returns '1'.
### R version
largest_divisor <- function(n) {
    for (i in rev(seq_len(n - 1))) {
        if (n %% i == 0) {
            return(i)
        }
    }
    return(1)
}    // Translation is correct!
```

Figure 1: Compared to **direct** code translation prompt, **exp** (ours) prompts models to explain the code before translating. Blue highlights are model completions, and red highlights point out the crucial difference between the two translations. Example prompts and explanations for **exp-lbl** and **exp-lbl-d** in Apx C, E.19

plain individual fragment of the line, and then summarize the purpose for the entire line. This prompting method allows us to decompose compositionally difficult fragments of the code down, re-use individual fragments of explanation before explaining the whole line, similar to Zhou et al. (2022).

When generating explanations, we treat the token sequence \n# as a stopping sequence in order to prevent models from generating target translations (since we condition target program with translated signatures in addition to explanations). Sometimes, a model might generate target-language-specific details (equivalent classes, attempted translation, etc.). In order to control for inconsistencies caused by the target-language-specific explanations, we re-use the same explanations (from Python-Java) for all Python-to-X translation experiments (Section 3.1). Before reusing, we also remove any target specific information with programmatic rules so it can be reused across experiments. For completeness, in Apx G.1 we show the impact of removing target-language-specific details for the **exp** experiments: the effects are generally insignificant, but are more pronounced in low-resource languages.

Additional details on language-specific stop tokens and how few-shot programs are selected are described in Apx E and Apx F, respectively.

## 2.2 Dataset: MultiPL-C2C

MultiPL-E (Cassano et al., 2022) is a benchmark that was recently introduced in an effort to evaluate NL-to-code generation capabilities of language models in 19 different programming languages.[3] Cassano et al. (2022) groups these languages by resource level:

- **High-resource**: JavaScript (**js**), Python(**py**), Java*[4] (**jv**), C++* (**cpp**), TypeScript* (**ts**)
- **Medium-resource**: PHP (**php**), Ruby (**rb**), C#* (**cs**), Go* (**go**)
- **Low-resource**: Perl (**pl**), R (**r**), Rust* (**rs**), Scala* (**sc**), Swift* (**sw**)
- **Extremely-low-resource**: Bash (**sh**), Lua (**lua**), Racket (**rkt**), Julia* (**jl**), D* (**d**)

_______

[3]Concretely, taking the original HumanEval (Chen et al., 2021) and MBPP (Austin et al., 2021) datasets (where models are prompted with problem description and are tasked to generate a Python program that solves the problem), MultiPL-E built transpilers for the unit tests as well as code generation prompts such that models can be evaluated from NL-to-code in 19 different languages (Python + 18 additional languages).

[4]*: indicates statically typed language (vs. dynamically)

| Res | High | | | | Medium | | | | Low | | | | | Extremely-Low | | | | |
|---|---|---|---|---|---|---|---|---|---|---|---|---|---|---|---|---|---|---|
| Trial | js | cpp | jv | ts | php | rb | cs | go | pl | r | rs | sc | sw | sh | lua | rkt | jl | d |
| direct(0) | 85.5 | 76.6 | 76 | 78.9 | 68.4 | 78.3 | 79.2 | 42.4 | 58.3 | 40.4 | 70.6 | 63.7 | 64.4 | 51.3 | 56.2 | 31.3 | 61.6 | 42 |
| exp(0) | 84.8 | 82.1 | 82.9 | 85.1 | 77 | 78.3 | 83.6 | 45.7 | 68.2 | 46.9 | 74.8 | 74.7 | 70.5 | 55.4 | 60 | 41.3 | 70.5 | 44.4 |
| exp-lbl(0) | 84.3 | 80.3 | 82.5 | 84.7 | 77.5 | 80 | 83.8 | 45.6 | 65.3 | 48 | 76 | 74.5 | 75.1 | 56.8 | 61.5 | 39 | 69.2 | 44.4 |
| exp-lbl-d(0) | 83.2 | 79.1 | 84 | 85.2 | 77.6 | 82.7 | 82.7 | 45.8 | 62.9 | 45.5 | 76.2 | 73.8 | 74.8 | 56.4 | 59.5 | 36.1 | 68.9 | 43 |
| direct(4) | 86.6 | 84.3 | 85 | 86.5 | 79.7 | 82.9 | 85.7 | 52.4 | 69.8 | 56.4 | 76.4 | 76.7 | 72.7 | 74.8 | 67.7 | 43.2 | 65.3 | 45.6 |
| exp(4) | 87.3 | 77.8 | 82.1 | 88.4 | 81.7 | 80.4 | 83.3 | 51.4 | 73.2 | 55.6 | 73 | 73.7 | 67.1 | 73.1 | 67.9 | 44.7 | 69.4 | 45.9 |
| exp-lbl(4) | 87.6 | 82.4 | 85.3 | 87.1 | 81.5 | 80.1 | 84.7 | 50.8 | 72 | 56.9 | 74.9 | 73.9 | 70.1 | 71.9 | 68.6 | 45.6 | 72.1 | 46.2 |
| exp-lbl-d(4) | 87.8 | 83.9 | 86.5 | 88.3 | 82 | 80.7 | 84.3 | 50.6 | 70.9 | 57 | 76.3 | 75.5 | 70.9 | 72.5 | 69.4 | 45.3 | 71.1 | 45.1 |
| exp*(4) | 88.9 | 80.8 | 85 | 88.3 | 78.6 | 81.4 | 85.1 | 52.2 | 71.6 | 57.3 | 74.5 | 76.1 | 71.3 | 73.1 | 67.9 | 46.2 | 72.9 | 50.7 |
| exp-lbl-d*(4) | 87.9 | 84.6 | 87.8 | 88.5 | 81.5 | 80.6 | 85.3 | 51.6 | 70.1 | 57 | 78.7 | 77.7 | 71.9 | 72.3 | 69.4 | 46.3 | 71.6 | 44 |

Table 1: Translation pass@1 from Python to X. * uses heuristically selected explanations (Sec 3.4). Parenthesis in **trial** indicates # of shots. Best within same-shot (no heuristics) is underscored and overall best is in **bold**.

To repurpose MultiPL-E into a code-to-code translation dataset, we change the task formulation by including canonical Python programs in the prompt and removing the NL problem descriptions. We dub this version of the dataset as MultiPL-C2C, and release it for future work in this area.[5]

## 2.3 Metrics

We evaluate our methods using unit test pass rate (Chen et al. 2021; Cassano et al. 2022) as string match-based evaluations do not capture the diverse ways in which a program can be translated and still be functionally equivalent to the source. We calculate the pass rate as:

$$\text{pass@}k := \mathop{\mathbb{E}}_{\text{Problems}} \left[ 1 - \frac{\binom{n-c}{k}}{\binom{n}{k}} \right]$$

where $n$ is the total number of generations, and $c$ is the number of correct generations. The best sampling temperature $t$ (or top-p) (Holtzman et al., 2020) is often dependent on $n/k$, where smaller temperatures are best for small $n/k$, while larger temperatures increase the generation diversity (better recall) and can improve pass rate with large $n/k$. We prioritize precision and calculate pass@1 with $n = 20, t = 0.2$, and top-p$= 0.95$ following Cassano et al. (2022).

## 2.4 Models

We evaluated four models of varying sizes. We main report the results from GPT-3.5 (`gpt3.5-turbo-0301`) in the main paper unless otherwise specified, and defer the results from open source models (CodeGen2-1B, CodeGen2-16B, and Llama2CodeInstruct-34B (Nijkamp et al., 2023; Rozière et al., 2023)) to the appendix.

---

[5]We also considered into CodeXGLUE (Lu et al., 2021) and TransCoder (Lachaux et al., 2020) for the unit tests evaluations, but initial studies suggested a significant number of examples (more than 25%) contain mistakes in gold programs or inadequate unit tests (see Apx A, B).

## 3 Experiments and Discussion

In our study we focus on two main sets of translation directions: Python-to-X, where we translate from Python to 18 other target languages ranging from high to extremely-low-resource (§3.1), and X-to-X, where we target a representative set of translation directions varying source and target language resource levels and typing characteristics (§3.2). We analyze translation improvements across models of 4 different sizes (§3.3) and discuss improving individual explanations through heuristics (§3.4). Finally we show our method improves more on difficult-to-translate examples (§3.5) and provide ablations to understand what NL explanations improves performance and whether alternative self-generated contexts could help (§3.6).

### 3.1 Python-to-X Translation

In Table 1 we present results of the Python-to-X experiments in the zero- and four-shot settings with GPT-3.5. Results with open-source models results show similar trends and are shown in Apx G.9.

**Natural language explanations improve performance in the zero-shot setting, and this effect is more pronounced in low-resource languages.** Providing explanations improves relative performance by 11.5% on average across 18 target languages. Regardless of the target language resource level, the best explanation improves translation with average relative improvement of 6.2% in high-resource languages and 14.5% in extremely-low-resource languages. There is no significant difference between improvements on translating into statically vs. dynamically typed languages. Self-generated explanations even slightly outperform human-written doc-string instructions that are part of the original HumanEval dataset (see Apx J).

**High-resource target languages benefit from detailed explanations while low-resource alternatives benefit from abstract explanations.** We

hypothesize that high-resource languages benefit from more detailed explanations due to higher co-occurrences of NL and PL in the pretraining corpora; whereas in low-resource languages we speculate the additional detail may introduce spurious correlations. Since we re-use explanations across translation directions, the translation performance difference can be attributed only to the code generation step.

**Natural language explanations are less helpful in the few-shot setting, but good few-shot examples are crucial.** In the four-shot setting, the average improvement is much less at 1.1%, although some language pairs observe as much as a 10.1% improvement. Average improvement in high-resource languages (1.2%) is smaller than that in extremely-low-resource languages (3.4%). The most detailed explanations perform the best in 12 out of 18 language directions amongst explanation types. This is likely due to the carefully curated few-shot examples, which are semantically and syntactically complex enough to benefit from decomposition and explanations (see in Apx F for more details).

**Few-shot explanations result in worse performance than zero-shot explanations.** The most abstract explanation (**exp**) performs the worst (best in only 3 out of 18 directions) in the few-shot setting. Since we source the few-shot explanations from minimally modified zero-shot explanations, including these self-generated explanations simply restricts the model's explanation to follow stylistic patterns and decreases the diversity of the explanations. In Apx G.2, we disentangle the effect of target specific explanation and zero/four-shot setting to provide further evidence to this point.

**Improvements in the zero-shot setting correlate with improvements in the few-shot setting.** Except for a few outliers, Fig 2 shows a good correlation. This is interesting because few-shot is manually curated and written in PL, while explanation is self-generated and written in NL. In our ablation 3.6 and Apx J we further analyze to what extent the source of information provides the structure of the output, and whether the correctness of the sequence actually matters.

**Additional analyses.** In the appendix we provide the breakdown of error types (Apx G.4, G.5), source program lengths (Apx G.6), qualitative

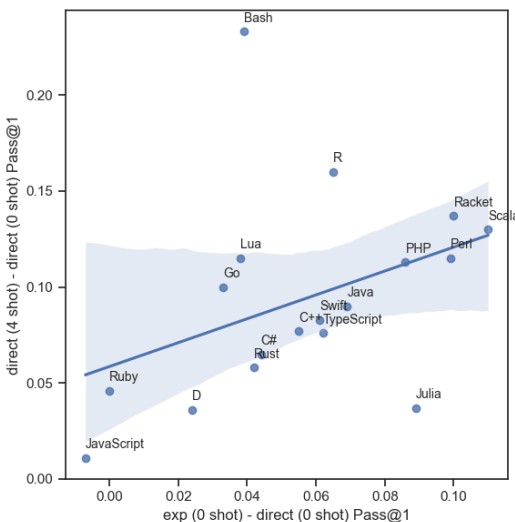

Figure 2: Zero-shot **exp** improvements correlate with few-shot improvements over baselines ($r^2 = 0.151$).

analysis of explanations (Apx E.19), pass@10 (Apx G.7), correlation between NL-to-code vs. translation results (Apx G.8). and improvements vs. translation difficulty (Sec 3.5, Apx I).

### 3.2 Alternative Translation Directions

To understand whether our findings only hold for Python (or only high-resource languages), we experiment on additional translation directions from different resource groups and typing characteristics, and present our results in Table 2. Since source languages are different, we do not re-use explanations. In the four-shot explanation, we use zero-shot generated explanations (3.1). In the following section, we have **High**=high-resource languages and **Ext-Low**=extremely-low-resource languages. Results from open-source model results are in Apx H.

**High-to-ExtLow and High-to-High follow a similar patterns as Python-to-X.** In zero-shot, High-to-High has varied performance across different explanation types, whereas High-to-ExtLow benefits mostly from simple explanations (**exp**). In four-shot, there is little to no improvements in High-to-High, but some improvements in High-to-ExtLow.

**ExtLow-to-High: Models are poor at explaining low-resource language programs.** The improvement in ExtLow-to-High trials is limited in both zero- and four-shot directions. Across explanation methods, we can see a general decrease in performance from high-level (**exp**) to detailed (**exp-lbl-d**) explanations. We speculate that LLMs generally struggle to understand and explain lower-resource PLs; more details may introduce more errors which

| | | | 0 shot | | | | 4 shot | | | |
|---|---|---|---|---|---|---|---|---|---|---|
| Resource | Type | src-tgt | direct | exp | exp-lbl | exp-lbl-d | direct | exp | exp-lbl | exp-lbl-d |
| High-to-High | D-D | py - js | 85.5 | **85.7** | 84.3 | 83.2 | 86.6 | 86.1 | 87.6 | **87.8** |
| | D-S | js - jv | 77 | 77.3 | 82.5 | **83.5** | **91.5** | 90 | 89.3 | 86.7 |
| | S-D | cpp - py | **92.3** | 90.2 | 90.5 | 91.3 | **93.9** | 89.9 | 89.4 | 89.2 |
| | S-S | jv - cpp | 77 | 79.8 | **83** | 79.8 | **82.3** | 81.1 | 78.4 | 78.7 |
| High-to-ExtLow | D-D | js - rkt | 30.2 | **41.9** | 41.5 | 37.9 | 44.7 | **51.8** | 49.8 | 46.3 |
| | D-S | py - d | 42 | **42.7** | 44.4 | 43 | 45.6 | **48.5** | 46.2 | 45.1 |
| | S-D | cpp - lua | 69.2 | 71.9 | 72.5 | **75** | 79.2 | 77.2 | 75.1 | 74.9 |
| | S-S | jv - jl | 60.2 | **75.4** | 68.8 | 72.2 | 72.7 | **72.9** | 72.4 | 71 |
| ExtLow-to-High | D-D | lua - py | **89.5** | 85.9 | 86.5 | 84 | 88.5 | **89.3** | 88.4 | 83.2 |
| | D-S | rkt - jv | 65.9 | **77.1** | 74.4 | 71.7 | **86.4** | 84.2 | 80.8 | 79.4 |
| | S-D | jl - js | 83.1 | **83.5** | 82 | 80.4 | 87.4 | **90** | 87.1 | 88.3 |
| | S-S | d - cpp | **88.4** | 81.4 | 80.5 | 85 | **88.4** | 86.3 | 81.7 | 85.5 |
| ExtLow-to-ExtLow | D-D | lua - rkt | 29.6 | **45.2** | 38 | 37.7 | 49.2 | 49 | **50** | 48.7 |
| | D-S | rkt - jl | 63.3 | 64.3 | **67.7** | 62.1 | 71.7 | 70.4 | **73** | 70.5 |
| | S-D | d - lua | 68.4 | 69.4 | 66.8 | **69.9** | 71.7 | 70.3 | 72.5 | **73.5** |
| | S-S | jl - d | 41.6 | 43.4 | **43.6** | 41.4 | 43.8 | **44.2** | 43.8 | 39.8 |

Table 2: Translation pass@1 between 16 different pairs of languages. **Resource** indicates the language resource levels of the source and target. **Type** indicates the source and target language typing characteristics (D/S=dynamically/statically typed). The best runs within the same-shot setting are in **bold**.

| Exp Type | src-tgt | **random** | **len** | **line-e** | **line** | **frag** | **logprob** | **oracle** | **pass@1** |
|---|---|---|---|---|---|---|---|---|---|
| exp | py-jv | $83.0 \pm 1.5$ | 84.2 | 84.8 | 84.2 | 81.0 | **85.4** | 93.7 | 85.4 |
| exp-lbl | py-jv | $85.1 \pm 1.0$ | 84.4 | 84.4 | 84.4 | **85.1** | 83.8 | 90.3 | - |
| exp-lbl-d | py-jv | $86.1 \pm 1.2$ | 86.4 | 85.1 | 85.1 | **88.3** | 87.0 | 92.2 | 87.6 |
| exp | py-rkt | $41.1 \pm 2.0$ | 41.0 | 39.8 | 39.8 | 41.6 | **43.5** | 70.2 | 40.7 |

Table 3: Explanation selection heuristics performance. We estimate heuristics performance (pass@1, n=1) and validate the best method by generating 20 programs and calculate **pass@1** ($n = 20$). No heuristic is able to outperform baseline with **exp-lbl** so we did not verify with **pass@1**.

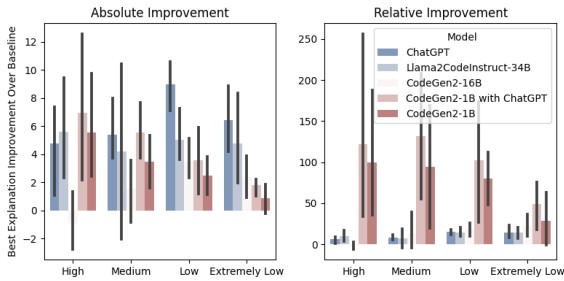

Figure 3: Py-to-X translation (pass@1, zero-shot) improvements (best explanation over baseline) across models grouped by target language resource level.

may compound into the translation phase.

### 3.3 Comparisons Across Different LMs

**Improvements are robust across models.** From Fig 3 and 9, we can see that in general, the larger the model, the larger the absolute improvement with self-generated explanations. In terms of improvement over resource levels, our method improves low-resource language generations more with larger models, while improving high-resource languages more with smaller models. See detailed result tables in Apx H and G.9. CodeGen2-16B is the only model that does not improve consistently with explanations.

**Better explanations are transferable and lead to better translations.** We also experimented with CodeGen2-1B by using GPT-3.5 generated explanations (Fig 3) and found it to improve performance further, outperforming self-generated explanations in 12 out of 18 directions. Comparing absolute improvements against CodeGen2-1B with self-explanations, we find that GPT-3.5-generated explanations improve more when generating higher resource than lower resource languages, indicating that smaller models are less sensitive to improvements. More analyses are given in Apx G.9.

### 3.4 Explanation Selection with Heuristics

In the context of chain-of-thought prompting, Wang et al. (2022) demonstrate the importance of sampling diverse "reasoning" paths. It is difficult to ensemble sampled programs from language models, but we find sampling diverse explanations, where we first sample 20 explanations and then sample one program each, to improve recall for correct programs (pass@10) than sampling 20 programs from 1 explanation, or direct translation in zero/four-shot settings. This indicates that there is significant room for improvement if we are to be able to select the best explanation that can gener-

ate a correct program with the highest probability (**oracle** column in Table 3).

Motivated by the potential of diverse explanations to improve translation results, we explore five explanation re-ranking heuristics: 1) **len**gth of explanation (in characters) excluding code; 2) lines of source code explained (**line-e**); 3) number of **line**s of explanations; 4) number of code **frag**ments enclosed in ' '; [6] 5) with **logprob** (Zhang et al. 2022; Min et al. 2022a), ranking the explanations with a weighted combination of $\alpha * p(\text{code}|\text{explanation}) + (1 - \alpha) * p(\text{explanation}|\text{code})$ using CodeGen2 (Nijkamp et al., 2023) (more details in Apx L).[7]

For each explanation type, we generate 20 explanations and 1 program from each explanation (**train set**). We estimate each heuristics' performance by averaging the pass rates of its selected (argmax) explanations for each individual problem in the **train set**.[8] For **random** baseline, we select 1 explanation for each program randomly;[9] and for **oracle**, we select the explanations with the highest pass rates in the **train set**. For each explanation type, we pick the heuristics with the best estimated pass@1 ($n = 1$), and generate 20 programs from these explanations for **pass@1** ($n = 20$) score (right most column in Table 3). We use zero-shot explanations for **exp** (see Sec 3.1) and four-shot for **exp-lbl** and **exp-lbl-d**. Our main results are shown in Table 3, from which we observe the following.

**Heuristics can improve performance, and this is robust across different target languages.** With **exp**, **logprob** improves upon **random** by absolute 2.54% ($p = 0.055$),[10] and **frag** improves **explain-lbl-d** upon random baseline by absolute 2.2% ($p = 0.033$) with simulation. Both improvements can be reproduced with **pass@1**, so we include these heuristically selected explanations as two additional rows in Table 1. With **logprob** selected **exp**, we improve or match performance in 15/18 directions, with an average improvement of 1.7% ($p < 0.001$). With **frag** selected **exp-lbl-simp**, we improve or match performance in 13/18 directions, averaging 0.48% ($p = 0.022$). See Apx G.3 for more comparisons.

---

[6] Markdown pattern for referencing code

[7] We tried scoring with GPT-3.5 directly as well but found it to not outperform the random baseline (Apx K)

[8] We discuss the trade-offs of alternative settings (sampling 4 explanations and 5 programs from each) in Apx M.

[9] Repeated 100 times to obtain mean and variance.

[10] The resulting $p$-values are from a one-tailed paired $t$-test.

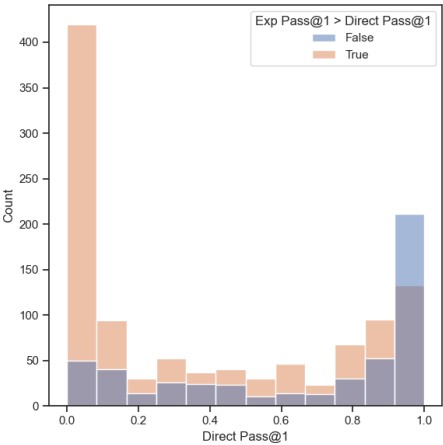

Figure 4: We count explanation improvement cases over direct pass@1. Results include all trials between Python-to-X and X-to-X directions. For better contrast, all problems with the same exp pass@1 and direct pass@1 are removed.

**Some heuristics generalize across explanation types.** Only **frag** and **logprob** perform robustly. Intuitively, **frag** makes sense because data containing a piece of code and an explanation is more likely to be correct if the author refers to the code more frequently. With **logprob**, since we are effectively measuring the mutual information between codes and explanations (Zhang et al., 2022).

**There is still ample room for improvement.** As we can see in the difference between **oracle** and **pass@1**, no heuristics is able to to come close to this **oracle** upper bound. This gap is much larger in high-to-low-resource translation direction (**py-rkt**). Qualitatively, we could not distinguish a good explanation from a bad one manually (Apx L.3 and L.4), suggesting that the distinction between "good" and "bad" explanations may be hidden due to stylistic noise (wording, spacing, etc.), or potentially due to chance.

## 3.5 Which programs benefit from explanations?

To understand where natural language explanations benefit most, we investigate how **exp** improvement varies across problem hardness, which is approximated through **direct** translation pass@1. Through Fig 4, we discovered that explanation improves difficult problems (left of x-axis), but could hurt easy problems (right of x-axis). This potentially suggests we could determine which program to explain using a hardness threshold, and improve performance further. We verified the validity of such approach through our oracle metric (**direct** pass@1),

and show the full results in Apx I. We found selective explanation to improve performance over **direct** and **exp** in 35/36 directions. We leave building such difficulty-based problem selector for future work.

## 3.6 Ablation Studies

We perform additional ablation studies to understand what aspects of the explanations improve translation (§3.6.1), whether explanations are robust to variations in surface semantics/readability of the source code (§3.7, Apx N), and if self-generated context in PL could help few-shot examples (§3.8, Apx J). Additionally, we explore the relationship between context length and improvements in Apx G.6.

### 3.6.1 Explanation Ablation

We select 4 target languages of different resource levels where explanations provide the most improvements (zero-shot) for Python-to-X. With each explanation, we ablate in following ways:

**swap-s:** We randomly reorder sentences (**exp**) or code-explanation segments (**exp-lbl**) to test if *explanation provides structural supervision.*

**obf-exp:** We obfuscate source programs (see examples in Apx N), where function and variable names are replaced systematically with templates (`FUNC_0`, `VAR_1`, etc.). This tests *whether an explanation uses specific variable references (structural supervision at token level).*

**ret-exp, rand-exp, no-exp:** We replace the explanation with another program's explanation randomly (**rand-exp**), or through retrieval (**ret-exp**, details in Apx O), or with an empty string (**no-exp**) to verify *if explanations need to be correct/relevant.*

**del-w:** We randomly remove some of the words and see *if fluency (i.e. high logprob) is necessary.*

**del-s:** We randomly remove a percentage of sentences (**exp**) or code-explanation paragraphs (**exp-lbl**) to see the *dependency of the translation on the completeness of the explanation.*

**Explanation needs to be coherent, relevant, and correct.** From what we can observe in Table 4, explanations do not provide much structural guidance (**swap-s**), and models do not overly rely on their completeness (**del-s**). Models do not rely on the surface form to redirect attention as much (**obf-exp**), but they do require explanations to be fluent

|           | jv    | php   | sw    | rkt   |
|-----------|-------|-------|-------|-------|
| exp       | **6.9** | 8.6 | 6.1 | **10** |
| exp-lbl   | 6.5   | 9.1   | 10.7 | 7.7 |
| swap-s (exp) | 5.1 | 7.9 | 4.5 | 7.6 |
| swap-s (lbl) | 5.8 | 9.2 | 9.3 | 3.5 |
| obf-exp (exp) | 3.8 | **9.3** | **7.9** | 7.3 |
| obf-exp (lbl) | 5.3 | 9.8 | 9.6 | 5.2 |
| del-s-0.25 (exp) | 5.3 | 7.7 | 4 | 5.4 |
| del-s-0.5 (exp) | 5.6 | 7.2 | 4 | 7.1 |
| del-s-0.25 (lbl) | 6.7 | 7.7 | 8.5 | 6.2 |
| del-s-0.5 (lbl) | 5.2 | 7.7 | 7 | 4.3 |
| del-w-0.25 (exp) | 1.8 | 7.7 | 3.4 | 6 |
| del-w-0.5 (exp) | 0.1 | 3.3 | -1.1 | 5.8 |
| del-w-0.25 (lbl) | 4.5 | 6.5 | 5.1 | 3.3 |
| del-w-0.5 (lbl) | 0.8 | 6.5 | 3.2 | 3 |
| rand-exp (exp) | 3.8 | 6.1 | 3.1 | 5.1 |
| ret-exp (exp) | -0.2 | 5.3 | 1.8 | -0.2 |
| no-exp (exp) | -1.5 | -54.1 | -10 | 3 |
| rand-exp (lbl) | 3.6 | 7.7 | 7.4 | 4 |
| ret-exp (lbl) | 0.8 | 5.2 | 5.4 | -1.2 |
| no-exp (lbl) | -2.7 | -48.2 | -3.9 | 0.4 |

Table 4: Performance improvements over the baseline with various explanation variations from Python-to-X (see §3.4). **lbl=exp-lbl**. **del-w-0.5**=deleting 50% of words at random. Best (ablated) **exp** in **bold** and best (ablated) **exp-lbl** underlined.

(**del-w**). Lastly, when models receive completely irrelevant explanations (**rand-exp**), they are able to recover performance to some extent; but if the explanations are convincingly misleading (**ret-exp**) performance deterioates.

**Models rely on semantics of explanations less when generating lower-resource languages.** Different types of ablations affect lower-resource languages more uniformly than higher-resource languages. Relative to **exp/exp-lbl**, ablations that completely alter the semantics of the explanations (**del-w**) decreases improvements less in lower-resource languages than higher counterparts, while ablations that keep overall semantics of the explanation (**swap-s**) decreases improvements less in higher-resource languages.

**Semantics of explanation is not the only picture.** Despite explanations having completely wrong semantics (**rand-exp**, **ret-exp**), models still improve from the added context. CoT self-generated reasoning has been found to follow unmentioned/hidden biases within the context (Turpin et al., 2023). It would be interesting to investigate further what remaining biases (if any) contribute to the improvements in program translation.

## 3.7 Source Program Ablation

To test whether our explanation methods work with a different distribution of source programs, we ob-

|         | quality | jv   | php  | pl   | rkt  |
|---------|---------|------|------|------|------|
| direct  | -       | 76   | 68.4 | 58.3 | 31.3 |
| fixed   | gold    | 82.1 | 78.5 | 70.1 | 41.7 |
| fixed   | unk     | 82.2 | 74.3 | 63.7 | 39.1 |
| fixed   | bad     | 78.4 | 75   | 67.2 | 40.8 |
| # problem | -     | 158  | 161  | 161  | 161  |
| retrieve  | gold  | 83.5 | 78.9 | 68.4 | 46.7 |
| # problem | -     | 158  | 161  | 161  | 161  |
| retrieve  | unk   | 81.9 | 76.5 | 67.2 | 38.3 |
| # problem | -     | 138  | 80   | 132  | 154  |
| retrieve  | bad   | 67.2 | 61.3 | 54.3 | 36.1 |
| exp     | -       | 82.9 | 77   | 68.2 | 41.3 |

Table 5: Translation performance using different source programs and quality of target as one-shot example. **fixed** indicate fixed example for one-shot, and **retrieve** uses BM25 retrieved program as one-shot example. **quality** indicates the correctness of the target program in the one-shot. **unk** is any program output sampled randomly from GPT-3.5 and **bad** is programs sampled from the incorrect pool. Since not every retrieved problem has incorrect (or correct) generations, we report the **# problems** evaluated for each retrieval setting.

fuscate variables and funciton names source programs, removing all surface form semantic information (Apx N). When comparing **direct** translation vs. **exp**, in Table 18, we find explanations to be robust regardless of surface semantics of the code. In half the trials, relative improvements using explanation are even higher for obfuscated source code than original code. This is potentially due to the fact that explanations become more reliant on actual syntax of the program, rather than hallucinating on the program semantics from surface variable names. This is promising because when using models in the real world, such as for app modernization, there is no guarantee of having readable code.

### 3.8 Few-shot Ablations

Since few-shot improvements correlate with explanation improvements (§3.1) we conduct ablations to check how sensitive the models are to the correctness of few-shot examples, and whether unverified self-generated few-shots can also improve as well as explanation does. Here, we replace our correct few-shot examples with naturally generated programs from GPT-3.5 (high logprob, but formally unverified (**unk**) or incorrect (**bad**)), and observe how much self-generated few-shots improve translation and models' sensitivity towards their correctness. We experiment with a fixed one-shot example as well as retrieval one-shot to observe the improvement/sensitivity when the exemple program is similar or different from the testing program.

**When the few-shot program is similar, verification is important.** In Table 5, we observe that when the retrieved one-shot is paired with a wrong target program, the decrease in performance is much more significant than in the fixed-shot setting.[11] In other words, curated few-shots are robust to label noise. This is consistent with the earlier conclusion in Table 4 that an "almost-correct" explanation (**ret-exp**) could influence generation more than when it is obviously incorrect (**rand-exp**). If verification is available, **retrieve-gold** shows that a formally correct (similar) program is more useful than a natural language explanation. However, on average, self-generated *unverified* explanations (**exp**) still outperform one-shot in all directions (**fixed-unk** by 0.7-4.5%; **retrieve-unk** by 0.5-2.0%), indicating that NL generations often have higher quality than programs and can serve as a better medium for intermediate reasoning step.

To further compare NL explanations with other formal/non-formal reasoning steps, in Apx J, we experiment with translating to a pivot programming language before translating to the target language (e.g. Python-Java-PHP). By controlling the pivot language correctness, we can more closely verify the model's translation performance sensitivity to correctness in context. The result indicates mistakes in intermediate PL steps corrupt translation performance more than imperfect NL explanations. This indicates that using self-generated NL as an intermediate step often helps more than self-generated PL, and reasoning in the NL space is advantageous for language models.

## 4 Related Work

**Explanation in deep learning.** Many works have explored using explanations to improve language models. Hase and Bansal (2022) investigate various ways explanations can be introduced in modeling and find it most useful for retrieving similar data. Joshi et al. (2022) find explanation regularization to improve OOD performance. Most works in LLMs generate explanation using zero-shot, few-shot, or finetuning, before generating the target response (Ling et al., 2017; Nye et al., 2021; Wei et al., 2022; Mukherjee et al., 2023; Hsieh et al., 2023). A few works have also explored post-hoc explanations (Lampinen et al., 2022; Krishna

---

[11]**retrieval-bad** (Table 5) should be taken lightly since the subset of problems evaluated for **bad** also likely contains harder problems.

et al., 2023). Wiegreffe et al. (2021) and Chan et al. (2022) propose metrics to quantify rationale-label association. We refer readers with further interest to surveys (Miller, 2019; Hartmann and Sonntag, 2022; Zhao et al., 2023).

**Language model for code.** Much work has been dedicated to applying transformer-based language models to NL and PL generation (Brown et al., 2020; Ahmad et al., 2021; Chen et al., 2021; Li et al., 2022; Ouyang et al., 2022). TransCoder leverages unsupervised pretraining and supervised finetuning to build one of the first neural transpilers (Lachaux et al., 2020, 2021; Roziere et al., 2021). Later works obtain impressive zero and few-shot learners by simply pretraining on NL and PL data with language modeling or derivative training objective (Ahmad et al., 2021; Nijkamp et al., 2022; Chen et al., 2021; Scao et al., 2022; Xu et al., 2022; Nijkamp et al., 2023; Allal et al., 2023; Li et al., 2023; Rozière et al., 2023; Athiwaratkun et al., 2022).

**Intermediate state prompting.** The Explain-then-Translate approach is an instance of chain-of-thought prompting (Wei et al. 2022; Nye et al. 2021), where the model is prompted to generate reasoning steps before the final answer. Follow-up works have found it to be useful on numerous tasks outside of niche reasoning tasks (Wang et al., 2022; Zhou et al., 2022; Chowdhery et al., 2022; Suzgun et al., 2022; Yao et al., 2023). In our setting, we find most improvements to come from the zero-shot setting (Kojima et al., 2022). Different from previous works, our task focuses on program translation, with significant token level correspondence between the source and target. Ghazvininejad et al. (2023) and Lu et al. (2023) improve NL translation by augmenting prompts with dictionary translations, but their contexts are not self-generated. It would be interesting to explore whether other forms of "explanations" (e.g., BNF grammars (Wang et al., 2023a)) could further improve performance, especially on low-resource languages which may not have been encountered frequently during pre-training.

**Code prompting and feedback.** In the code-generation space, Zelikman et al. (2022) incorporate model-generated rationales given question-answer pairs as part of fine-tuning to improve model reasoning capabilities. Jiang et al. (2023) use few-shot examples to teach models to create NL steps from NL instructions before generating the code. Zelikman et al. (2023) and decomposes complex problems in NL and generated/verified subproblems to achieve high performance in NL-to-code. Chen et al. (2023a) finetune policy models to correct code given human critique. Wang et al. (2023c) searches multiple hypothesis in NL before generating PL targets. Our method uses self-generated context without overly relying on feedback, few-shot examples, or complicated frameworks, and is targeting code-translation specifically instead of NL-to-code generation. Chen et al. (2023b) briefly mentions in their ablation studies that explanation improves translation in Python→C++, but our analysis reveals a more nuanced settings in which explanations improve code translation.

## 5 Conclusion

This work conducts a thorough analysis of the performance of large language models in program translation by using different types of self-generated explanations as an intermediate step. Models generate higher quality detailed explanations for high-resource languages, while still generating good enough abstract explanations for low-resource languages. With simple heuristics, we have also demonstrated the potential to improve explanation quality and consequently translation quality. We identify key requirements for explanation and find that on average, mistakes in NL are less detrimental to performance, and do not require verification to perform well compared to using PL as self-generated contexts.

## Limitations

There are several limitations to our work. First, while we focus on the (adapted) MultiPL-E benchmark due to its widespread use, it is unclear whether programs in this benchmark are representative of programs that are targets for code-to-code translation. Second, while we saw our conclusions to largely hold across GPT-3.5 and other open-source models, it is unclear whether they will hold for more powerful LLMs such as GPT-4. Finally, somewhat disappointingly we found natural language explanations to be not as helpful in the few-shot setting, and insofar as obtaining several demonstration examples for each language pair is quite practical, natural language explanations for code-to-code translation may not actually be useful for many applications of interest.

## Acknowledgements

We thank all anonymous reviewers. We also thank Justin Weisz, Afra Feyza Akyürek, Najoung Kim, Aditya Yetetore, Garry Kuwanto, Reuben Tan, and Yuwen Pu for their helpful discussions and suggestions. This project is supported by the MIT-IBM Watson AI lab and DARPA HR001118S0044 (the LwLL program). ZT is supported by Lu Lingzi Fellowship from Boston University. Any opinions, findings, conclusions, or recommendations expressed here are those of the authors and do not necessarily reflect the view of the sponsor.

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

## A Transcoder Evaluation Clean-up

TransCoder (Lachaux et al., 2020) and other followups (Chowdhery et al., 2022; Chen et al., 2023b) evaluated their translation model using mined programs from "GeeksforGeek", an online programming practice website. Within the evaluation (valid and test) set, we find more than 25% of the data points were erroneous due to mistakes in gold programs or inadequate unit tests. After

cleaning, we find the dataset to be too easy for our baseline model GPT-3.5 (Apx B) and seek more challenging benchmarks (MultiPL-E).

Within Transcoder validation and test set, in the direction of C++ to Python, and we find several type of errors that are hindering existing evaluations:

**Inadequate test: exact comparison** Python tests on integers and float types are often inadequate in establishing equality. Depending on the library and specific function called, integers or floats can be rounded with different precision. Because tests in TransCoder evaluation dataset work by comparing a gold Python program's output with that of the generated Python program, if the generated program does not use the exact function call, exact equality (==) is not sufficient. Thus, we changed all cases of such violation from checking exact equality to approximate equality like below:

```
# Old way of evaluating exact equality
if f_gold(*parameters_set) == f_filled(*
parameters_set):
    n_success+=1
```

```
# New way of evaluating approximate equality,
borrowed from other instances of float
comparison tests in TransCoder.
if abs(1 - (0.0000001 + abs(f_gold(*
parameters_set))) / (abs(f_filled(*
parameters_set)) + 0.0000001)) < 0.001:
    n_success+=1
```

There are 40+ examples of such case.

**Inadequate test: default large number** In C++ programs, integer variables are often initialized with `INT_MAX` and `INT_MIN` as default values when comparing with other values in a loop. When such values are not replaced and returned, comparing the exact values of such placeholders does not make sense, especially when there is not a single correct way of translating it into Python. Hence, to accommodate such programs, we check for approximate equality between input-output or check that the output of gold and generated program both return an extremely large/small placeholder number.

```
# Old way
if abs(1 - (0.0000001 + abs(f_gold(*
parameters_set))) / (abs(f_filled(*
parameters_set)) + 0.0000001)) < 0.001:
    n_success+=1
```

```
# New way of evaluating approximate equality,
as well as including cases where outputs are
extremely large/small value placeholders.
if abs(1 - (0.0000001 + abs(f_gold(*
parameters_set))) / (abs(f_filled(*
parameters_set)) + 0.0000001)) < 0.001 or ((
```

```
f_gold(*parameters_set) > 10e7) and (f_filled
(*parameters_set) > 10e7)):
    n_success+=1
```

There are about 12 programs for such cases.

**Inadequate test: non-sensible test inputs** Original unit tests sometimes contain non-sensible test inputs. For instance, some programs are intended to process string representation of numbers, but the unit test input would contain non-digit strings. This is especially harmful for translation based on semantics because extreme edge-cases like this should not be used as regular tests. Therefore, we replace these tests inputs with values from the appropriate space. There are 7 instances of such programs.

**Wrong source program** Sometimes, C++ programs are incorrect or non-standardized (such as returning *0*/1 vs returning `true`/`false`, or missing return statements). We standardized all C++ programs. There are around 5 instances of such program we fixed.

**Wrong gold program** Of all errors, this is the most devastating type of error. Without a correct gold program to compare output with, no matter what the generation is there will be no chance of success. Errors range vastly from in-correct range value, wrong indentation, missing return statements, to using wrong/undefined variables. One of the most frequent errors that spans more than 60 programs is the in-correct translation of integer division from C++ to Python. Often it is translated to / when it should be //. We found 100+ examples of this type of error.

We stress that we have only fixed existing issues we have noticed in C++ to Python direction. With sampled programs from Java, we have also observed errors in gold programs and tests frequently.

**Leakage to training data** One of the other reasons that we do not formally evaluate our methods on TransCoder dataset is because that the TransCoder dataset is very likely to have been included in GPTs' training corpora. We are often able to generate the entire program in test/valid split by only providing the signature. The unique detokenized program pattern makes it easy for the model to detect and regurgitate from training sequences.

## B  TransCoder Evaluation

We report the full evaluations on TransCoder eval+test set with GPT-3.5 in Table 6. The model

| | Original | Fixed |
|---|---|---|
| **Other work** | | |
| Codex | 80.4 | ? |
| self-debug | 92.5 | ? |
| **Our work** | | |
| direct (no-sig) | 80.8 | 90 |
| exp (no-sig) | 69.1 | 77 |
| direct (gold-sig) | 81 | 88 |
| exp (gold-sig) | **81.7** | 89.9 |
| direct (typed-sig) | 80.6 | **90.8** |
| exp (typed-sig) | 80.2 | 90.6 |

Table 6: TransCoder evaluation with the original and fixed dataset, C++ to Python, total of 567/566 programs). Entries under our work are pass@1(n=1).

improves around (absolute) 10% on the fixed dataset, indicating the lack of difficulty in the program. Through manual inspection we find the syntax of the programs to be rudimentary. We conducted three type of evaluations regarding the amount of target program signature specification we provided:

- **no-sig**: we only prompt the beginning of Python program generation with def. This signals the beginning of python program without specifying any part of the signature. This is what we use to compare with Self-debug (Chen et al., 2023b) baselines.

- **gold-sig**: we use the gold programs signature to prompt the rest of Python generation. Some of the programs are translated incorrectly after being prepended with signature because a few Python gold programs contain non-equivalent program name and input variable name. This is the same evaluation setting in which we conducted other experiments in HumanEval, and is our main source of comparison.

- **typed-sig**: In addition to prompting target program with gold programs, which do not contain Python type hints, we built rule-based transpiler to translate the C++ program signatures to Python with type hints, and prompt the rest of the program.

We do not have direct measurement of the self-debug method on the fixed dataset. We can, however, infer that self-debug method's improvements over baseline is from the additional access to unit

test and compiler feedback. When unit tests or gold programs are wrong, relying on these feedback information is the **only** way to improve translation performance. In another word, translation models start to deviate from a faithful translation and "over-correct" itself to pass unit tests. Our methods, however, assumes no such access to such feedback signals, and slightly under-perform.

When looking at **gold-sig** trials, we see that by asking model to explain the program and then translate consistently improve over baseline. In **no-sig** trials, we see a consistent under-performance of our methods. Most of the errors are result of generated program having incorrect signature (incorrect number of input variables). Often, these source C++ programs contain non-intuitive or redundant input variables. For example, in the following example FREQUENT_ELEMENT_ARRAY_1, the input variable n is non-intuitive. If a program were to find the most frequent element in a array, it should normally process the whole input array arr.

```python
# program generated through direct
translation
def f_gold ( arr , n ) :
    Hash = dict ( )
    for i in range ( n ) :
        if arr [ i ] in Hash.keys ( ) :
            Hash [ arr [ i ] ] += 1
        else :
            Hash [ arr [ i ] ] = 1
    max_count = 0
    res = - 1
    for i in Hash :
        if ( max_count < Hash [ i ] ) :
            res = i
            max_count = Hash [ i ]
    return res
```

```python
# program generated with explanation
def f_filled ( arr ) :
    hash = { }
    for i in arr :
        if i in hash :
            hash [ i ] += 1
        else :
            hash [ i ] = 1
    max_count = 0
    res = - 1
    for key , value in hash.items ( ) :
        if max_count < value :
            res = key
            max_count = value
    return res
```

By explaining before translating, the model aligns the generation to the explanation, which often follows the distribution of "typical programs" in the wild.

In typed-sig, we see general improvements over other trials. This is expected because we can

provided more accurate information regarding the input-output types. The slight under-performance of **exp** compared to **direct** could be due to small experiment trial size.

## C   Full Prompts

### C.1   Python-Java direct (four-shot)

You are a helpful and faithful compiler that transpiles Python code to Java code. Please translate the following Python code to Java?

### Python version

```python
from typing import Tuple

def even_odd_palindrome(n: int) ->
Tuple[int, int]:
    def is_palindrome(n):
        return str(n) == str(n)[::-1]

    even_palindrome_count = 0
    odd_palindrome_count = 0

    for i in range(1, n+1):
        if i%2 == 1 and is_palindrome(i
):
            odd_palindrome_count += 1
        elif i%2 == 0 and is_palindrome
(i):
            even_palindrome_count += 1
    return (even_palindrome_count,
odd_palindrome_count)
```

### Java version

```java
import java.util.*;
import java.lang.reflect.*;
import org.javatuples.*;
import java.security.*;
import java.math.*;
import java.io.*;
import java.util.stream.*;
class Problem {
    public static Pair<Long, Long>
    evenOddPalindrome(long n) {
        long evenPalindromeCount = 0;
        long oddPalindromeCount = 0;

        for (long i = 1; i <= n; i++) {
            if (i % 2 == 1 & &
            isPalindrome(i)) {
                oddPalindromeCount++;
            } else if (i % 2 == 0 & &
            isPalindrome(i)) {
                evenPalindromeCount++;
            }
        }
        return Pair.with(
        evenPalindromeCount,
        oddPalindromeCount);
    }

    private static boolean isPalindrome
(long n) {
```

```java
        return Long.toString(n).equals(
        new StringBuilder(Long.toString
        (n)).reverse().toString());
    }
}
```

### Python version

```python
from typing import List

def is_sorted(lst: List[int]) -> bool:
    count_digit = dict([(i, 0) for i in
 lst])
    for i in lst:
        count_digit[i]+=1
    if any(count_digit[i] > 2 for i in
lst):
        return False
    if all(lst[i-1] <= lst[i] for i in
range(1, len(lst))):
        return True
    else:
        return False
```

### Java version

```java
import java.util.*;
import java.lang.reflect.*;
import org.javatuples.*;
import java.security.*;
import java.math.*;
import java.io.*;
import java.util.stream.*;
class Problem {
    public static boolean isSorted(
    ArrayList<Long> lst) {
        HashMap<Long, Long> count_digit
        = new HashMap<Long, Long>();
        for (long i : lst) {
            if (count_digit.containsKey
            (i)) {
                count_digit.put(i,
                count_digit.get(i) + 1)
                ;
            } else {
                count_digit.put(i, 1L);
            }
        }
        for (long i : lst) {
            if (count_digit.get(i) > 2)
             {
                return false;
            }
        }
        for (int i = 1; i < lst.size();
         i++) {
            if (lst.get(i - 1) > lst.
            get(i)) {
                return false;
            }
        }
        return true;
    }
}
```

### Python version

```python
from typing import List
```

```python
def separate_paren_groups(paren_string:
 str) -> List[str]:
    result = []
    current_string = []
    current_depth = 0

    for c in paren_string:
        if c == '(':
            current_depth += 1
            current_string.append(c)
        elif c == ')':
            current_depth -= 1
            current_string.append(c)

            if current_depth == 0:
                result.append(''.join(
current_string))
                current_string.clear()

    return result
```

```java
import java.util.*;
import java.lang.reflect.*;
import org.javatuples.*;
import java.security.*;
import java.math.*;
import java.io.*;
import java.util.stream.*;
class Problem {
    public static ArrayList<String>
    separateParenGroups(String
    paren_string) {
        ArrayList<String> result = new
        ArrayList<String>();
        ArrayList<Character>
        current_string = new ArrayList<
        Character>();
        int current_depth = 0;

        for (int i = 0; i <
        paren_string.length(); i++) {
            char c = paren_string.
            charAt(i);
            if (c == '(') {
                current_depth += 1;
                current_string.add(c);
            } else if (c == ')') {
                current_depth -= 1;
                current_string.add(c);

                if (current_depth == 0)
                 {
                    StringBuilder sb =
                    new StringBuilder(
                    current_string.size
                    ());
                    for (Character ch :
                     current_string) {
                        sb.append(ch);
                    }
                    result.add(sb.
                    toString());
                    current_string.
                    clear();
```

```java
                }
            }
        }
        return result;
    }
}
```

```python
from typing import List

def sort_array(array: List[int]) ->
List[int]:
    return [] if len(array) == 0 else
sorted(array, reverse= (array[0]+array
[-1]) % 2 == 0)
```

```java
import java.util.*;
import java.lang.reflect.*;
import org.javatuples.*;
import java.security.*;
import java.math.*;
import java.io.*;
import java.util.stream.*;
class Problem {
    public static ArrayList<Long>
    sortArray(ArrayList<Long> array) {
        if (array.size() == 0) {
            return new ArrayList<Long
            >();
        } else {
            boolean reverse = (array.
            get(0) + array.get(array.
            size() - 1)) % 2 == 0;
            Collections.sort(array);
            if (reverse) {
                Collections.reverse(
                array);
            }
            return array;
        }
    }
}
```

<insert Python program here>

<insert Java completion here>

## C.2 Python-Java exp (four-shot)

You are an helpful AI assistant who
understands all programming languages
and can translate between them at ease.
 Can you explain what this Python
program does with a couple of sentences
? The goal with the explanation, is so
that a reader can easily rewrite the
program in Java?

```python
from typing import Tuple

def even_odd_palindrome(n: int) ->
Tuple[int, int]:
    def is_palindrome(n):
        return str(n) == str(n)[::-1]

    even_palindrome_count = 0
    odd_palindrome_count = 0

    for i in range(1, n+1):
        if i%2 == 1 and is_palindrome(i
):
            odd_palindrome_count += 1
        elif i%2 == 0 and is_palindrome
(i):
            even_palindrome_count += 1
    return (even_palindrome_count,
odd_palindrome_count)
```

### Explanation

This function takes an integer 'n' as
input and returns a tuple of two
integers. The first integer is the
number of even palindromes between 1
and 'n', and the second integer is the
number of odd palindromes between 1 and
 'n'. A palindrome is a number that
reads the same forwards and backwards.

The function first defines a helper
function 'is_palindrome' that takes a
number and returns 'True' if it is a
palindrome and 'False' otherwise. This
is done by converting the number to a
string and checking if the string is
equal to its reverse.

The function then initializes two
counters, 'even_palindrome_count' and '
odd_palindrome_count', to 0. It then
loops through all numbers between 1 and
 'n' (inclusive) and checks if each
number is a palindrome and even or odd.
 If the number is a palindrome and odd,
 it increments 'odd_palindrome_count'.
If the number is a palindrome and even,
 it increments 'even_palindrome_count'.

Finally, the function returns a tuple
of the two counters.

### Java version

```java
import java.util.*;
import java.lang.reflect.*;
import org.javatuples.*;
import java.security.*;
import java.math.*;
import java.io.*;
import java.util.stream.*;
class Problem {
    public static Pair<Long, Long>
evenOddPalindrome(long n) {
        long evenPalindromeCount = 0;
        long oddPalindromeCount = 0;

        for (long i = 1; i <= n; i++) {
            if (i % 2 == 1 & &
            isPalindrome(i)) {
                oddPalindromeCount++;
            } else if (i % 2 == 0 & &
            isPalindrome(i)) {
                evenPalindromeCount++;
            }
        }
        return Pair.with(
        evenPalindromeCount,
        oddPalindromeCount);
    }

    private static boolean isPalindrome
(long n) {
        return Long.toString(n).equals(
        new StringBuilder(Long.toString
(n)).reverse().toString());
    }
}
```

### Python version

```python
from typing import List

def is_sorted(lst: List[int]) -> bool:
    count_digit = dict([(i, 0) for i in
 lst])
    for i in lst:
        count_digit[i]+=1
    if any(count_digit[i] > 2 for i in
lst):
        return False
    if all(lst[i-1] <= lst[i] for i in
range(1, len(lst))):
        return True
    else:
        return False
```

### Explanation

This function takes in a list of
integers and returns a boolean
indicating whether the list is sorted
in non-descending order and has no more
 than two occurrences of any integer.
The function first creates a dictionary
 where the keys are the integers in the
 list and the values are the number of
times that integer appears in the list.
 It then checks if any integer appears
more than twice in the list, and
returns False if so. Finally, it checks
 if the list is sorted in non-
descending order, and returns True if
so, and False otherwise.

### Java version

```java
import java.util.*;
import java.lang.reflect.*;
import org.javatuples.*;
import java.security.*;
import java.math.*;
import java.io.*;
```

```java
import java.util.stream.*;
class Problem {
    public static boolean isSorted(
    ArrayList<Long> lst) {
        HashMap<Long, Long> count_digit
         = new HashMap<Long, Long>();
        for (long i : lst) {
            if (count_digit.containsKey
            (i)) {
                count_digit.put(i,
                count_digit.get(i) + 1)
                ;
            } else {
                count_digit.put(i, 1L);
            }
        }
        for (long i : lst) {
            if (count_digit.get(i) > 2)
             {
                return false;
            }
        }
        for (int i = 1; i < lst.size();
         i++) {
            if (lst.get(i - 1) > lst.
            get(i)) {
                return false;
            }
        }
        return true;
    }
}
```

```python
from typing import List

def separate_paren_groups(paren_string:
 str) -> List[str]:
    result = []
    current_string = []
    current_depth = 0

    for c in paren_string:
        if c == '(':
            current_depth += 1
            current_string.append(c)
        elif c == ')':
            current_depth -= 1
            current_string.append(c)

            if current_depth == 0:
                result.append(''.join(
current_string))
                current_string.clear()

    return result
```

### Explanation

This function takes a string of
parentheses and returns a list of
strings, where each string is a group
of parentheses. A group is defined as a
 set of parentheses that are balanced,
meaning that each opening parenthesis
has a corresponding closing parenthesis

. The function works by iterating over
each character in the input string, and
 keeping track of the current depth of
nested parentheses. When an opening
parenthesis is encountered, the depth
is incremented and the character is
added to the current string. When a
closing parenthesis is encountered, the
 depth is decremented and the character
 is added to the current string. If the
 depth reaches 0, it means that a
complete group has been found, so the
current string is added to the result
list and cleared for the next group.
The function returns the list of groups
.

### Java version

```java
import java.util.*;
import java.lang.reflect.*;
import org.javatuples.*;
import java.security.*;
import java.math.*;
import java.io.*;
import java.util.stream.*;
class Problem {
    public static ArrayList<String>
    separateParenGroups(String
    paren_string) {
        ArrayList<String> result = new
        ArrayList<String>();
        ArrayList<Character>
        current_string = new ArrayList<
        Character>();
        int current_depth = 0;

        for (int i = 0; i <
        paren_string.length(); i++) {
            char c = paren_string.
            charAt(i);
            if (c == '(') {
                current_depth += 1;
                current_string.add(c);
            } else if (c == ')') {
                current_depth -= 1;
                current_string.add(c);

                if (current_depth == 0)
                 {
                    StringBuilder sb =
                    new StringBuilder(
                    current_string.size
                    ());
                    for (Character ch :
                     current_string) {
                        sb.append(ch);
                    }
                    result.add(sb.
                    toString());
                    current_string.
                    clear();
                }
            }
        }

        return result;
    }
```

```
}
```

```python
from typing import List

def sort_array(array: List[int]) ->
List[int]:
    return [] if len(array) == 0 else
sorted(array, reverse= (array[0]+array
[-1]) % 2 == 0)
```

### Explanation

This function takes a list of integers
'array' as input and returns a sorted
list of integers. The function first
checks if the input list is empty. If
it is, it returns an empty list.
Otherwise, it sorts the list using the
'sorted' function. The 'reverse'
parameter is set to 'True' if the sum
of the first and last elements of the
array is even (and the array will be
sorted in descending order), otherwise,
 it is set to 'False' (and the array
will be sorted in ascending order).

### Java version

```java
import java.util.*;
import java.lang.reflect.*;
import org.javatuples.*;
import java.security.*;
import java.math.*;
import java.io.*;
import java.util.stream.*;
class Problem {
    public static ArrayList<Long>
    sortArray(ArrayList<Long> array) {
        if (array.size() == 0) {
            return new ArrayList<Long
            >();
        } else {
            boolean reverse = (array.
            get(0) + array.get(array.
            size() - 1)) % 2 == 0;
            Collections.sort(array);
            if (reverse) {
                Collections.reverse(
                array);
            }
            return array;
        }
    }
}
```

### Python version

<insert Python program here>

### Explanation

This<insert Python program explanation
here>

### Java version

<insert Java completion here>

## C.3 Python-Java exp-lbl (four-shot)

You are an helpful AI assistant who understands all programming languages and can translate between them at ease. Can you explain what this Python program does line by line? The goal with the explanation, is so that a reader can easily rewrite the program in Java?

### Python version

```python
from typing import Tuple

def even_odd_palindrome(n: int) ->
Tuple[int, int]:
    def is_palindrome(n):
        return str(n) == str(n)[::-1]

    even_palindrome_count = 0
    odd_palindrome_count = 0

    for i in range(1, n+1):
        if i%2 == 1 and is_palindrome(i
):
            odd_palindrome_count += 1
        elif i%2 == 0 and is_palindrome
(i):
            even_palindrome_count += 1
    return (even_palindrome_count,
odd_palindrome_count)
```

### Explanation

The code is an implementation of finding the number of even and odd palindrome numbers between 1 and 'n' ( inclusive). It takes an integer 'n' as input and returns a tuple of two integers: even and odd palindrome number counts. A palindrome is a number that reads the same forwards and backwards.

```python
def is_palindrome(n):
    return str(n) == str(n)[::-1]
```
The function first defines a helper function 'is_palindrome' that takes a number and returns 'True' if it is a palindrome and 'False' otherwise. This is done by converting the number to a string and checking if the string is equal to its reverse.

```python
even_palindrome_count = 0
odd_palindrome_count = 0
```
The counter variable ' even_palindrome_count' and ' odd_palindrome_count' are used to record the result.

```python
for i in range(1, n+1):
```
The loops goes through all numbers between 1 and 'n' (inclusive)

```python
if i%2 == 1 and is_palindrome(i):
    odd_palindrome_count += 1
```

```python
elif i%2 == 0 and is_palindrome(i):
    even_palindrome_count += 1
```
Within the for-loop, the program checks if each number is a palindrome and even or odd. If the number is a palindrome and odd, it increments ' odd_palindrome_count'. If the number is a palindrome and even, it increments ' even_palindrome_count'.

```python
return (even_palindrome_count,
odd_palindrome_count)
```
Finally, the function returns a tuple of the two counters.

### Java version

```java
import java.util.*;
import java.lang.reflect.*;
import org.javatuples.*;
import java.security.*;
import java.math.*;
import java.io.*;
import java.util.stream.*;
class Problem {
    public static Pair<Long, Long>
    evenOddPalindrome(long n) {
        long evenPalindromeCount = 0;
        long oddPalindromeCount = 0;

        for (long i = 1; i <= n; i++) {
            if (i % 2 == 1 & &
            isPalindrome(i)) {
                oddPalindromeCount++;
            } else if (i % 2 == 0 & &
            isPalindrome(i)) {
                evenPalindromeCount++;
            }
        }
        return Pair.with(
        evenPalindromeCount,
        oddPalindromeCount);
    }

    private static boolean isPalindrome
    (long n) {
        return Long.toString(n).equals(
        new StringBuilder(Long.toString
        (n)).reverse().toString());
    }
}
```

### Python version

```python
from typing import List

def is_sorted(lst: List[int]) -> bool:
    count_digit = dict([(i, 0) for i in
 lst])
    for i in lst:
        count_digit[i]+=1
    if any(count_digit[i] > 2 for i in
lst):
        return False
    if all(lst[i-1] <= lst[i] for i in
range(1, len(lst))):
        return True
    else:
```

```python
        return False
```

### Explanation

The code is an implementation of
checking whether a list is sorted in
non-descending order and has no more
than two occurrences of any integer.
takes in a list of integers and returns
a boolean.

```
count_digit = dict([(i, 0) for i in lst
])
for i in lst:
    count_digit[i]+=1
```
The function first creates a dictionary
where the keys are the integers in the
list and the values are the number of
times that integer appears in the list.

```
if any(count_digit[i] > 2 for i in lst)
:
    return False
```
It then checks if any integer appears
more than twice in the list, and
returns False if so.

```
if all(lst[i-1] <= lst[i] for i in
range(1, len(lst))):
    return True
else:
    return False
```
Finally, it checks if the list is
sorted in non-descending order, and
returns True if so, and False otherwise
.

### Java version

```java
import java.util.*;
import java.lang.reflect.*;
import org.javatuples.*;
import java.security.*;
import java.math.*;
import java.io.*;
import java.util.stream.*;
class Problem {
    public static boolean isSorted(
    ArrayList<Long> lst) {
        HashMap<Long, Long> count_digit
         = new HashMap<Long, Long>();
        for (long i : lst) {
            if (count_digit.containsKey
            (i)) {
                count_digit.put(i,
                count_digit.get(i) + 1)
                ;
            } else {
                count_digit.put(i, 1L);
            }
        }
        for (long i : lst) {
            if (count_digit.get(i) > 2)
             {
                return false;
            }
        }
```

```java
        for (int i = 1; i < lst.size();
         i++) {
            if (lst.get(i - 1) > lst.
            get(i)) {
                return false;
            }
        }
        return true;
    }
}
```

### Python version

```python
from typing import List

def separate_paren_groups(paren_string:
 str) -> List[str]:
    result = []
    current_string = []
    current_depth = 0

    for c in paren_string:
        if c == '(':
            current_depth += 1
            current_string.append(c)
        elif c == ')':
            current_depth -= 1
            current_string.append(c)

            if current_depth == 0:
                result.append(''.join(
current_string))
                current_string.clear()

    return result
```

### Explanation

The code is an implementation of a
function `separate_paren_groups` which
takes a string of parentheses and
returns a list of strings containing
the groups of parentheses.

```
result = []
current_string = []
current_depth = 0
```
The function initializes an empty list
`result` to store the groups of
parentheses. It also initializes an
empty list `current_string` to store
the current group of parentheses being
processed. The variable `current_depth`
 is initialized to 0 to keep track of
the depth of the current group of
parentheses.

```
for c in paren_string:
```
The function iterates through each
character `c` in the input string `
paren_string`.

```
if c == '(':
    current_depth += 1
    current_string.append(c)
```
If the character is an opening
parenthesis ''('', the function

increments the `current_depth` variable and appends the character to the `current_string` list.

```
elif c == ')':
    current_depth -= 1
    current_string.append(c)
```
If the character is a closing parenthesis `')'`, the function decrements the `current_depth` variable and appends the character to the `current_string` list.

```
if current_depth == 0:
    result.append(''.join(
current_string))
    current_string.clear()
```
If the `current_depth` variable reaches 0, it means that the current group of parentheses has been fully processed, so the function joins the `current_string` list into a string and appends it to the `result` list. The `current_string` list is then cleared to start processing the next group of parentheses.

```
return result
```
The function returns the `result` list containing the groups of parentheses.

### Java version

```java
import java.util.*;
import java.lang.reflect.*;
import org.javatuples.*;
import java.security.*;
import java.math.*;
import java.io.*;
import java.util.stream.*;
class Problem {
    public static ArrayList<String>
    separateParenGroups(String
    paren_string) {
        ArrayList<String> result = new
        ArrayList<String>();
        ArrayList<Character>
        current_string = new ArrayList<
        Character>();
        int current_depth = 0;

        for (int i = 0; i <
        paren_string.length(); i++) {
            char c = paren_string.
            charAt(i);
            if (c == '(') {
                current_depth += 1;
                current_string.add(c);
            } else if (c == ')') {
                current_depth -= 1;
                current_string.add(c);

                if (current_depth == 0)
                {
                    StringBuilder sb =
                    new StringBuilder(
                    current_string.size
                    ());
```

```java
                    for (Character ch :
                     current_string) {
                        sb.append(ch);
                    }
                    result.add(sb.
                    toString());
                    current_string.
                    clear();
                }
            }
        }

        return result;
    }
}
```

### Python version

```python
from typing import List

def sort_array(array: List[int]) ->
List[int]:
    return [] if len(array) == 0 else
sorted(array, reverse= (array[0]+array
[-1]) % 2 == 0)
```

### Explanation

The code is an implementation of sorting an array of integers. It takes a list of integers `array` as input and returns a sorted list of integers.

```
return [] if len(array) == 0 else
sorted(array, reverse= (array[0]+array
[-1]) % 2 == 0)
```
The function first checks if the input list is empty. If it is, it returns an empty list. Otherwise, it sorts the list using the `sorted` function. The `reverse` parameter is set to `True` if the sum of the first and last elements of the array is even (and the array will be sorted in descending order), otherwise, it is set to `False` (and the array will be sorted in ascending order).

### Java version

```java
import java.util.*;
import java.lang.reflect.*;
import org.javatuples.*;
import java.security.*;
import java.math.*;
import java.io.*;
import java.util.stream.*;
class Problem {
    public static ArrayList<Long>
    sortArray(ArrayList<Long> array) {
        if (array.size() == 0) {
            return new ArrayList<Long
            >();
        } else {
            boolean reverse = (array.
            get(0) + array.get(array.
            size() - 1)) % 2 == 0;
            Collections.sort(array);
```

```java
                if (reverse) {
                    Collections.reverse(
                    array);
                }
                return array;
            }
        }
    }
}
```

### Python version

`<insert Python program here>`

### Explanation

The code is an implementation of`<insert Python program explanation here>`

### Java version

`<insert Java completion here>`

## C.4  Python-Java exp-lbl-d (four-shot)

You are an helpful AI assistant who understands all programming languages and can translate between them at ease. Can you explain what this Python program does line by line? If a line is too long or too complicated, simplify it and explain what individual parts of the line mean first before explaining the whole line. The goal with the explanation, is so that a reader can easily rewrite the program in Java?

### Python version

```python
from typing import Tuple

def even_odd_palindrome(n: int) ->
Tuple[int, int]:
    def is_palindrome(n):
        return str(n) == str(n)[::-1]

    even_palindrome_count = 0
    odd_palindrome_count = 0

    for i in range(1, n+1):
        if i%2 == 1 and is_palindrome(i
):
            odd_palindrome_count += 1
        elif i%2 == 0 and is_palindrome
(i):
            even_palindrome_count += 1
    return (even_palindrome_count,
odd_palindrome_count)
```

### Explanation

The code is an implementation of finding the number of even and odd palindrome numbers between 1 and 'n' (inclusive). It takes an integer 'n' as input and returns a tuple of two

integers: even and odd palindrome number counts. A palindrome is a number that reads the same forwards and backwards.

```python
def is_palindrome(n):
    return str(n) == str(n)[::-1]
```
The function defines a helper function. Let's break it down step by step:
`str(n)`
The function first converts the input from integers to strings, because Python does not allow one to easily reverse an integer by its digits.
`str(n)[::-1]`
It then uses string manipulation '[::-1]' to reverse the string.
`return str(n) == str(n)[::-1]`
It then checks if the string version of the integer is equal to its reversed string version, thereby determining whether the input integer is a palindrome or not.
Hence, the helper function' is_palindrome' takes a number and returns 'True' if it is a palindrome and 'False' otherwise.

```python
even_palindrome_count = 0
odd_palindrome_count = 0
```
The counter variable ' even_palindrome_count' and ' odd_palindrome_count' are used to record the result.

```python
for i in range(1, n+1):
```
The loops goes through all numbers between 1 and 'n' (inclusive)

```python
if i%2 == 1 and is_palindrome(i):
    odd_palindrome_count += 1
elif i%2 == 0 and is_palindrome(i):
    even_palindrome_count += 1
```
Within the for-loop, the program defines a set of if-else statements. Let's break it down step by step:
`i%2 == 1`
This checks if the number is odd by checking if the mod 2 of the number has remainder of 1
`is_palindrome(i)`
Here the program uses previously defined helper function to determine if the number is a palindrome or not.
```python
if i%2 == 1 and is_palindrome(i):
    odd_palindrome_count += 1
```
Putting it together, both of above components check if the number is odd and a palindrome, and if so, increment the counter 'odd_palindrome_count'. The same logic is applied to even palindrome numbers.
Hence, the if-else conditions checks if each number is a palindrome and even/ odd and increment respective counters.

```python
return (even_palindrome_count,
odd_palindrome_count)
```

Finally, the function returns a tuple of the two counters. Even palindrome count first, odd palindrome count next.

### Java version

```java
import java.util.*;
import java.lang.reflect.*;
import org.javatuples.*;
import java.security.*;
import java.math.*;
import java.io.*;
import java.util.stream.*;
class Problem {
    public static Pair<Long, Long>
    evenOddPalindrome(long n) {
        long evenPalindromeCount = 0;
        long oddPalindromeCount = 0;

        for (long i = 1; i <= n; i++) {
            if (i % 2 == 1 & &
            isPalindrome(i)) {
                oddPalindromeCount++;
            } else if (i % 2 == 0 & &
            isPalindrome(i)) {
                evenPalindromeCount++;
            }
        }
        return Pair.with(
        evenPalindromeCount,
        oddPalindromeCount);
    }

    private static boolean isPalindrome
    (long n) {
        return Long.toString(n).equals(
        new StringBuilder(Long.toString
        (n)).reverse().toString());
    }
}
```

### Python version

```python
from typing import List

def is_sorted(lst: List[int]) -> bool:
    count_digit = dict([(i, 0) for i in
 lst])
    for i in lst:
        count_digit[i]+=1
    if any(count_digit[i] > 2 for i in
lst):
        return False
    if all(lst[i-1] <= lst[i] for i in
range(1, len(lst))):
        return True
    else:
        return False
```

### Explanation

The code is an implementation of checking whether a list is sorted in non-descending order and has no more than two occurrences of any integer. It takes in a list of integers and returns a boolean.

```python
count_digit = dict([(i, 0) for i in lst
])
for i in lst:
    count_digit[i]+=1
```
The function first creates a dictionary where the keys are the integers in the list and the values are the number of times that integer appears in the list.

```python
if any(count_digit[i] > 2 for i in lst)
:
    return False
```
It then checks if any integer appears more than twice in the list, and returns False if so.

```python
if all(lst[i-1] <= lst[i] for i in
range(1, len(lst))):
    return True
else:
    return False
```
Let's break it down step by step:
`lst[i-1] <= lst[i]`
Within the list comprehension, this checks whether the current element at index `i` is greater or equal to the previous element at index `i-1`.
```python
if all(lst[i-1] <= lst[i] for i in
range(1, len(lst))):
```
The whole expression then checks if every element is greater or equal to the element before in the list. In another word, this checks if the list is sorted in non-descending order.
```python
    return True
else:
    return False
```
If the list is sorted in non-descending order the function returns True, otherwise it returns False.
Hence, the if-else statements returns True if the list is sorted in non-descending order, and False otherwise.

### Java version

```java
import java.util.*;
import java.lang.reflect.*;
import org.javatuples.*;
import java.security.*;
import java.math.*;
import java.io.*;
import java.util.stream.*;
class Problem {
    public static boolean isSorted(
    ArrayList<Long> lst) {
        HashMap<Long, Long> count_digit
          = new HashMap<Long, Long>();
        for (long i : lst) {
            if (count_digit.containsKey
            (i)) {
                count_digit.put(i,
                count_digit.get(i) + 1)
                ;
            } else {
                count_digit.put(i, 1L);
            }
        }
```

```java
        for (long i : lst) {
            if (count_digit.get(i) > 2)
             {
                return false;
            }
        }
        for (int i = 1; i < lst.size();
         i++) {
            if (lst.get(i - 1) > lst.
            get(i)) {
                return false;
            }
        }
        return true;
    }
}
```

### Python version

```python
from typing import List

def separate_paren_groups(paren_string:
 str) -> List[str]:
    result = []
    current_string = []
    current_depth = 0

    for c in paren_string:
        if c == '(':
            current_depth += 1
            current_string.append(c)
        elif c == ')':
            current_depth -= 1
            current_string.append(c)

            if current_depth == 0:
                result.append(''.join(
current_string))
                current_string.clear()

    return result
```

### Explanation

The code is an implementation of a
function `separate_paren_groups` which
takes a string of parentheses and
returns a list of strings containing
the groups of parentheses.

result = []
current_string = []
current_depth = 0
The function initializes an empty list
`result` to store the groups of
parentheses. It also initializes an
empty list `current_string` to store
the current group of parentheses being
processed. The variable `current_depth`
 is initialized to 0 to keep track of
the depth of the current group of
parentheses.

for c in paren_string:
The function iterates through each
character `c` in the input string `
paren_string`.

if c == '(':
    current_depth += 1
    current_string.append(c)
If the character is an opening
parenthesis ''(''', the function
increments the `current_depth` variable
 and appends the character to the `
current_string` list.

elif c == ')':
    current_depth -= 1
    current_string.append(c)
If the character is a closing
parenthesis ''')''', the function
decrements the `current_depth` variable
 and appends the character to the `
current_string` list.

if current_depth == 0:
    result.append(''.join(
current_string))
    current_string.clear()
Let's break it down step by step:
''.join(current_string)
The program joins the `current_string`
(which is represented as a list) into a
 string
Hence, the if-statement does the
following: if the `current_depth`
variable reaches 0, it means that the
current group of parentheses has been
fully processed, so the function joins
the `current_string` list into a string
 and appends it to the `result` list.
The `current_string` list is then
cleared to start processing the next
group of parentheses.

return result
The function returns the `result` list
containing the groups of parentheses.

### Java version

```java
import java.util.*;
import java.lang.reflect.*;
import org.javatuples.*;
import java.security.*;
import java.math.*;
import java.io.*;
import java.util.stream.*;
class Problem {
    public static ArrayList<String>
    separateParenGroups(String
    paren_string) {
        ArrayList<String> result = new
        ArrayList<String>();
        ArrayList<Character>
        current_string = new ArrayList<
        Character>();
        int current_depth = 0;

        for (int i = 0; i <
        paren_string.length(); i++) {
            char c = paren_string.
            charAt(i);
            if (c == '(') {
```

```java
                    current_depth += 1;
                    current_string.add(c);
                } else if (c == ')') {
                    current_depth -= 1;
                    current_string.add(c);

                    if (current_depth == 0)
                    {
                        StringBuilder sb =
                        new StringBuilder(
                        current_string.size
                        ());
                        for (Character ch :
                         current_string) {
                            sb.append(ch);
                        }
                        result.add(sb.
                        toString());
                        current_string.
                        clear();
                    }
                }
            }

        return result;
    }
}
```

### Python version

```python
from typing import List

def sort_array(array: List[int]) ->
List[int]:
    return [] if len(array) == 0 else
sorted(array, reverse= (array[0]+array
[-1]) % 2 == 0)
```

### Explanation

The code is an implementation of a
function that sorts an array of
integers. The function takes an array
of integers as input and returns a
sorted array of integers.

return [] if len(array) == 0 else
sorted(array, reverse= (array[0]+array
[-1]) % 2 == 0)
This return statement contains an
inline if-else statement. Let's break
it down step by step:
[] if len(array) == 0
This inline if statement checks if the
length of the input array is zero. If
the length of the input array is zero,
the function returns an empty array. If
 the length of the input array is not
zero, the function proceeds to else
statement.
sorted(array, reverse= (array[0]+array
[-1]) % 2 == 0)
This function sorts the list 'array'.
Let's break it down step by step:
(array[0]+array[-1]) % 2 == 0
The reverse parameter of the sorted
function is set to True if the sum of
the first and last elements of the

input array is even. If the sum of the
first and last elements of the input
array is odd, the reverse parameter of
the sorted function is set to False.
Hence, the else statement sorts the
input array. If the sum of the first
and last elements of the input array is
 even, it is sorted with reverse=True (
descending order), otherwise, it is
sorted with reverse=False (ascending
order).
Hence, the entire inline if-else
statement (and the return statement)
returns an empty array if the length of
 the input array is zero. Otherwise, if
 the sum of the first and last element
of the array is even, the array is
sorted with reverse parameter set to
True. If the sum of the first and last
element of the array is odd, the
reverse parameter is set to False.

### Java version

```java
import java.util.*;
import java.lang.reflect.*;
import org.javatuples.*;
import java.security.*;
import java.math.*;
import java.io.*;
import java.util.stream.*;
class Problem {
    public static ArrayList<Long>
    sortArray(ArrayList<Long> array) {
        if (array.size() == 0) {
            return new ArrayList<Long
            >();
        } else {
            boolean reverse = (array.
            get(0) + array.get(array.
            size() - 1)) % 2 == 0;
            Collections.sort(array);
            if (reverse) {
                Collections.reverse(
                array);
            }
            return array;
        }
    }
}
```

### Python version

<insert Python program here>

### Explanation

The code is an implementation of<insert
Python program explanation here>

### Java version

<insert Java completion here>

## D  Computational Resources

All completion queries are made to GPT-3.5 (gpt-3.5-turbo-3010) between March - June 2023. We use the Azure completion endpoint (as opposed to the OpenAI chat completion endpoint). Compute credits are provided by MIT-IBM Watson AI Lab. The average query time for a single experiment (e.g. exp, Python → Julia) takes around 10-40 minutes with one API key (of which we only used one). For Tables 1 and 2, the total query time is between 21 - 84 hours. The experiments for CodeGen2-1B are conducted on BU SCC and MIT Satori clusters with NVIDIA RTX V100, A100, and A6000. Each single experiment takes 12-24 hours so the whole compute time for Tables 1 and 2 completions is around 72-144 GPU days. Experiments for CodeGen2-16B and Llama2CodeInstruct-34B are done through IBM hosted model inference APIs. Query time is about 2-3X to that of GPT-3.5. Code execution/evaluation is done locally on the same MacBook Pro 2015.

## E  Language Specific Stop Token and Post-Processing

To stop model generations and extract the relevant code needed for the problem, we use a few language-specific tokens. In the case of OpenAI APIs, such token size limit is 4. In order to accommodate multi-function and remove irrelevant generations, we modify the stop tokens from MultiPL-E and add post processing to some languages. There are two types of post-processing we use to truncate completions:

**`truncate_after_function_ends`** requires two function implementations from each language translator: `is_end_of_function` and `is_function_signature`. The function works by greedily search the completion line-by-line. As soon as we encounter a end of a function, we start looking for the next non-empty line. We remove the rest if the next non-empty line is not a function signature. Otherwise, we keep the line and continue. Below is the Python implementation of `truncate_after_function_ends`:

```python
def truncate_after_function_ends(completion,
translator):
    lines = completion.split("\n")
    in_function = True
    for i, l in enumerate(lines):
        if in_function:
            # if encounter end of function,
start looking out for next non empty line
            if translator.is_end_of_function(
l):
                in_function = False
        else: # if we are not in function,
check if line is signature
            if len(l.strip()) != 0:
                if translator.
is_function_signature(l):
                    in_function = True
                else:
                    lines = lines[:i]
                    break
    truncated_completion = "\n".join(lines)
    return truncated_completion
```

**`truncate_after_additional_stops`** requires one additional implementation `get_function_name` and **additional_stops** property. This function is used to deal with languages that are difficult to determine the end of functions without using lexers (hard to implement `is_end_of_function`). In this case, we put additional stop tokens that indicate for certain that the line is not inside a function (usually `\n` with every single letter in the alphabet along with special symbols such as `\n#`, `\n!`.) We additionally add the main completion function's name to make sure the completion doesn't call the function itself at the base level.

### E.1  JavaScript

- original stops: `\nfunction`, `/*`, `//`, `console.log`

- modified stops: `\n}`, `/*`, `//`, `console.log`. Instead of stopping at the beginning of the next irrelevant code segment, we stop right before where the function ends, and add closing bracket back with test strings. This speeds up experiments by removing all possible additional un-expected completions.

### E.2  C++

- original stops: `\n}`.

- no modification.

### E.3  Java

- original stops: `\n    }\n`

- modified stops: `public static void main`, `###`, `\n}`. Nested functions are illegal in Java. In order to allow models to generate multiple functions, instead of stopping at the end of

a function, we stop right before the main function, or end of class.

### E.4 TypeScript

- original stops: `\nfunction, /*, //, console.log`

- modified stops: `\n}, /*, //, console.log`. Same reason as JavaScript

### E.5 PHP

- original stops: `\nfunction, \n?>, \n//, \n#`

- modified stops: `\n}, \n?>, \n//, \n#`. Same reason as JavaScript

### E.6 Ruby

- original stops: `\nclass, \ndef, \n#, \n\n`

- modified stops: `\nclass, \ndef, \n#, \nputs`. The original stop `\n\n` is problematic because it prematurely stops translation as soon as the source program has an extra empty line in the program. Additionally, `\nputs` to ensure we don't have extra generations (self-calls) at the end.

- post-processing: `truncate_after_function_ends`

### E.7 C#

- original stops: `\n    }\n`

- modified stops: `public static void Main, static void Main, \n#, \n}`. Same reason as Java. Although nested/local function is allowed in C#, this increases the variety of generations which can be accepted by unit tests.

### E.8 Go

- original stops: `\nfunc, struct, \n//`

- modified stops: `\nfunc, struct, \n// , \n}`. Same reason as JavaScript.

### E.9 Perl

- original stops: `\nsub, \n#, \n\n`.

- modified stops: `\nsub, \n#, \n}`. Same reason as Ruby.

### E.10 R

- original stops: `\n#, \n``` ``

- modified stops: `\n}`. Same reason as JavaScript.

### E.11 Rust

- original stops: `\n}`.

- no modification.

### E.12 Scala

- original stops: `\n    }\n`.

- no modification.

### E.13 Swift

- original stops: `\n}`.

- no modification.

### E.14 Bash

- original stops: `\n}`

- modified stops: `\n#, \nAnswer, \necho, \n``` ``. Although nested functions are technically allowed, it is quite conventional to write helper functions in a separate function. We also use post-processing to truncate additional unwanted generations.

- post-processing: `truncate_after_function_ends`

### E.15 Lua

- original stops: `\nlocal, \nfunction, \n--, \n\n`

- modified stops: `\n--, \n#, \nend`. The original stop `\n\n` is problematic because it prematurely stops translation as soon as the source program has an extra empty line in the program. We add base indentation level `\nend` to truncate after the function ends.

### E.16 Racket

- original stops: `\n(define , \n#|, \n;, \n(`

- modified stops: `\n(define , \n#|, \n;, \n(`

- additional stops: `\n#", \n``` ``, and `"\n"` with all letters in alphabet.

- post-processing: `truncate_after_additional_ stops`

## E.17 Julia

- original stops: \nfunction, \nmacro, \n\n,

- modified stops: \nend, \n#. Same as JavaScript: \nend is a stricter way of stopping multi-function generation since Julia allows nested functions.

## E.18 D

- original stops: \n\n, \nvoid, \nbool, \nint

- modified stops: \nvoid, \nbool, \nint, \n}. More strict stopping at the end of function, and remove problematic \n\n

## E.19 Explanation Qualitative Analysis

```
### Python version

def find_max(words: List[str]) -> str:
    return sorted(words, key = lambda x
: (-len(set(x)), x))[0]

### Explanation (exp)

This function receives a list of
strings and returns the string with the
 highest number of unique characters,
and if there are ties, the string that
comes first in lexicographic order.

### Explanation (exp-lbl)

The code is an implementation of
finding the longest word in a list of
words that has the fewest unique
characters. It takes a list of strings
'words' as input and returns the
longest string in the list that has the
fewest unique characters.

sorted(words, key = lambda x: (-len(set
(x)), x))[0]
The function first sorts the list of
words using the 'sorted' function. The
'key' parameter is set to a lambda
function that returns a tuple of two
values: the negative length of the set
of characters in the word (to sort by
fewest unique characters
first), and the word itself (to break
ties by alphabetical order).

The '[0]' at the end of the line
returns the first (i.e., longest) word
in the sorted list.

### Explanation (exp-lbl-d)

The code is an implementation of a
function that finds the word with the
maximum number of unique characters.
```

```
The function takes a list of strings as
 input and returns the string with the
maximum number of unique characters.

sorted(words, key = lambda x: (-len(set
(x)), x))[0]
This statement sorts the input list '
words'. Let's break it down step by
step:
set(x)
The 'set' function returns a set of
unique characters in the string 'x'.
len(set(x))
The 'len' function returns the number
of unique characters in the string 'x'.
-len(set(x))
The '-' sign is used to sort the list
in descending order of the number of
unique characters.
x
The 'x' parameter is used to sort the
list in ascending order of the string
itself.
Hence, the 'key' parameter of the '
sorted' function is set to a lambda
function that returns a tuple of two
values: the negative length of the set
of unique characters in the string and
the string itself. The 'sorted'
function sorts the input list based on
this tuple. The '[0]' at the end of the
 statement returns the first element of
 the sorted list, which is the string
with the maximum number of unique
characters.
```

Above is an example explanation of a program in Humaneval (`humaneval_158_find_max`). With **exp**, the explanation is very high-level and does not go into details with the Python implementation. With **exp-lbl**, each line of the program is referenced along with an explanation. However, we can see that in the second paragraph **exp-lbl** explains that `words` sorted "by fewest unique characters first" when it should be the other way. This led to the wrong translation where the negative sign is ignored. On the other hand, with **exp-lbl-d**, complicated line like ones above is decomposed first, explained separately and then combined together. This particular explanation emphasizes the negative sign and improves translation pass rate.

## F How did we select with few-shot programs and write explanations

To be consistent across trials, and due to the fact that not every program has gold translations, we use fixed few-shot in our main experiments. In this case, the few-shot examples we pick is crucial to the performance: they need to be representative of

the dataset characteristics and contain features that may demonstrate the usefulness of explanations. Once we select the few-shot programs, we simply use the model to generate zero-shot explanations, and modify for correctness and structural preferences (e.g. code lines followed by explanations with **exp-lbl**).

Our "development" translation direction is Python → Java. We quickly notice that GPT-3.5 is bad at translating nested functions, which occur many times in the canonical solutions in HumanEval. Compounded with the fact that the original MultiPL-E stop tokens do not allow models to generate multiple functions and that Java does not allow nested functions, the only way for GPT-3.5 to generate a correct translation of these program is to use lambda expressions, which can be extremely convoluted if the nested function is long. Therefore, after loosening the constraint of single function only (see Java section in Apx E), we decide to use the first few-shot example `humaneval_107_even_odd_palindrome` as a demonstration to show the model how to translate these type of functions (by adding private helper function after the main function).

Second and third few-shot examples were selected quite arbitrarily. The only criteria we had in-mind was that these programs need to be somewhat difficult, on the longer side length-wise, and that the semantics of the program is not immediately clear after looking at the function name or first sentence explanation of the function. Lastly, the programs also should not be close to each other in the problem sequence (just in case when designing the dataset, (Chen et al., 2021) decided to stress-test different aspect of code-generation in batches). Hence, we picked `humaneval_126_is_sorted` and `humaneval_1_separate_paren_groups`.

Last example is often the most influential due to the proximity to test examples (Lu et al., 2022). We decide on this program after careful error analysis of **exp** and **exp-lbl**. With few-shot **exp-lbl**, we found the explanations to be of great qualities already: the explanation chunks the model into several semantically independent segments and explain each of them separately. We notice that the quality of the explanations (through manual inspection) are worse when the segments are longer or more complicated. In many cases, these coincides with programs that contain one extremely

semantically complicated line. This is because the model has to spend paragraphs explaining these lines, and often produces confusing/wrong statements. Many of the remaining 17 assertion errors (semantic errors) in four-shot **exp-lbl** are due to in-correct/insufficient explanation). Section E.19 shows an example program.

These errors, however, can be effectively mitigated if they were further decomposed into smaller chunks of code statements. Due to the inherent tree-like structure (AST) of program, asking models to decompose is another way of learning fine-grain parsing of the code. Surprisingly, models like GPT-3.5 is able to parsing very well. Hence, we developed our third explanation method **exp-lbl-d**: we ask the model to explain line-by-line, and breakdown the line into smaller parts if the line is too long or complicated. To demonstrate the usefulness of such method, we pick a program that is short but contains a long and complicated line (`humaneval_88_sort_array`). See E.19 for the result of explanations generated with few-shots across 3 explanation methods. With **exp-lbl-d** four-shot, we find the explanations generated for almost all 17 previously failing programs (in **exp-lbl**) to be correct, and the remaining 12 assertion errors are all results of wrong program generation given correct explanations.

# G Python-to-X Detailed Result Investigation

All experiments detailed here are conducted with GPT-3.5 unless otherwise specified.

## G.1 Removing target language specific information in zero-shot explanation

```
### Python version

def strlen(string: str) -> int:
    return len(string)

### Explanation

This function takes a string as input
and returns the length of the string.
It does this by using the built-in 'len
' function in Python. The function is
defined with the name 'strlen' and
takes a single argument, which is a
string. The return type of the function
 is an integer.

To rewrite this program in Java, you
would need to define a function with
the same name and argument type, and
```

|  | Java | Lua |
| :---: | :---: | :---: |
| exp | 0.829 | 0.593 |
| exp (remove) | 0.829 | 0.592 |
| exp-lbl | 0.825 | 0.615 |
| exp-lbl (remove) | 0.825 | 0.615 |
| exp-lbl-d | 0.846 | 0.595 |
| exp-lbl-d (remove) | 0.84 | 0.595 |

Table 7: Comparing results for different explanation types before and after removing target-specific information in zero-shot setting

```
use the `length()` method of the
`String` class to get the length of the
string. The return type would also need
to be changed to `int`. Here is an
example of what the Java code might
look like:

```java
public static int strlen(String string)
{
  return string.length();
}
```
```

Above is an example of what target specific information in generated explanations looks like.

Since different types of explanations generate different types of kind of information, we remove all of them in Python-to-Java direction to disentangle the effects of target-specific information vs the level of detail in the explanation. The removal is done through a script, where we remove from the first sentence that mentions anything about the target language.

Here, we compare the pass@1 before and after removing such information for all explanation types. As seen, there is only a slight decrease in the performance. When translating into low-resource languages, the occurrence of target-specific information is much less frequent, leading to much less difference between regular explanations vs. explanations with target information removed.

In addition, we compare using target-independent explanations (**exp (java)**) vs. target-dependent explanations (**exp (tgt-specific)**) in Table 8. Target specific explanations do not impact performance significantly in zero-shot ($p = 0.090$). Target-specific explanations tend to decrease performance in lower-resource languages.

## G.2 Four-shot explanation variations

To observe the difference between the Python-to-Java four-shot explanation, four-shot target language-specific explanation, and re-using zero-shot target-language specific explanation in four-shot translation, we compare their Python-to-X translation performance in 18 target languages (Table 9). Just by observing best trials across language directions in Table 9, **exp (java)** wins in 6/18 directions, whereas **exp (tgt-specific)** and **exp (zero-shot)** each win 5/18 and 11/18 directions. 2-tail paired t-test indicates that both alternative trials are significantly different from re-using 4 shot examples from Java ($p_{tgt\_specific} = 0.036$) and $p_{zero-shot} = 0.024$)). **Four-shot explanations are worse than zero-shot generated explanation**. This is intuitive because the explanations with **exp** method in the zero-shot setting are good enough. By incorporating mostly its own explanation in a few-shot setting, the model is not obtaining more information, but restricting its potential to generate diverse types of explanation.

## G.3 heuristically selected explanations

For more fair comparisons, we include here the heuristically selected explanations with respective baselines. Since **exp** selection was done over zero-shot explanations, we compare **exp (logprob)** against **exp (zero-shot)**. There is still a mean improvement of 0.73% with standard deviation of 1.8%.

## G.4 Error types breakdown in Python-to-X

For each program, in addition to determine unit tests pass rate, we also use string matching on stderr and stdout to categorize the error type. In order to generalize across different languages, we group the errors into the following 6 types:

- **Type Error** includes all errors related to interactions between variables with the wrong types. For example, in Python, floats cannot be used to index list, and a string cannot be used to multiply with another string.

- **Undeclared Error** includes all errors calling methods or variables that do not exists. It ranges from undeclared variable, to unable to find equivalent built-in function such as `string.swapcase()` in Python

- **Assertion Error** catches all cases where the function output does not match the expected

| | High | | | | Medium | | | | Low | | | | | Extremely-Low | | | | |
|---|---|---|---|---|---|---|---|---|---|---|---|---|---|---|---|---|---|---|
| trial | js | cpp | jv | ts | php | rb | cs | go | pl | r | rs | sc | sw | sh | lua | rkt | jl | d |
| exp (java) | 84.8 | 82.1 | 82.9 | 85.1 | 77.0 | 78.3 | 83.6 | 45.7 | 68.2 | 46.9 | 74.8 | 74.7 | 70.5 | 55.4 | 60.0 | 41.3 | 70.5 | 44.4 |
| exp (tgt-specific) | 85.7 | 80.7 | 82.9 | 83.7 | 76.4 | 78.4 | 85.2 | 45.0 | 66.9 | 47.5 | 75.0 | 74.6 | 70.3 | 53.2 | 59.3 | 39.5 | 71.2 | 42.7 |

Table 8: Comparing zero-shot target-independent explanation vs target-specific explanations across 18 languages. 2-tail paired t test $p = 0.090$

| | High | | | | Medium | | | | Low | | | | | Extremely-Low | | | | |
|---|---|---|---|---|---|---|---|---|---|---|---|---|---|---|---|---|---|---|
| trial | js | cpp | jv | ts | php | rb | cs | go | pl | r | rs | sc | sw | sh | lua | rkt | jl | d |
| exp (java) | 87.3 | 77.8 | 82.1 | 88.4 | 76.4 | 80.4 | 83.3 | 51.4 | 73.2 | 55.6 | 73.0 | 73.7 | 67.1 | 73.1 | 67.9 | 44.7 | 69.4 | 45.9 |
| exp (tgt-specific) | 85.1 | 79.0 | 82.1 | 87.4 | 78.2 | 80.4 | 82.9 | 52.7 | 73.1 | 59.1 | 76.4 | 75.7 | 68.7 | 72.9 | 68.6 | 45.2 | 69.2 | 48.7 |
| exp (zero-shot) | 86.1 | 80.5 | 81.6 | 85.7 | 78.4 | 80.4 | 83.7 | 52.7 | 73.1 | 57.5 | 76.8 | 75.8 | 70.0 | 73.8 | 68.8 | 47.0 | 68.3 | 48.5 |

Table 9: Comparing different four-shot translation with different explanations: Python-Java explanation, target-specific 4 explanation and target-specific 0 shot explanation across 18 languages. 2-tail paired t-test between first and second row is $p = 0.036$ and between first and third row is $p = 0.024$

output. This indicates that the program runs, but is not functionally the same as the source.

- **Runtime Error** generally includes all errors that do not occur for every unit tests. For instance, index out of bound error may only occur with input of long lists.

- **Other Syntax Errors** includes all other type of errors not captured by a specific groups from above.

- **Unhelpful** includes cases where the generated program contain exclusively comments like TODO, Your Code Here.

For better generalization, we also combine **assertion error** and **unhelpful** into **semantic error** and **Type, Undeclared** and **other syntax errors** into **syntactic error**. Here are some of our main conclusions:

**Less syntax error across the board in zero-shot** In general we see a decrease of syntactic error across all target language resource level (Fig 5). Specifically, there is a significant decrease in **unhelpful** generations in trials with explanations. This is similar to effects of having few shot examples Min et al. (2022b), except in this case we do not actually provide the actual format of target translation. Other than reducing **unhelpful** generations, self-generated explanations also decrease **undeclared** and **type error** (more so in higher resource directions). This is intuitive because as model reasons through program explanations, it may generated sequences that specify variable type or specific methods used within source program, which in term provides more information for the

translation step to generate appropriate method to call. Surprisingly, there is no sign of decrease in **semantic error**. This is likely due to the fact that by resolving syntactic errors, those programs switched to having semantic errors. In Fig 6 we look-into this phenomenon specifically.

**No significant difference in four-shot** Errors seem to be distributed very similarly across all trials. There are two exceptions. First, in high-resource target languages, **other syntax errors** seem to drop significantly in **exp-lbl-d** than the other explanations, which both contain more error than **direct** baseline. In extremely low-resource target language, there also seem to be a somewhat significant drop in **other syntax errors**.

### G.5 Error conversion between direct translation and with explanations

To observe the program status with and without using explanations, we track each problem's status in **direct** and explanation trials. In Fig 6, we plot **direct** status on the x-axis and corresponding status with explanation on the y-axis. Here are some key take-aways:

**More detailed explanations decrease semantic error rate** In the top two rows of Fig 6, we can observe the differences between three explanation methods. In zero-shot setting (row 1), we can see that **exp-lbl** converts more **semantic errors** in **direct** to **pass**, and slightly more **syntactic errors** to **semantic error**, which are both indication of improvements. In four-shot setting (row 2), with more detailed explanations, we see consistent decrease in **pass**→**semantic error** (explanation misleading translation), **semantic error->pass** but an increase

| | High | | | | Medium | | | | Low | | | | | Extremely-Low | | | | |
|---|---|---|---|---|---|---|---|---|---|---|---|---|---|---|---|---|---|---|
| Trial | js | cpp | jv | ts | php | rb | cs | go | pl | r | rs | sc | sw | sh | lua | rkt | jl | d |
| exp (zero-shot) | 86.1 | 80.5 | 81.6 | 85.7 | 78.4 | 80.4 | 83.7 | 52.7 | 73.1 | 57.5 | 76.8 | 75.8 | 70.0 | 73.8 | 68.8 | 47.0 | 68.3 | 48.5 |
| exp* | 88.9 | 80.8 | 85 | 88.3 | 78.6 | 81.4 | 85.1 | 52.2 | 71.6 | 57.3 | 74.5 | 76.1 | 71.3 | 73.1 | 67.9 | 46.2 | 72.9 | 50.7 |
| exp-lbl | 87.6 | 82.4 | 85.3 | 87.1 | 81.5 | 80.1 | 84.7 | 50.8 | 72 | 56.9 | 74.9 | 73.9 | 70.1 | 71.9 | 68.6 | 45.6 | 72.1 | 46.2 |
| exp-lbl-d | 87.8 | 83.9 | 86.5 | 88.3 | 82 | 80.7 | 84.3 | 50.6 | 70.9 | 57 | 76.3 | 75.5 | 70.9 | 72.5 | 69.4 | 45.3 | 71.1 | 45.1 |
| exp-lbl-d* | 87.9 | 84.6 | 87.8 | 88.5 | 81.5 | 80.6 | 85.3 | 51.6 | 70.1 | 57 | 78.7 | 77.7 | 71.9 | 72.3 | 69.4 | 46.3 | 71.6 | 44 |

Table 10: Four-shot translation with explanations vs. respective heuristically selected explanations. **exp\*** represent heuristically selected **exp** using **logprob**. **exp-lbl-d\*** represent heuristically selected **exp-lbl-d** using **frag**

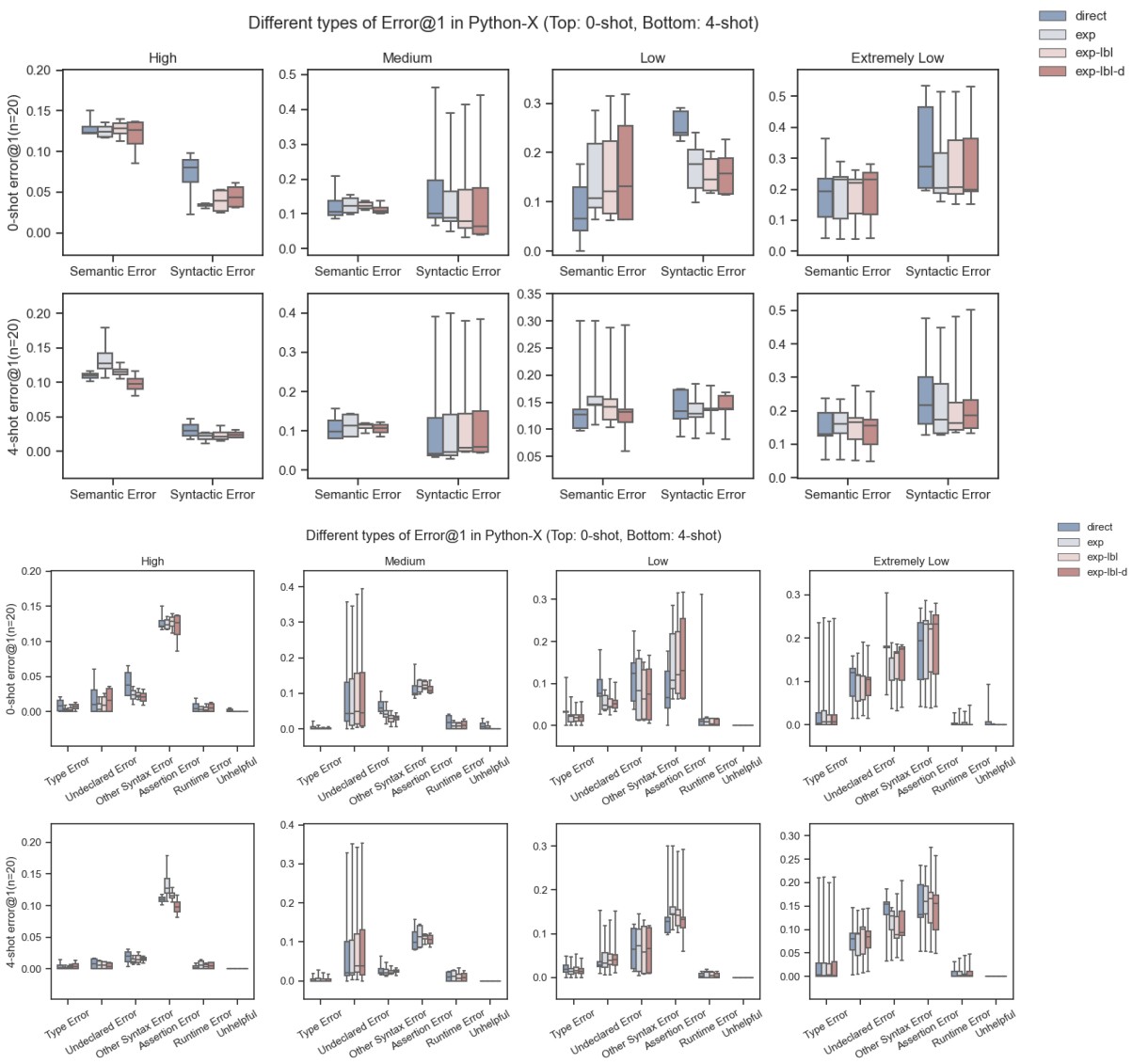

Figure 5: Python-to-X translation Error@1 rate (of different types) across different target language resources. The top plot displays semantic vs. syntactic error while the bottom displays detailed error types. We use string simple string matching to determine error type and use the same formal as Pass@1 to calculate Error@1

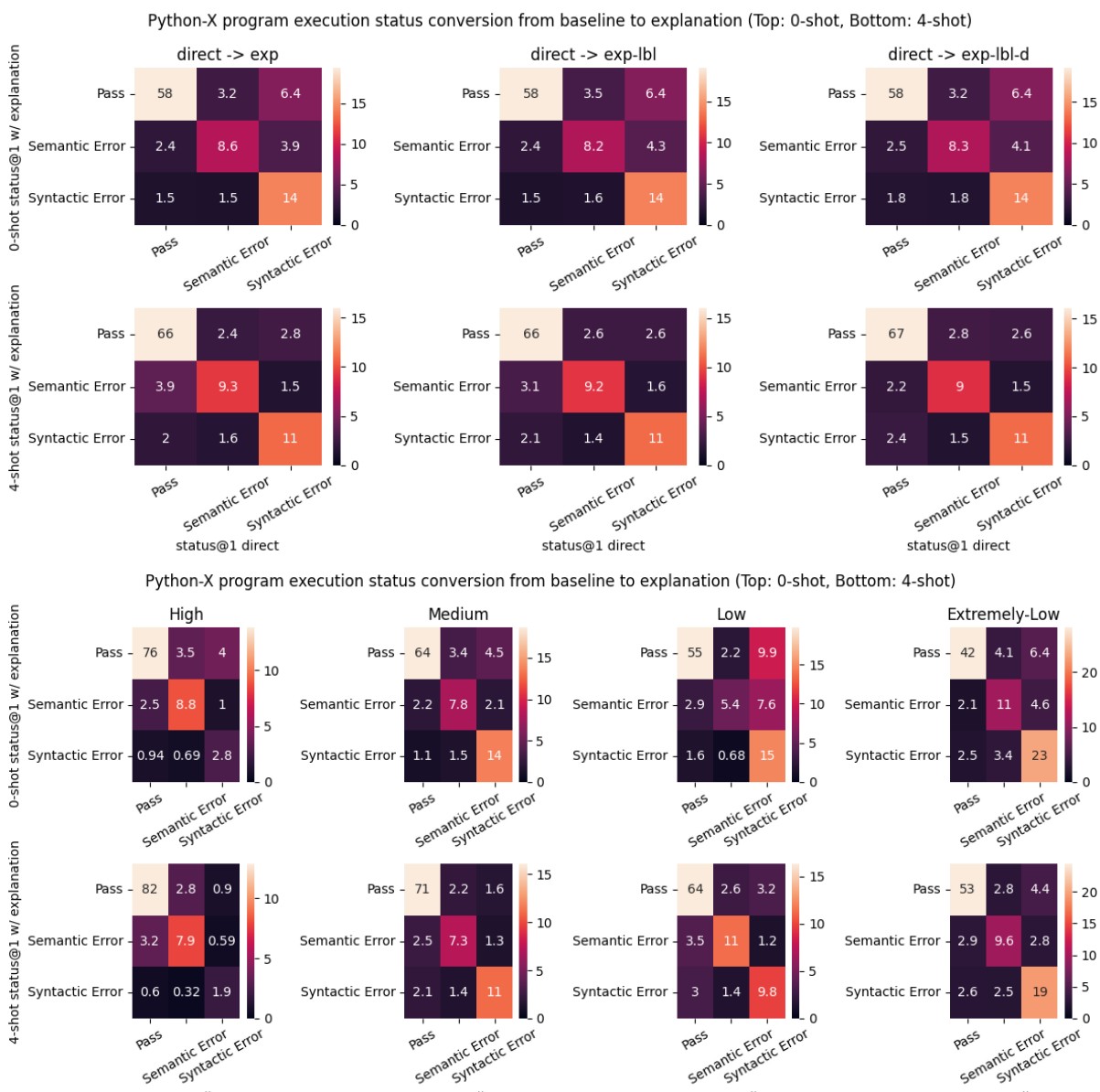

Figure 6: Python-to-X translation status conversion between baseline and explanation. X-axis indicate baseline status and y-axis indicate translation status with explanations. In the top figure, results are aggregated across target languages. In the bottom figure, results are aggregated across **exp**, **exp-lbl**, and **exp-lbl-d**

of **pass→syntactic error**. These all indicate that a more detailed explanation indeed decreases the amount of semantic errors

**Higher target language resource, proportionally more improvement with explanation, less misleads** In the bottom two rows of Fig 6, we can see the effectiveness of explanation across target languages of different resource level. In zero-shot (row 3), we see a majority of the improved cases (**not pass->pass**) come from improving syntactic error. However, if we count the improvements of **syntactic error->semantic error**, the effect becomes similar. Proportionally, high-resource benefits the most in improving syntactic errors. In lower resources, there's proportionally more chance of explanation misleading the translation (**pass->no pass**). In four-shot (row 4), the effect of explanation is much smaller (**pass->no pass** or **no pass->pass**)

### G.6  Translation pass rate with different program lengths

Typically, generating longer programs is harder. We look into the success rate of each our trials with respect to the source program length to observe if there are any patterns. We find that **explanation affects translation across length uniformly, with better performance in high-resource long programs**. In the top left box plot of Fig 7, we can see a more significant improvement for longest set of programs with explanation. This effect dampens slightly as we translate to lower-resource languages.

### G.7  Python-to-X translation Pass@10 (n=20)

In the main result table 1, we presented Pass@1 results with GPT-3.5. For convenience and cost, we also report @10 results from the same trial (Table 11), but note that for more accurate and optimal estimation @10 should be estimated with $n = 200$ and $t = 0.8$ (Cassano et al., 2022).

**Less relative improvements than pass@1**  Overall, from Table 11, we can see less improvements in pass@10. This is reasonable because ultimately adding explanation restricts the generation space and lowers the diversity of the output generations. Still, we see consistent improvements with explanations.

**Zero-shot exp provide best coverage**  In the top 4 rows of Table 11, we can see **exp** outperforms the rest in the most directions (9/19). This is probably because there are countless ways of explaining

a program in free-form natural language, and abstract explanation provide the least constraints on generating a diverse set of programs (better recall)

**Four-shot better/detailed explanation leads better coverage**  In the bottom 4 row of Table 11, we can see that either **direct** translation, **exp-lbl-d**, or heuristically selected explanation wins. Indicating that with a good quality explanation, we can still obtain improvements in few-shot setting.

### G.8  Python-to-X translation performance vs. NL-to-Code performance

To investigate whether NL-to-code performance correlates to python-to-X translation performance, we compare our zero-shot results with Cassano et al. (2022) with code-davinci-002. In Fig 8, we can see a strong correlation between the two. On top of direct translation, explanations (best explanation for each target language) improve translations (absolute difference) uniformly across source languages, and a higher relative improvements in languages which are hard for NL-to-code task (lower-resource languages)

### G.9  Python-to-X for Opensource models

## H  X-to-X Opensource Model Results

All experiments detailed here are conducted with GPT-3.5 unless otherwise specified.

**There are still improvements with self-generated explanations across most directions In weaker opensource models**  CodeGen2-1B improves more consistently (than 16B) using self-generated explanations (baseline is the worst in all X-to-X directions, and in 17/18 Python-to-X directions), as much as 300%+ improvement (12, Lua→Python, Python→JavaScript). In Python→Ruby, model with explanations obtains a pass rate of 5.9, while baseline does not generate any single correct translation (pass rate of 0). CodeGen2-16B shows weaker results, with baseline outperforming in 10/18 directions in Table 1 and 2. Perhaps it has a weaker alignment between natural language and programming language, resulting in worse explanations generated for each problem. The majority of errors from translation with explanation are syntactic. For Llama2CodeInstruct-34B, there are consistent absolute improvements of 5%-10% and maximum relative improvements of up to  40% (Java → C++ in table 2).

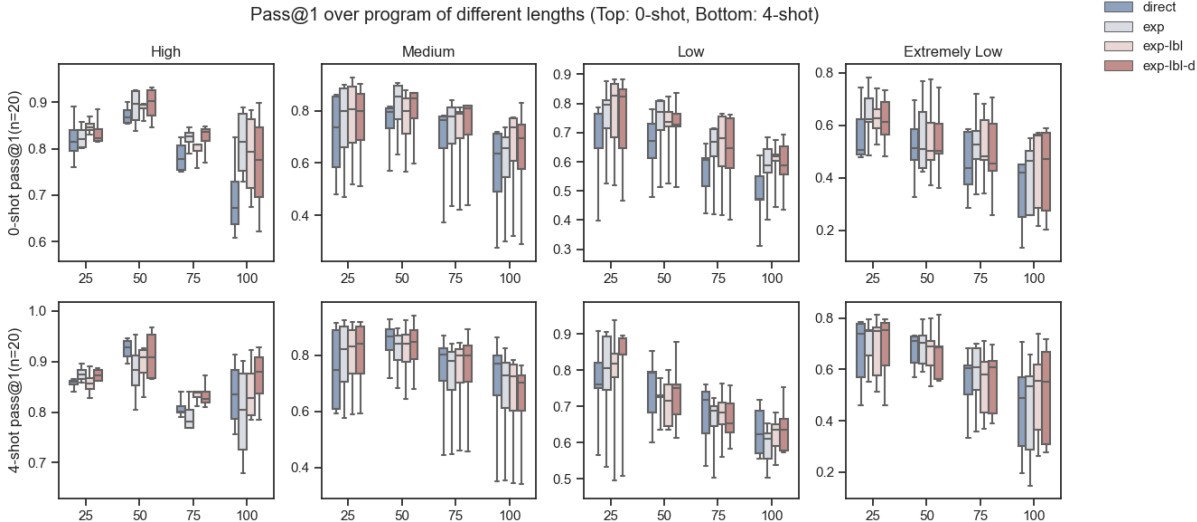

Figure 7: Python-to-X translation pass@1 rate across programs binned in quartiles. On x-axis, 25=shortest, 100=longest programs

| Res | High | | | | | Medium | | | | Low | | | | Extremely-Low | | | | |
|---|---|---|---|---|---|---|---|---|---|---|---|---|---|---|---|---|---|---|
| Trial | js | cpp | jv | ts | php | rb | cs | go | pl | r | rs | sc | sw | sh | lua | rkt | jl | d |
| direct(0) | 90.3 | 83.5 | 83.1 | 84.4 | 75.8 | 83.7 | 85.4 | 50.6 | 67.2 | 51.9 | 82.2 | 75.3 | 75.2 | 70.3 | 66.8 | 45.3 | 69.6 | 49.1 |
| exp(0) | 89.1 | 87.3 | 88.6 | 88.7 | 80 | 81.8 | 87.5 | 52.5 | 74.4 | 57.4 | 81.3 | 80.8 | 78.3 | 71.9 | 70.2 | 58.3 | 75.8 | 51.5 |
| exp-lbl(0) | 88.6 | 87.8 | 87.8 | 88.3 | 79.4 | 83.8 | 87.4 | 50.6 | 73.4 | 59.2 | 83.4 | 80.6 | 80.6 | 72.6 | 71.6 | 52.2 | 75.3 | 51.7 |
| exp-lbl-d(0) | 87.7 | 83.7 | 87.9 | 88.3 | 78.7 | 85.4 | 85.9 | 49.1 | 71.4 | 55.8 | 82.7 | 78.9 | 81.7 | 67.9 | 68 | 49.3 | 73.8 | 50.4 |
| direct (4) | **92** | **88.4** | 88.6 | 90.9 | 84 | 86 | **88.1** | **56.9** | 78.9 | **65.1** | 84.7 | 81.2 | 79.6 | 83.8 | 73.7 | 59.7 | 71.5 | 52.5 |
| exp (4) | 90.3 | 81.2 | 86.4 | 88.9 | 78.7 | 85.1 | 85.4 | 56.5 | 79.8 | 61.1 | 82.4 | 80.5 | 78.4 | 82.3 | 75.1 | 57.6 | 74.5 | 52.9 |
| exp-lbl (4) | 89 | 85.3 | 88.1 | 90.6 | 84.6 | 85.7 | 86 | 57 | 79.9 | 64.6 | 81.7 | 79.9 | 77.9 | 82.9 | 73.5 | 61.4 | **77.9** | **55** |
| exp-lbl-d (4) | 89.7 | 88.1 | 89.6 | **92.2** | **85.2** | 86.3 | 87 | 55.7 | **80.3** | 64 | 83.9 | 82.1 | 79.9 | 84 | 74 | 61.6 | 77 | 51.7 |
| exp (4)* | 89.8 | 84.8 | 87.5 | 90.1 | 83.2 | 86.6 | 88.1 | 55.8 | 77.4 | 64.8 | 81.4 | 82.6 | 80.1 | 81 | 73.8 | 62.5 | 77.1 | 54.9 |
| exp-lbl-d (4)* | 89.8 | 87.6 | **90.9** | 92.1 | 84.5 | 84.9 | 88 | 56.1 | 79.9 | 63.4 | 84.5 | 82.3 | 78.4 | **84.1** | **75.3** | **64.2** | 77.3 | 50.7 |

Table 11: Translation pass@10 from Python to X. * indicate heuristically selected explanations (Section 3.4). Parenthesis in **trial** indicates # of shots. Res=target language resource level. Best within same shot setting (no heuristics) is underscored and overall best is in **bold**

| Res | High | | | | | Medium | | | | Low | | | | Extremely-Low | | | | |
|---|---|---|---|---|---|---|---|---|---|---|---|---|---|---|---|---|---|---|
| Trial | js | cpp | jv | ts | php | rb | cs | go | pl | r | rs | sc | sw | sh | lua | rkt | jl | d |
| | | | | | | | CodeGen2-1B | | | | | | | | | | | |
| direct | 4.6 | 9.4 | 5.3 | 7 | 13.4 | 0 | 6.4 | 3.5 | 3.6 | 0.6 | 4.7 | 2.6 | 3.5 | 5.7 | 2.7 | 0 | 2.4 | 6.2 |
| exp | 15.4 | 11.7 | 7.8 | 13.5 | 14.2 | 5.9 | 8.3 | 8.3 | 3.9 | 0.6 | 8.9 | 5.8 | 6.9 | 4.8 | 4.9 | 0 | 2.9 | 7.9 |
| exp-lbl | 13.9 | 10.7 | 6.4 | 9.6 | 14.7 | 4.9 | 7.5 | 7.3 | 3.9 | 0.9 | 8.5 | 4 | 7.2 | 4.6 | 4.1 | 0.1 | 3.3 | 7.3 |
| exp-lbl-d | 12.7 | 11.7 | 7.6 | 10.5 | 14.5 | 3.8 | 8 | 7.5 | 4.5 | 0.4 | 7.9 | 4.7 | 7.4 | 4.6 | 5.2 | 0 | 3 | 5.9 |
| | | | | | | CodeGen2-1B with explanations from ChatGPT | | | | | | | | | | | | |
| exp | 19.3 | 15.5 | 6.1 | 10.8 | 16.5 | 3.9 | 11.2 | 9.8 | 4.2 | 0.4 | 10.3 | 7 | 10.4 | 5.2 | 4.8 | 0.1 | 3.7 | 7.2 |
| exp-lbl | 17.2 | 14 | 6.1 | 11.7 | 16.1 | 4.8 | 11.6 | 10.7 | 3.6 | 0.6 | 9.5 | 6.5 | 9.8 | 5.6 | 4.9 | 0.1 | 4.5 | 8.5 |
| exp-lbl-d | 18.2 | 14.8 | 0.4 | 13.2 | 16.5 | 4.9 | 12 | 12.1 | 3.7 | 0.6 | 9.6 | 7.2 | 7.9 | 4.8 | 4.8 | 0.3 | 3.7 | 7.8 |
| | | | | | | | CodeGen2-16B | | | | | | | | | | | |
| direct | **48.4** | **41.2** | 47.7 | **44.8** | 38.4 | 8.6 | 46 | **32.5** | 18.4 | 10.8 | 29.6 | 29.6 | 20.5 | 12.6 | 20.2 | 1.1 | 22 | 11.7 |
| exp | 45.1 | 38.2 | 40.4 | 43.7 | 37.5 | 2.8 | 41.8 | 30.5 | 20.1 | **13.3** | 32.7 | 29.1 | 23.8 | 10.5 | 23.4 | **1.6** | 23.8 | 14.4 |
| exp-lbl | 45.1 | 39.7 | 46.7 | 43 | **41** | 11 | 45.3 | 29.3 | 19.2 | 12.6 | **32.9** | **31.6** | 25.4 | **12.8** | 22.8 | 1.3 | **24.9** | **15.8** |
| exp-lbl-d | 43.2 | 38.4 | **50** | 39.9 | 37.6 | **13** | **47.1** | 30.1 | **21** | 11.7 | 32.6 | 30.6 | **27.3** | 11.7 | 24.3 | 1.3 | 24.8 | 15 |
| | | | | | | | Llama2CodeInstruct-34B | | | | | | | | | | | |
| direct | 64 | 55.7 | 61 | 55.9 | 52.1 | 55.5 | **67.7** | **34.4** | 37.7 | 23.2 | 42.2 | 45.6 | 30.8 | 29.5 | 28.6 | 23.7 | 47.5 | 26.5 |
| exp | **69.9** | **58** | 59.1 | **67.7** | 58.5 | 67.5 | 63.6 | 33.2 | **47.1** | 26.3 | 45.8 | **48.9** | **34.8** | 29.3 | **33.4** | **28** | **59.4** | 24.8 |
| exp-lbl | 69.3 | 56 | 62.7 | 64.7 | 56.5 | 67.7 | 64.6 | 31.6 | 45 | 25.4 | **47** | 42.9 | 34.1 | **30.5** | 31.3 | 25.1 | 58.9 | 25.4 |
| exp-lbl-d | 69.4 | 57.1 | **63.3** | 64.2 | **59.1** | **69.3** | 64.8 | 30.4 | 44.7 | **26.8** | 42.6 | 46.1 | 34.3 | 28.2 | 33.3 | 23.7 | 55.1 | **28.2** |

Table 12: Translation pass@1 from Python to X (0-shots) with open source models. Best within same the same model is in **bold**. Between **CodeGen2-1B** and **CodeGen2-1B with explanation from ChatGPT**, best amongst using same model's explanation is underlined.

| | | | CodeGen2-1B | | | | CodeGen2-16B | | | | Llama2CodeInstruct-34B | | | |
|---|---|---|---|---|---|---|---|---|---|---|---|---|---|---|
| Resource | Type | src-tgt | direct | exp | exp-lbl | exp-lbl-d | direct | exp | exp-lbl | exp-lbl-d | direct | exp | exp-lbl | exp-lbl-d |
| High-to-High | D-D | py - js | 4.6 | **14.9** | 13.9 | 13.1 | **48.4** | 45.1 | 45.1 | 43.1 | 64 | **69.9** | 69.3 | 69.4 |
| | D-S | js - jv | 7.8 | **11.2** | 9.6 | 10.1 | **47.3** | 44.3 | 42 | 40 | 67.1 | 69.1 | **70.4** | 66 |
| | S-D | cpp - py | 8.3 | **13.9** | 13.5 | 12.5 | 38 | **57.2** | 55.1 | | 84.4 | **86.1** | 84.4 | 85.4 |
| | S-S | jv - cpp | 14.2 | 15.6 | 14.8 | **15.9** | **52.7** | 49.2 | 47.7 | 46.9 | 43.3 | 60 | **60.3** | 59.2 |
| High-to-ExtLow | D-D | js - rkt | 0 | 0 | 0 | 0 | 2 | **2.1** | 2 | 1.7 | 21.3 | **26.6** | 23.7 | 19 |
| | D-S | py - d | 6.2 | **7.9** | 7.3 | 5.9 | 11.7 | 14.4 | **15.8** | 15 | 26.5 | 24.8 | 25.4 | **28.2** |
| | S-D | cpp - lua | 8.9 | **11.8** | 10 | 9.8 | 33.6 | **35.2** | 33.2 | 33.5 | 34.8 | **43** | 38.9 | 36.7 |
| | S-S | jv - jl | 3.2 | **4.9** | 3.9 | 4.4 | 26.4 | **27.9** | 25.8 | 25.9 | 51.5 | **62.2** | 59 | 58.1 |
| ExtLow-to-High | D-D | lua - py | 1.9 | **6** | 5.9 | 5.6 | 37.6 | 45.6 | **47.8** | 46.7 | 55.1 | **70.9** | 59.9 | 61.5 |
| | D-S | rkt - jv | 7.9 | 7.6 | **8.1** | 7.9 | 28.4 | 27.8 | 24.7 | 24.8 | **61.2** | 59.4 | 58.2 | 57.3 |
| | S-D | jl - js | 3.7 | **6.9** | 3.9 | 5.1 | 40.1 | 39.7 | 40 | **42.9** | 55.1 | 67.7 | 66.9 | **68.8** |
| | S-S | d - cpp | 11.7 | 12.7 | 12.6 | **14.1** | **53.1** | 47.6 | 47.1 | 50.4 | 55 | **65.3** | 59.8 | 62 |
| ExtLow-to-ExtLow | D-D | lua - rkt | 0 | 0 | 0 | 0 | 0.8 | 0.9 | **1** | 0.9 | 18.7 | **23.5** | 19.4 | 18 |
| | D-S | rkt - jl | 2 | **2.1** | 1.5 | 0.8 | 13.1 | **14.6** | **14.6** | 14.4 | 52.4 | **58.3** | 50.2 | 47.8 |
| | S-D | d - lua | 7.7 | **8.2** | 6.3 | 7.2 | 29.3 | 28.1 | 28.8 | 28.9 | 27.6 | **35.3** | 31.1 | 32 |
| | S-S | jl - d | 0 | **0.007** | 0 | 0 | 10.1 | 7.8 | 5.4 | 5.3 | 24.9 | 26.5 | 28.2 | **28.3** |

Table 13: 0-Shot translation pass@1 between 16 different pairs of languages for open source models. **Resource** indicates the language resource levels of the source and target. **Type** indicates the source and target language typing characteristics (D/S=dynamically/statically typed). The best runs within the same model are in **bold**.

**Better explanation leads to better translation** even in smaller model cases. We compare CodeGen2-1B performance given self-generated explanation vs. GPT-3.5 generated explanation and see that the better explanation outperforms self-generated explanation in 12/18 Python-to-X directions, with a maximum improvement of 400%+ in Python→ JavaScript.

**In smaller/weaker models, detailed explanations (exp-lbl or exp-lbl-d) do not improve as much as exp does** Often this is due to lower quality explanations generated when the model is asked to do something it is not capable of. The line-by-line explanations often lead to repetitive content when source programs contain several repetitive lines (library import in C++ → Python direction, with CodeGen2-1B)

**Explanation's effectiveness at improving downstream translation is transferable between models** Explanations that lead to higher pass rates in GPT-3.5 also tend to lead to higher success rates in CodeGen2-1b. If we compare Table 12 **exp**, **exp-lbl**, **exp-lbl-d** trials with CodeGen2-1b with GPT-3.5 against the same three rows in Table 1, we see the best explanation type in each translation direction (e.g. Python →TypeScript) are typically the same between two models. This is an indication of the "robustness" of the explanations.

## I Explanation Improvement correlation with problem difficulty

To understand whether self-generated explanation improve more difficult problems (or vice versa), we

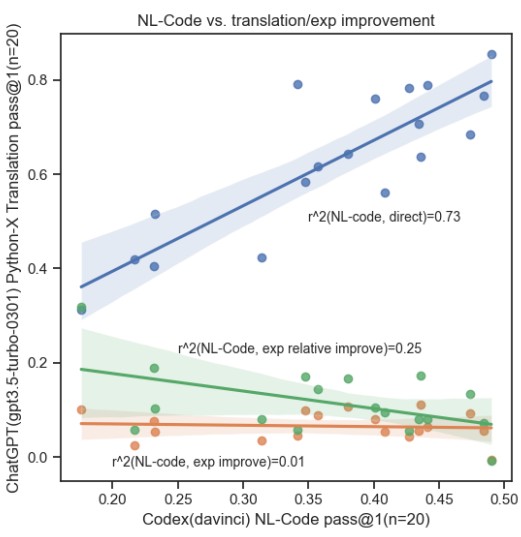

Figure 8: NL-to-code generation vs. Python-to-X translation performance (zero-shot)

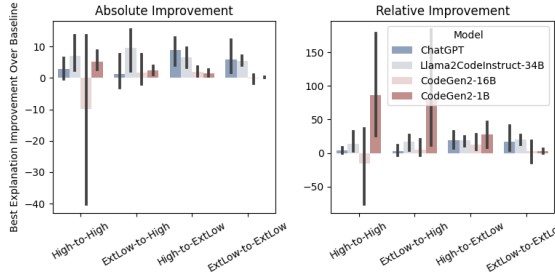

Figure 9: X-to-X translation with GPT-3.5 (pass@1, zero-shot) improvements (best explanation over baseline) across models grouped by source-target resource level

plot per-problem **direct** pass@1 and whether **exp** improves over **direct** (Fig 4). **direct** pass@1 rate serves as a good approximation of how difficult the problem is given the model. In main text we discuss that the **exp** improves difficult problems more often than easy problems. For easy problems (the right-most column), explanations can often decrease performance. Perhaps this is a result of redundant or inconsistent information leading to confusion. This indicates that a potential way to improve performance further is to automatically pick the difficult problems to explain.

To show that such strategy works, we assume access to oracle metric (**direct pass@1**) and leverage our cached generations from **direct** and **exp** translations. For each problem, if the **direct pass@1** rate is smaller than threshold (i.e. difficult problem), we use explanation, otherwise we use direct translation. We repeat this for all 36 translation directions in Python-to-X and X-to-X and present full results in Table 14. Immediately we can see that 1) low-resource languages typically require more explanations. 2) **select** almost always outperform **direct** and **exp** (only lost in 1 case with D→C++). In best case scenario (Racket → Julia), we see as much as 9.6% relative improvement over **exp** with **select**, while explaining less than half of the problems.

This is however still impractical for inference during test time. Having to approximate hardness through direct translation requires more computation than generating a single explanation. Ideally, one could build classifiers or use heuristics to select programs to translate. We leave this for future directions.

## J   Alternative latent language guidance

In addition to asking the model to generate explanations, we experimented with various forms of latent languages (in the order of more structurally formal to more free-form natural language). We report here their pass@1 (n=1) for Python-Java

- **Pivot language**: Instead of generating target program language directly, we also asked the program to translate to a pivot language and then translate to the target language. For initial experiment, we take the first generation from direct translation to the pivot language as intermediate step regardless of their accuracy. $pass@1_{C++} = 0.732, pass@1_{Bash} = 0.81, pass@1_R = 0.703$. More experiments in the next section.

- **Pseudocode**: An intermediate form of program sketch described with a mix of mathematical operations and natural language. To ensure the format of the pseudocode, we prompt with \\begin{algorithm} and use \\end{algorithm} as stop token. $pass@1 = 0.861$

- **CoT**: In Chain-of-thought (CoT) prompting, we break down the the input program space and translate each sub-components before combining all results together as a whole. In the decomposition phase, we try decomposing through model's perception of "steps" within algorithm, as well as programmatically extracting function calls within source programs that are often hard to translate (especially in low resource languages) $pass@1 = 0.734$

- **Steps**: ordered list of natural language steps describing major steps of the program following work by Jiang et al. (2023). $pass@1 = 0.824$

- **Summary**: free-form natural language sentences summary of what the program does. $pass@1 = 0.854$

- **Gold summary**: We use the original human written docstring instructions (from HumanEval) as gold summaries for the program and ask the model to translate given the program and summary. $pass@1 = 0.813$

| direction | # exp | total | direct | exp | select | $\delta$ |
|---|---|---|---|---|---|---|
| d-lua | 48 | 116 | 68.4 | 69.4 | 75 | 5.6 |
| py-d | 100 | 156 | 42 | 44.4 | 45.4 | 1 |
| py-sh | 115 | 158 | 47.9 | 55.4 | 55.8 | 0.4 |
| cpp-py | 19 | 149 | 92.3 | 90.2 | 93.6 | 1.3 |
| lua-py | 22 | 144 | 89.5 | 85.9 | 89.9 | 0.4 |
| py-rb | 47 | 161 | 78.3 | 78.3 | 81.7 | 3.4 |
| py-scala | 77 | 160 | 63.7 | 74.7 | 75.8 | 1.1 |
| py-swift | 69 | 161 | 64.4 | 70.5 | 71.4 | 0.9 |
| py-ts | 47 | 159 | 78.9 | 85.1 | 85.9 | 0.8 |
| py-cpp | 49 | 161 | 76.6 | 82.1 | 83.1 | 1 |
| java-cpp | 43 | 155 | 77 | 79.8 | 80.9 | 1.1 |
| py-jl | 71 | 159 | 61.6 | 70.5 | 71.2 | 0.7 |
| sh-java | 39 | 138 | 71.7 | 73.9 | 76.8 | 2.9 |
| jl-d | 90 | 137 | 41.6 | 43.4 | 43.7 | 0.3 |
| rkt-java | 58 | 136 | 65.9 | 77.1 | 82.6 | 5.5 |
| js-rkt | 117 | 154 | 30.2 | 41.9 | 44.3 | 2.4 |
| rkt-jl | 67 | 137 | 63.3 | 64.3 | 70.5 | 6.2 |
| java-jl | 72 | 154 | 60.2 | 75.4 | 76.1 | 0.7 |
| py-go | 96 | 154 | 42.4 | 45.7 | 47.5 | 1.8 |
| py-java | 50 | 158 | 76 | 82.9 | 84.2 | 1.3 |
| py-r | 113 | 161 | 40.4 | 46.9 | 48.6 | 1.7 |
| py-pl | 83 | 161 | 58.3 | 68.2 | 71.2 | 3 |
| py-rkt | 131 | 161 | 31.3 | 41.3 | 41.9 | 0.6 |
| js-java | 43 | 152 | 77 | 77.3 | 81.3 | 4 |
| jl-js | 29 | 138 | 83.1 | 83.5 | 85.8 | 2.3 |
| py-php | 59 | 161 | 68.4 | 77 | 77.8 | 0.8 |
| py-lua | 86 | 161 | 56.2 | 60 | 61.1 | 1.1 |
| py-rs | 58 | 156 | 70.6 | 74 | 74.8 | 0.8 |
| d-cpp | 21 | 116 | 88.4 | 81.4 | 87.8 | -0.6 |
| cpp-lua | 58 | 149 | 69.2 | 71.9 | 74 | 2.1 |
| py-cs | 45 | 158 | 79.2 | 83.6 | 85.1 | 1.5 |
| lua-rkt | 121 | 145 | 29.6 | 45.2 | 46.4 | 1.2 |
| py-js | 36 | 161 | 85.5 | 84.8 | 86.1 | 0.6 |

Table 14: Pass@1 for **direct**, **exp**, and **select** trails, where **select** picks only hard problems with **direct** pass@1 less than 0.9 to explain. **# exp** records the number of problems explained. $\delta$ is calculated as **select** $-$ max(**direct**, **exp**)

### J.1 model's dependency on pivot program accuracy

Within pivot program experiments (Python-C++-Java, Python-Bash-Java, Python-R-Java), we further analyzed the Java accuracy by measuring the subset accuracy: we split the set of source problems into those with a correct pivot translation, vs those with an incorrect pivot translation. Here is the result:

As seen in Table 15, there is no clear differences between the subset in which the pivot language passes or fails:

- For C++ and R, regardless of the pivot accuracy, Java translation accuracy **drops** with pivot.

- For Bash, regardless of the pivot accuracy, Java translation accuracy **improves** with pivot.

Although the improvement given pivot language seem monotonic on an aggregated level, this is not to say that pivot language has no effects on translation accuracy because the programs that are correctly translated in pivot language likely has some characteristic that can confound the translation accuracy.

To further investigate whether we can change individual behavior in an individual problem setting, we pick either only correct or incorrect programs sampled from ChatGPT and observe translation performance (Table 16). If we do not have such a correct/incorrect pivot program we discard that problem. Since there could be bias in the dataset where for a specific language, harder problems might have more likelihood of having incorrect problems than correct problems, we experiment with various high-/low resource combinations of target language and pivot, to be able to make conclusions overcoming such bias.

**Formal language as intermediate step can achieve equivalent or better results than natural language.** In Table 16, comparing **exp** (bottom row) against **Correct Pass@1**, we can see that sometimes using formal language as intermediate step can indeed reach or surpass using explanation as intermediate steps. Using higher-resource pivot language than the target language always seem to help more than using lower-resource language, except for **rkt-java**, which could just be evaluated on

an easier subset. This is intuitive because higher resource language generations in general are of better quality, and if the benefit of obtaining more information and generation length out-weights the noise, this is a valid way of boosting performance. Natural language can be thought of an extreme case of this with highest level of resource, with high probability of quality self-generated context.

**Formal intermediate steps are highly unpredictable.** By glancing at the difference between **Correct Pass@1** and **Incorrect Pass@1**, we can see incorrect pivot programs lead to drastically worse performance. If we observe the breakdown of the errors, we see a lot of **Incorrect Unhelpful@1**, indicating that the pivot programs themselves are unhelpful. Even if we assume models do not generate any unhelpful generations and combine **Incorrect Unhelpful@1** with **Incorrect Pass@1**, we still see a significant gap between incorrect and correct pivot programs. Specifically, **Incorrect Semantic Error@1** tends to be much higher than **Correct Semantic Error@1**. In ablation studies Table 4 we learned that when the wrong intermediate step is highly related to the source program in semantics, it decreases the translation performance more. In this experiment, since the semantics of source program and pivot program is almost identical, the mistakes in pivot program can have deleterious effects on translation.

**Natural language mistakes are taken less seriously** To compare the effect of having mistakes in natural language vs pivot programs, we included **Incorrect retrieved exp Pass@1** from Table 4. Since swapping an explanation with a closely related that of a similar problem guarantees the explanation to be wrong, we can compare this with **Incorrect Pass@1**. We found that on average, mistakes in natural language explanations do not decrease translation performance as much as programming language mistakes do.

## K GPT-3.5 score heuristics

In addition to two heuristics mentioned in Sec 3.4, we also try prompting GPT-3.5 to select the best explanation. We follow works in automatic generation evaluation with LLMs (Kocmi and Federmann, 2023; Wang et al., 2023b; Chen et al., 2023c; Fu et al., 2023; Wang et al., 2023b) and experiment with multiple-choice, direct assessment (generating a score between 0 and 100), and summarizing from

| pivot | java pass | overall pass | pass(direct\|T) | pass(exp\|T) | pass(direct\|F) | pass(direct\|F) |
|-------|-----------|--------------|-----------------|--------------|-----------------|-----------------|
| C++   |           | 76           | 86              | 81           | 42              | 31              |
| Bash  | 74        | 81           | 82              | 89           | 70              | 74              |
| R     |           | 70           | 85              | 77           | 71              | 68              |

Table 15: Translation using formal language as pivot intermediate "explanation". Results are obtained with pass@1(n=1). **direct** indicate translation accuracy with direct translation (no explanation), and **exp** means explain then translate. **T** indicate the pivot program is correct, and similarly for **F**. Hence, **pass(direct\|T)** means the pass rate for direct translation within the subset of programs that the pivot program is correct.

| pivot ablation (0 shot, n=20) | php-java | rkt-java | java-php | rkt-php | java-rkt | php-rkt |
|-------------------------------|----------|----------|----------|---------|----------|---------|
| # correct problems            | 154      | 136      | 155      | 138     | 155      | 147     |
| Correct Pass@1                | 80.3     | 84.3     | 79.7     | 69.6    | 40.8     | 37.9    |
| Correct Semantic Error@1      | 7.2      | 8.1      | 12.6     | 15.6    | 21.3     | 22.5    |
| Correct Syntax Error@1        | 12.5     | 7.6      | 7.7      | 14.8    | 37.8     | 39.6    |
| Correct Unhelpful@1           | 0        | 0        | 0        | 0       | 0        | 0       |
| # incorrect problems          | 154      | 149      | 140      | 152     | 140      | 157     |
| Incorrect Pass@1              | 35       | 75.8     | 43       | 63.9    | 23.7     | 8.3     |
| Incorrect Semantic Error@1    | 12.1     | 13.7     | 38.3     | 24.4    | 29.2     | 12.3    |
| Incorrect Syntax Error@1      | 11.2     | 10.5     | 6.5      | 11.8    | 36.3     | 17.4    |
| Incorrect Unhelpful@1         | 41.7     | 0        | 12.1     | 0       | 10.7     | 62      |
| direct Pass@1                 | 76       |          | 68.4     |         | 31.3     |         |
| exp Pass@1                    | 82.9     |          | 77       |         | 41.3     |         |
| Incorrect retrieved exp Pass@1| 75.8     |          | 73.7     |         | 31.1     |         |

Table 16: We use correct/ incorrect pivot program translation to observe the sensitivity models have in formal intermediate reasoning steps. All entries indicate 0-shot Pass@1(n=20). The column label indicate translation direction, pivot language, and the available problems generated. For example, **rkt-java** refers to translating **Python** to Java, with Racket as pivot language. **Semantic Error** is equivalent to assertion error, **Unhelpful** generations include incomplete code with comments like `"// TODO"`, `"// Write your code here"`. All other errors are grouped under **Syntax Error**s

multiple explanations. None of these methods outperformed random selection, so we do not include this method in Table 3. GPT-3.5-scores (direct assessment) of the explanations almost always fall between 90-100.

## L  Coder-reviewer details

Coder-Reviwer is a re-ranking method introduced by (Zhang et al., 2022) to re-rank code generations without verifying through symbolic tools (i.e. compilers) in NL-code tasks. The method found that averaging the logprob score from "coder" (which estimates a length-normalized $p(\text{code}|\text{NL})$) and "reviewer" (which estimates a length-normalized $p(\text{NL}|\text{code})$) can be used as a good metric to re-rank code generations. Formally the score is defined as:

$$\alpha \frac{logp(x|y)}{|x|} + (1 - \alpha)\frac{logp(y|x)}{|y|} \qquad (1)$$

where $x$ represents the natural language description of the code, $y$ represents generated code, and $\alpha$ is the hyperparameter that weighs the importance between the two terms.

In our problem, we have the inverse task of trying to find the best explanation $x$ given $y$. Since the score is symmetric, we use the same formula during re-ranking.

To calculate the logprobs, we used CodeGen2-12B(Nijkamp et al., 2023). We use prompt in L.1 and L.2. To obtain the best performance in estimated pass rate (Table 3), we try 0,1, or 2-shots (if GPU memory allows), and vary $\alpha$ between 0-1 with 0.1 interval (except for Python-Racket exp, which we tried 0.02 between 0.8-1.0 in addition to the rest). In Figure 10 we plot the best performing setting for each experiment trials across $\alpha$.

In section L.3 and section L.4, we show an example of correct and incorrect selection of explanations by coder-reviewer (To provide an idea of what explanations look like, we include only 5 out of 20 total explanations).

### L.1  Coder prompt

```
Can you write a Python program given
this explanation?

### Explanation

This function takes in a list of
integers and returns a boolean
```

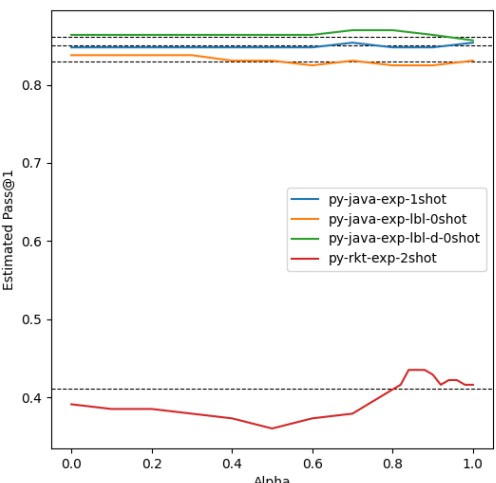

Figure 10: Coder-Reviewer best few-shot setting varying $\alpha$ hyper-parameter. Black dotted lines are average baseline performance where explanations are selected randomly.

```
indicating whether the list is sorted
in non-descending order and has no more
 than two occurrences of any integer.
The function first creates a dictionary
 where the keys are the integers in the
 list and the values are the number of
times that integer appears in the list.
 It then checks if any integer appears
more than twice in the list, and
returns False if so. Finally, it checks
 if the list is sorted in non-
descending order, and returns True if
so, and False otherwise.

### Python version

from typing import List

def is_sorted(lst: List[int]) -> bool:
    count_digit = dict([(i, 0) for i in
 lst])
    for i in lst:
        count_digit[i]+=1
    if any(count_digit[i] > 2 for i in
lst):
        return False
    if all(lst[i-1] <= lst[i] for i in
range(1, len(lst))):
        return True
    else:
        return False

### Explanation

This function takes a list of integers
'array' as input and returns a sorted
list of integers. The function first
checks if the input list is empty. If
it is, it returns an empty list.
```

### Python version

```python
from typing import List

def sort_array(array: List[int]) -> List[int]:
    return [] if len(array) == 0 else sorted(array, reverse= (array[0]+array[-1]) % 2 == 0)
```

### Explanation

<insert explanation here>

### Python version

<insert Python program here, calculate normalized log p on this sequence>

## L.2  Reviewer prompt

Can you explain what this Python program does in a couple of sentences?

### Python version

```python
from typing import List

def is_sorted(lst: List[int]) -> bool:
    count_digit = dict([(i, 0) for i in lst])
    for i in lst:
        count_digit[i]+=1
    if any(count_digit[i] > 2 for i in lst):
        return False
    if all(lst[i-1] <= lst[i] for i in range(1, len(lst))):
        return True
    else:
        return False
```

### Explanation

This function takes in a list of integers and returns a boolean indicating whether the list is sorted in non-descending order and has no more than two occurrences of any integer. The function first creates a dictionary where the keys are the integers in the list and the values are the number of times that integer appears in the list. It then checks if any integer appears more than twice in the list, and returns False if so. Finally, it checks

 if the list is sorted in non-descending order, and returns True if so, and False otherwise.

### Python version

```python
from typing import List

def sort_array(array: List[int]) -> List[int]:
    return [] if len(array) == 0 else sorted(array, reverse= (array[0]+array[-1]) % 2 == 0)
```

### Explanation

This function takes a list of integers `array` as input and returns a sorted list of integers. The function first checks if the input list is empty. If it is, it returns an empty list. Otherwise, it sorts the list using the `sorted` function. The `reverse` parameter is set to `True` if the sum of the first and last elements of the array is even (and the array will be sorted in descending order), otherwise, it is set to `False` (and the array will be sorted in ascending order).

### Python version

<insert Python program here>

### Explanation

<insert explanation here, calculate normalized log p on this sequence>

## L.3  Coder-Reviewer correct explanation selection

You are an helpful AI assistant who understands all programming languages and can translate between them at ease. Can you explain what this Python program does with a couple of sentences? The goal with the explanation, is so that a reader can easily rewrite the program in Java.

### Python version

```python
from typing import List

def match_parens(lst: List[str]) -> str:
    def check(s):
        val = 0
        for i in s:
            if i == '(':
                val = val + 1
            else:
                val = val - 1
            if val < 0:
```

```python
            return False
        return True if val == 0 else
False

    S1 = lst[0] + lst[1]
    S2 = lst[1] + lst[0]
    return 'Yes' if check(S1) or check(
S2) else 'No'
```

### Explanation 1 (pass 0/1)

This Python program checks if it is
possible to create a valid string of
parentheses by concatenating two
strings of parentheses. The function '
match_parens' takes a list of two
strings of parentheses as input and
returns 'Yes' if it is possible to
create a valid string of parentheses by
 concatenating the two strings, and 'No
' otherwise.

The function 'check' checks if a given
string of parentheses is valid. It does
 this by iterating over the string and
keeping a count of the number of
opening and closing parentheses. If the
 count of closing parentheses ever
exceeds the count of opening
parentheses, the string is invalid. If
the count of opening and closing
parentheses is equal at the end of the
iteration, the string is valid.

The function 'match_parens'
concatenates the two input strings in
two different orders and checks if
either of the resulting strings is
valid using the 'check' function. If
one of the resulting strings is valid,
the function returns 'Yes', otherwise
it returns 'No'.

### Explanation 2 (pass 0/1)

This program checks if it is possible
to form a string of balanced
parentheses by concatenating two
strings from a given list of strings.
It does this by checking if the sum of
the number of opening parentheses is
equal to the sum of the number of
closing parentheses. If the sum of the
opening parentheses is greater than the
 sum of the closing parentheses, then
it is not possible to form a string of
balanced parentheses. The program
returns 'Yes' if it is possible to form
 a string of balanced parentheses and '
No' otherwise.

### Explanation 3 (pass 0/1)

This program checks if two strings
concatenated together form a valid
string of balanced parentheses. It does
 this by iterating through each
character of the concatenated string

and keeping track of the number of open
 and closed parentheses. If the number
of closed parentheses ever exceeds the
number of open parentheses, the string
is invalid. If the number of open and
closed parentheses is equal at the end
of the string, the string is valid. The
 function returns 'Yes' if either the
original concatenated string or the
reversed concatenated string are valid,
 and 'No' otherwise.

### Explanation 4 (pass 1/1,
Coder-reviewer selected, actual pass
20/20)

This  Python program takes a list of
two strings as input, and checks if the
 two strings can be concatenated in any
 order to form a string with balanced
parentheses. If the parentheses are
balanced, the function returns 'Yes',
otherwise it returns 'No'.

The function 'check' takes a string as
input and checks if the parentheses in
the string are balanced. It does this
by iterating through the string and
keeping track of the number of open and
 closed parentheses encountered so far.
 If the number of closed parentheses
encountered is greater than the number
of open parentheses encountered, the
function returns False. If the number
of open and closed parentheses is equal
 at the end of the string, the function
 returns True, otherwise it returns
False.

The function then concatenates the two
input strings in both possible orders
and checks if either of the resulting
strings have balanced parentheses. If
at least one of the strings has
balanced parentheses, the function
returns 'Yes', otherwise it returns 'No
'.

To rewrite this program in Java, you
would need to replace the 'List' type
with an array of strings, and use a for
 loop to iterate through the characters
 in the input strings. You would also
need to use the 'StringBuilder' class
to concatenate the strings. Here is an
example Java implementation:

```java
public static String matchParens(String
[] lst) {
    StringBuilder s1 = new
StringBuilder(lst[0]).append(lst[1]);
    StringBuilder s2 = new
StringBuilder(lst[1]).append(lst[0]);
    if (check(s1.toString()) || check(
s2.toString())) {
        return "Yes";
    } else {
```

```
        return "No";
    }
}

public static boolean check(String s) {
    int val = 0;
    for (int i = 0; i < s.length(); i
++) {
        if (s.charAt(i) == '(') {
            val++;
        } else {
            val--;
        }
        if (val < 0) {
            return false;
        }
    }
    return val == 0;
}
}
```<|im_end|>
```

This  program checks if it is possible to create a valid string of parentheses by concatenating two strings from a list of strings. The function 'match_parens' takes a list of strings as input and returns 'Yes' if it is possible to create a valid string of parentheses by concatenating two strings from the list, and 'No' otherwise. The function 'check' checks if a string of parentheses is valid by counting the number of opening and closing parentheses. If the number of closing parentheses is greater than the number of opening parentheses at any point, the string is invalid. If the number of opening and closing parentheses is equal at the end of the string, the string is valid. The function 'match_parens' concatenates the first two strings in the list in two different orders and checks if either of the resulting strings is valid. If either string is valid, the function returns 'Yes', otherwise it returns 'No'.

## L.4    Coder-Reviewer incorrect explanation selection

You are an helpful AI assistant who understands all programming languages and can translate between them at ease. Can you explain what this Python program does with a couple of sentences? The goal with the explanation, is so that a reader can easily rewrite the program in Java.

```python
from typing import List
```

```python
def words_string(s: str) -> List[str]:
    if not s:
        return []

    s_list = []

    for letter in s:
        if letter == ',':
            s_list.append(' ')
        else:
            s_list.append(letter)

    s_list = "".join(s_list)
    return s_list.split()
```

This program takes a string as input and returns a list of words. The input string is first converted into a list of characters. If a comma is encountered in the string, it is replaced with a space. The list of characters is then joined back into a string and split into a list of words using the default whitespace separator. If the input string is empty, an empty list is returned.

This program takes a string as input, and returns a list of words. It does this by first checking if the string is empty, and if it is, it returns an empty list. If the string is not empty, it creates an empty list called 's_list', and then iterates over each letter in the input string. If the letter is a comma, it appends a space to 's_list', otherwise it appends the letter itself. After iterating over all the letters in the input string, it joins all the elements in 's_list' into a single string, and then splits that string into a list of words using the default whitespace separator.

This  Python program takes a string as input and returns a list of words in the string. The function 'words_string' first checks if the input string is empty, and if it is, returns an empty list. If the input string is not empty, the function creates an empty list 's_list', and iterates over each character in the input string. If the character is a comma, the function appends a space to 's_list', otherwise it appends the character. The function then joins 's_list' into a single string, and splits the string into a list of words using the 'split()' method.

```
### Explanation 4 (pass 0/1)

This  program takes in a string as
input and returns a list of words. The
input string is first split into a list
 of characters. If a character is a
comma, it is replaced with a space. The
 list of characters is then joined back
 into a string and split into a list of
 words. If the input string is empty,
an empty list is returned.
```

```
### Explanation 5 (pass 0/1)

This function takes a string as input
and returns a list of words. It does
this by first converting all commas in
the string to spaces, and then
splitting the string into a list of
words. If the input string is empty, it
 returns an empty list.
```

## M Alternative explanation-selection setting

In Sec 3.4, we sample 20 explanations for each problem and generate 1 program from each explanation. By sampling more than 1 programs for each explanation, one could obtain variances of the performance estimates (by simulating pass status of each program according to the pass rates of the selected explanations in the **train set**), but we find sampling one program to be good enough at estimating final performance. Plus, given the same budget, it is also much better to sample 20 X 1 (Table 3) than 4 X 5 to maximize explanations diversity, and have the potential of sampling the best explanation. Table 17 shows the result for 4 X 5 experiments.

Comparing results in Table 17 to the main results in Table 3, we see much less improvements using heuristics. It is likely that 4 is not a large enough sample size to obtain the the correct explanation for some problems, resulting in low coverage and small improvements from heuristics.

## N Program obfuscation

```
### Original Python version

def make_palindrome(string: str) -> str
:
    if not string:
        return ''

    beginning_of_suffix = 0
```

```
    while string[beginning_of_suffix:]
!= string[beginning_of_suffix:][::-1]:
        beginning_of_suffix += 1

    return string + string[:
beginning_of_suffix][::-1]
```

```
### Original Explanation

This function takes a string as input
and returns a palindrome by appending
the reverse of the string's suffix to
the string. The suffix is the smallest
substring that needs to be appended to
the string to make it a palindrome. If
the input string is empty, the function
 returns an empty string.
```

```
### Obfuscated Python version

def FUNC_0(VAR_0: str) -> str:
    if (not VAR_0):
        return ''
    VAR_1 = 0
    while (VAR_0[VAR_1:] != VAR_0[VAR_1
:][::(- 1)]):
        VAR_1 += 1
    return (VAR_0 + VAR_0[:VAR_1][::(-
1)])
```

```
### Obfuscated Explanation

This function takes a string as input
and returns a string. If the input
string is empty, it returns an empty
string. Otherwise, it finds the longest
 suffix of the input string that is
also a prefix of the reverse of the
input string. It then returns the input
 string concatenated with the reverse
of the remaining part of the input
string.
```

Above is an example of the program (`humaneval_10_make_palindrome`) before and after obfuscation using tools from (Lachaux et al., 2020). After obfuscation, function and variable names are all replaced with respective surface forms, as the functionality of the program remains unchanged. As the example indicates, explanation quality does not really decrease. In fact, explanations often become more detailed just because there is not a generic way of describing some operation/term like "palindrome".

To qualitatively examine explanations' effect on translating semantically confusing programs, we translated obfuscated Python programs using **direct**, **exp**, and **exp-lbl** (Table 18). Similar to Python-to-X experiment, we generate explanations with Python-Java, remove Java specific explanations, and re-use explanations across the rest of the

| Exp Type | src-tgt | random | len | line-e | line | frag | oracle |
|---|---|---|---|---|---|---|---|
| exp | py-jv | **82.1** ± 1.4 | 81.0 ± 1.1 | 81.1 ± 0.9 | 81.0 ± 1.0 | 79.4 ± 1.1 | 89.9 |
| exp-lbl | py-jv | 85.7 ± 1.0 | **86.5** ± 0.9 | 86.3 ± 0.8 | 86.2 ± 0.8 | 85.2 ± 1.0 | 90.9 |
| exp-lbl-d | py-jv | 85.9 ± 1.1 | **86.4** ± 0.6 | 85.7 ± 0.7 | 85.9 ± 0.6 | 85.8 ± 0.7 | 90.3 |

Table 17: Explanation selection heuristics performance. We estimate heuristics performance (pass@1, n=1).

| | jv | php | sw | rkt | jv* |
|---|---|---|---|---|---|
| direct | 76.0 | 68.4 | 64.4 | 31.3 | 76.0 |
| exp | 82.9 | 77 | 70.5 | 41.3 | 82.9 |
| exp-lbl | 82.5 | 77.5 | 75.1 | 39 | 82.5 |
| max($\Delta$)% | 9.1 | 13.3 | 16.6 | 31.9 | 9.1 |
| obf direct | 72.6 | 60.3 | 50.9 | 27.8 | 72.6 |
| obf exp | 78.8 | 72.7 | 72.2 | 34.6 | 79.6 |
| obf exp-lbl | 79.9 | 73.2 | 72.7 | 35.9 | 79.4 |
| max($\Delta$)% | 10.1 | 21.4 | 42.8 | 29.1 | 9.6 |

Table 18: Translation performance after **obf**uscating source Python programs. Each entry measures pass@1 (n=20). max($\Delta$)% measures maximum improvement explanations can bring on top of direct translation. **jv\*** column lists explanation performance before removing Java specific information.

directions. We find that explanation is robust regardless of surface semantics / readability of source code. In **direct**, we see consistent drops in performance across all translation directions. However, explanations still provide consistent improvements across all four languages. The relative improvements from explanation is even larger in 2 out of 4 directions for obfuscated programs vs. non-obfuscated programs.

## O  Program Retrieval

```
### Query Python code

def correct_bracketing(brackets: str)
-> bool:
    depth = 0
    for b in brackets:
        if b == ""("":
            depth += 1
        else:
            depth -= 1
        if depth < 0:
            return False
    return depth == 0

### Retrieved Python code]

def correct_bracketing(brackets: str)
-> bool:
    depth = 0
    for b in brackets:
        if b == ""<"":
            depth += 1
        else:
            depth -= 1
```

```
        if depth < 0:
            return False
    return depth == 0

### Query Python code]

def fib(n: int) -> int:
    if n == 0:
        return 0
    if n == 1:
        return 1
    return fib(n - 1) + fib(n - 2)

### Retrieved Python code]

def fibfib(n: int) -> int:
    if n == 0:
        return 0
    if n == 1:
        return 0
    if n == 2:
        return 1
    return fibfib(n - 1) + fibfib(n -
2) + fibfib(n - 3)
```

Above are examples of retrieved Python programs given query programs. For retrieval, we tokenize the python program the same way as TransCoder using 'tokenize' library [12]. We then use `BM250kapi` algorithm in `rank_bm25` library to retrieve for the most similar program within the HumanEval Dataset. Although one can improve retrieval similarity by using more sophisticated methods (dense embeddings such as UniXCoder or syntax aware similarity metrics such as CodeBLEU), we find BM25 to be cheap and effective at retrieving similar programs for our ablation studies.

## P  Intermediate step lengths

In Table 19, we report the length of the intermediate steps (explanation, pivot programs) and their respective ratio to the source program length. One could argue, that the increased length in intermediate step could lead to more computation in attention, decoding, which leads to improvement in downstream translation. Here, we note several observations and leave a detailed investigation as a direction for future work.

[12]https://docs.python.org/3/library/tokenize.html

|  | exp | exp/src ratio |
|---|---|---|
| Python-to-X exp | $96 \pm 70$ | $1.55 \pm 1.04$ |
| Python-to-X exp (tgt-specific) | $101 \pm 69$ | $1.70 \pm 1.09$ |
| Python-to-X exp (four-shot) | $122 \pm 43$ | $2.06 \pm 0.90$ |
| Python-to-X exp (four-shot, tgt-specific) | $102 \pm 46$ | $1.71 \pm 0.79$ |
| Python-to-X exp (four-shot, coder-reviewer) | $228 \pm 153$ | $4.09 \pm 3.84$ |
| Python-to-X exp-lbl | $305 \pm 126$ | $5.02 \pm 2.02$ |
| Python-to-X exp-lbl (four-shot) | $195 \pm 80$ | $3.08 \pm 0.91$ |
| Python-to-X exp-lbl-d | $316 \pm 121$ | $5.26 \pm 2.27$ |
| Python-to-X exp-lbl-d (four-shot) | $275 \pm 118$ | $4.45 \pm 1.68$ |
| Python-to-X exp-lbl-d (four-shot, frag) | $305 \pm 123$ | $5.16 \pm 1.96$ |
| Pivot (java) | $124 \pm 52$ | $2.01 \pm 0.75$ |
| Pivot (php) | $94 \pm 52$ | $1.31 \pm 0.23$ |
| Pivot (rkt) | $80 \pm 51$ | $1.13 \pm 0.32$ |

Table 19: Intermediate step (explanations, pivot program) length and their ratio to source Python program

**More detailed explanation is longer** However, as we have noted in Table 1, more detailed explanations do not always lead to more improvements. In high-low-resource directions, more generic (shorter) explanations often works better. This is one of the examples where length does not correlate well with performance.

**Target-specific information improves performance** In zero-shot or four-shot settings (Table G.1, G.2), we see slight improvement with target specific explanations. However, length-wise, we do not see a pattern between target-specific explanation vs. target-independent explanation in 0 and four-shot setting.

**Heuristically selected explanations are longer** Compare **Python-to-X exp (four-shot, coder-reviewer)** vs. **Python-to-X exp (four-shot)** and **Python-to-X exp-lbl-d (four-shot, frag)** vs. **Python-to-X exp-lbl-d (four-shot)**, we can see both heuristically selected explanations are longer than their random baselines. However, as seen in Table 3, **len** heuristics do not do nearly as well as winning heuristics. This indicates that length is important, but is not all the signal in determining the success in translation.

**Formal intermediate steps can be more efficient** In Table 16, we see a similar scale improvements from using **correct** pivot programs as intermediate steps. We conclude from the table that using higher resource language as pivot works better, and in this case we do see higher-resource language tend to be longer than lower-resource languages. It would be interesting to understand how does the verbosity of a language correlate to their usefulness as an intermediate reasoning step.