# OpenReview forum: "Explain-then-translate: an analysis on improving program translation with self-generated explanations"
_EMNLP/2023/Conference — EMNLP 2023 Findings_

### Official Review · Reviewer_XyFJ · 2023-08-04

**Soundness:** 4

**Excitement:**

3: Ambivalent: It has merits (e.g., it reports state-of-the-art results, the idea is nice), but there are key weaknesses (e.g., it describes incremental work), and it can significantly benefit from another round of revision. However, I won't object to accepting it if my co-reviewers champion it.

**Missing References:**

N/A


**Paper Topic And Main Contributions:**

This paper investigates whether or not self-generated natural language explanations can improve code-to-code translation performance. The paper proposes an explain-then-translate method and analyzes its effectiveness across 19 different programming languages and 3 types of explanations. It also proposes 5 different heuristics to select better explanations to improve the performance of its explain-then-translate method. Finally, it conducts multiple ablation studies comparing the robustness and effectiveness of natural language explanations to programming language examples, commonly seen in few shot learning.

**Questions For The Authors:**

Question A: In Section 4.3, why were 0-shot explanations used for exp, while 4-shot explanations were used for exp-lbl and exp-lbl-d? I don’t see a good reason for being inconsistent here.
Question B: Is there any mention in the paper of the compute resources used to run the inference performed in the experiments? Since all of the inference was performed on ChatGPT, perhaps information on the number of API keys used, as well as the time it took to run all the inference would be helpful as well.
Question C: Why were only a subset of all the possible translation directions used when experimenting with alternative translation directions in Section 4.2?
Question D: Why was ChatGPT mentioned as the LLM model used for inference only until the conclusion? I believe that is a relevant detail that could be included much earlier.


**Reasons To Accept:**

1. This paper is to my knowledge the first highly detailed study into how self-generated natural language explanations can improve code-to-code translation performance.
2. This paper provides detailed examples of the various prompts used in the study, which will be helpful for the future reproduction of the results in the study as well as future research in this area.
3. This paper provides novel heuristic selection strategies that improve the performance of its explain-then-translate method, as well as showcase the potential for growth and, thus, can be used as a blueprint for future research in the area of code-to-code translation.
4. This paper provides results of novel ablation experiments that demonstrate how natural language explanations can be more robust and offer better performance when compared to automatically generated programming examples that are used in one-shot learning.


**Reasons To Reject:**

1. The authors claim to release their code and dataset that they used for this paper, however I cannot find any links to these resources in the paper.
2. Section 3, where the authors explain some of their methodology on how they are prompting the LLM, is quite vague, as it does not mention what LLM they are using nor does it clearly outline how they are using their explain-then-translate method to prompt the LLM, both of which may make reproducibility more difficult.


**Reproducibility:**

4: Could mostly reproduce the results, but there may be some variation because of sample variance or minor variations in their interpretation of the protocol or method.

**Reviewer Confidence:**

4: Quite sure. I tried to check the important points carefully. It's unlikely, though conceivable, that I missed something that should affect my ratings.

**Typos Grammar Style And Presentation Improvements:**

Lines 050-051, …0-shot setting… → …0-shot performance…
Line 094, …code generation prompt… → …code generation prompts…
Line 111-115, …over CodeXGLUE… → …over CodeXGLUE and TransCoder…
Line 491, …Python-Java++-PHP… → …Python-Java-PHP…

---

> ### Author Rebuttal · Authors · 2023-08-29
>
> Thank you for such a thorough review! We appreciate the attention to all the details. We will fix the typos and low-level mistakes pointed out and address some of the bigger questions here:
>
> > The authors claim to release their code and dataset that they used for this paper, however, I cannot find any links to these resources in the paper.
>
> Please find the source code and artifacts generated in the zip file submitted with the original draft in OpenReview. As we finish cleaning up our repository, we will open-source our code on GitHub and make our artifacts downloadable through a public drive link. (The Github link is removed from the paper for anonymity)
>
> > Section 3, where the authors explain some of their methodology on how they are prompting the LLM, is quite vague, as it does not mention what LLM they are using nor does it clearly outline how they are using their explain-then-translate method to prompt the LLM, both of which may make reproducibility more difficult.
>
> For the language model used in our experiments, we used ChatGPT through the Microsoft Azure completion endpoint. For the next version of the paper, we will incorporate additional experiment results with open-source models (CodeGen2-16B, CodeGen2-1B, CodeGen2-1B with ChatGPT explanations, and StarCoder), most of which can be found in the rebuttal response for reviewer#1. For detailed prompts, please see Appendix C. Prompts with explanations are generated in two separate steps. Once an explanation is generated for a source program, we append the explanation and the target code signature to the prompt (instruction, source program) and feed it to the model. Once programs are generated, we run a post-processing script (Appendix D) to remove extra natural language comments, and program statements outside of the function.
>
> > Question A:  In Section 4.3, why were 0-shot explanations used for exp, while 4-shot explanations were used for exp-lbl and exp-lbl-d? I don’t see a good reason for being inconsistent here.
>
> In Section 4.1.2 (second paragraph) and Apx F.2, we find few-shot explanations to have lower qualities than zero-shot explanations for exp (but not for exp-lbl or exp-lbl-d), hence we use the best shot-setting to produce the pool of explanations from which we do our heuristic selection. For exp-lbl and exp-lbl-d, since the task is a bit more complex and requires few-shot guidance, we use few-shot generated explanations.
>
> > Question B: Is there any mention in the paper of the compute resources used to run the inference performed in the experiments? Since all of the inference was performed on ChatGPT, perhaps information on the number of API keys used, as well as the time it took to run all the inference would be helpful as well.
>
> Thanks for pointing this out. We will include the following section in the next version of our paper:
>
> The completion queries are made to ChatGPT (gpt-3.5-turbo-3010) between March - June 2023. We use the Azure completion endpoint (as opposed to the OpenAI chat completion endpoint). Compute credits are from XXX lab. The average query time for a single experiment (e.g. exp, Python -> Julia) takes around 10-40 minutes with one API key (of which we only used one). For Tables 1 and 2, the total query time is between 21 - 84 hours. The experiments for CodeGen2-1B are conducted on XXX and XXX clusters with NVIDIA RTX V100, A100, and A6000. Each single experiment takes 12-24 hours so the whole compute time for table 1&2 completions is around 72-144 GPU days. Code execution/evaluation is done locally on the same MacBook Pro 2015. For future studies and reproducibility, we will release docker/podman evaluations in the same way as the MultiPL-E dataset.
>
> > Question C: Why were only a subset of all the possible translation directions used when experimenting with alternative translation directions in Section 4.2?
>
> The subset is chosen to mainly save computational cost while maintaining a representation of the dataset. Since we can divide the languages roughly by resource level and whether the languages are typed or not, we sample extreme ends of the resource level (high vs. extremely low) and make sure all combinations of type characteristics (typed -> dynamic, etc.) are included in the subset. The reason to select high-resource languages is to observe the best possible performance, while extremely low-resource languages are also crucial for understanding performance bottlenecks and are important for real-world applications (e.g. code modernization).
>
> > Question D: Why was ChatGPT mentioned as the LLM model used for inference only until the conclusion? I believe that is a relevant detail that could be included much earlier.
>
> Thanks for pointing this out. Since we are adding additional experiments on open-source models, we will remove that limitation and mention the models used at the beginning of our paper.
>
> In the time available during the rebuttal period, we are able to reproduce most of Table1, Table2 for 0-shot setting with more open source models (CodeGen2-16B, CodeGen2-1B). We plan to finish the experiments with StarCoder as well to compare with the current open source SOTA code model. Additionally, we tested whether smaller models can benefit from better explanations generated by big models (prompting CodeGen2-1B with ChatGPT generated explanations). We hope the tables of results and additional comparison of results across open-source models can be taken into consideration.
>
> ### Python-to-X Translation (Similar to paper table 1)
>
> Translation pass@1 from Python to X. Parenthesis in trial indicates # of shots. Best within same explanation source model is *italicized* and overall best is **bolded**. All parameters are kept the same as table 1 in paper.
>
> ##### Table 1.1 (w/ CodeGen2-1B, self generated explanation and ChatGPT-generated explanation)
>
> | exp  | js | cpp | jv | ts | php | rb | cs | go | pl | r | rs | sc | sw | sh | lua | rkt | jl | d |
> | --- | --- | --- | --- | --- | --- | --- | --- | --- | --- | --- | --- | --- | --- | --- | --- | --- | --- | --- |
> | baseline (0shot) | 4.6 | 9.4 | 5.3 | 7 | 13.4 | 0 | 6.4 | 3.5 | 3.6 | 0.6 | 4.7 | 2.6 | 3.5 | ***5.7*** | 2.7 | 0 | 2.4 | 6.2 |
> | exb (0shot) | *15.4* | *11.7* | ***7.8*** | *13.5* | 14.2 | ***5.9*** | 7.9 | 8.3 | 4.1 | 0.8 | *8.7* | *5.6* | *6.9* | 4.8 | 5.3 | 0 | *3.1* | ***7.9*** |
> | exb-lbl (0shot) | 13.9 | 11.1 | 6.5 | 11.3 | *17.3* |  |  |  |  |  |  |  |  | 3.1 |  |  |  | 7.3 |
> | exb-lbl-d (0shot) | 12.7 | *11.7* | 7.6 | 9.9 | 14.6 | 3.9 | *8.5* | *8.8* | *4.5* | ***0.9*** | 8.2 | 2.4 | *6.9* | 4.7 | ***5.4*** | 0 | 2.8 | 5.9 |
> | --- | --- | --- | --- | --- | --- | --- | --- | --- | --- | --- | --- | --- | --- | --- | --- | --- | --- | --- |
> | exb (0shot) w/ ChatGPT | ***19.3*** | ***15.5*** | 5.8 | 10.8 | 16.5 | 3.9 | 11.2 | 9.8 | *4.2* | 0.4 | ***10.3*** | ***7*** | ***10.4*** | 5.2 | 4.8 | **0.1** | **3.7** | 7.2 |
> | exb-lbl (0shot) w/ ChatGPT | 16.2 | 14.3 |  | ***14.3*** | ***17.6*** | *4.7* | ***11.5*** | ***10.7*** | 3.6 | *0.7* | 9.5 | 5.4 | 10.3 |  |  | **0.1** |  |  |
> | exb-lbl-d (0shot) w/ ChatGPT |  |  |  |  |  |  |  |  |  |  |  |  |  |  |  |  |  |  |
>
>
> ##### Table 1.2 (w/ CodeGen2-16B)
>
> | exp  | js | cpp | jv | ts | php | rb | cs | go | pl | r | rs | sc | sw | sh | lua | rkt | jl | d |
> | --- | --- | --- | --- | --- | --- | --- | --- | --- | --- | --- | --- | --- | --- | --- | --- | --- | --- | --- |
> | baseline (0shot) | **48.4** | **41.2** |  | **44.8** | **38.4** | **8.6** | **46** |  |  |  |  |  |  |  |  |  |  | 11.7 |
> | exp (0shot) | 45.1 | 38.2 |  | 43.7 | 37.5 | 2.8 | 41.8 |  |  |  |  |  |  |  |  |  |  | 14.4 |
> | explain-lbl (0shot) | 45.1 |  |  |  |  |  |  |  |  |  |  |  |  |  |  |  |  | **15.8** |
> | explain-lbl-d (0shot) | 43.2 |  |  |  |  |  |  |  |  |  |  |  |  |  |  |  |  | 15 |
>
>
>
>
> ### X-to-X Translation (Similar to paper table 2)
>
> For the two tables below, we report translation pass@1 between 16 different pairs of languages. **Resource** indicates the language resource levels of the source and target. **Type** indicates the source and target language typing characteristics
> (D/S=dynamically/statically typed). The best runs within the same-shot setting are **bolded**. All parameters are kept the same as table 2 in paper.
>
> ##### Table 2.1 (w/ CodeGen2-1B)
>
> |  |  |  | 0 shot |  |  |  |
> | --- | --- | --- | --- | --- | --- | --- |
> | Resource | Typing | src-tgt | direct | exp | exp-lbl | exp-lbl-d |
> | High-to-High | D-D | Python - JavaScript | 4.6 |  **14.9**  | 13.9 | 13.1 |
> |  | D-T | JavaScript - Java | 7.8 |  **11.2** | 9.6 | 10.1 |
> |  | T-D | C++ - Python | 8.3 | **13.9** | 13.5 | 10.7 |
> |  | T-T | Java - C++ | 14.2 | 15.6 | 14.8 | **17.5** |
> | High-to-ExtLow | D-D | JavaScript - Racket | 0 | 0 | 0 | 0 |
> |  | D-T | Python - D | 6.2 | **7.9** | 7.3 | 5.9 |
> |  | T-D | C++ - Lua | 8.9 | **11.8** | 10 | 9.8 |
> |  | T-T | Java - Julia | 3.2 | **4.9** | 3.9 | 4.4 |
> | ExtLow-to-High | D-D | Lua - Python | 1.9 | **6** | 5.9 | 5.6 |
> |  | D-T | Racket - Java | 7.9 | 7.6 | **8.1** | 7.9 |
> |  | T-D | Julia - JavaScript | 3.7 | **6.9** | 3.9 | 5.1 |
> |  | T-T | D - C++ | 11.7 | 12.7 | 12.6 | **14.1** |
> | ExtLow-toExtLow | D-D | Lua - Racket | 0 | 0 | 0 | 0 |
> |  | D-T | Racket - Julia | 2 | **2.1** | 1.5 | 0.8 |
> |  | T-D | D - Lua | 7.7 | **8.2** | 6.3 | 7.2 |
> |  | T-T | Julia - D | 0 | **0.007** | 0 | 0 |
>
> ##### Table 2.2 (w/ CodeGen2-16B)
>
> |  |  |  | 0 shot |  |  |  |
> | --- | --- | --- | --- | --- | --- | --- |
> | High-to-High | D-D | Python - JavaScript | **48.4** | 45.1 | 45.1 | 43.1 |
> |  | D-T | JavaScript - Java | **47.3** | 44.3 | 42 | 40 |
> |  | T-D | C++ - Python | 38 | **57.2** | 55.1 |  |
> |  | T-T | Java - C++ | 52.7 |  |  |  |
> | High-to-ExtLow | D-D | JavaScript - Racket | 2 |  |  |  |
> |  | D-T | Python - D | 11.7 | 14.4 | **15.8** | 15 |
> |  | T-D | C++ - Lua | 33.6 |  |  |  |
> |  | T-T | Java - Julia | 26.4 | **27.9** | 25.8 | 25.9 |
> | ExtLow-to-High | D-D | Lua - Python | 37.6 | 45.6 | **47.8** | 46.7 |
> |  | D-T | Racket - Java | **28.4** | 27.8 |  |  |
> |  | T-D | Julia - JavaScript | 40.1 | 39.7 | 40 | **42.9** |
> |  | T-T | D - C++ | **53.1** | 47.6 | 47.1 | 50.4 |
> | ExtLow-toExtLow | D-D | Lua - Racket | 0.8 |  |  |  |
> |  | D-T | Racket - Julia | 13.1 | **14.6** | **14.6** | 14.4 |
> |  | T-D | D - Lua | **29.3** | 28.1 | 28.8 | 28.9 |
> |  | T-T | Julia - D | **10.1** | 7.8 | 5.4 | 5.3 |
>
>
>
> From the tables above, we can draw the following additional conclusions, which we will include in the next version of the paper:
> - In weaker models, **we still see improvements with self-generated explanations across most directions**. CodeGen2-1B improves more consistently using self-generated explanations (baseline is the worst in all X-to-X directions, and in 17/18 Python-to-X directions), as much as 300%+ improvement (Table 2.3, Lua→Python, Python→JavaScript). In Python→Ruby, model with explanations obtains a pass rate of 5.9, while baseline does not generate *any* single correct translation (pass rate of 0). CodeGen2-16B shows weaker results, with baseline outperforming in 10/18 directions in Table 1 and 2. Perhaps it has a weaker alignment between natural language and programming language, resulting in worse explanations generated for each problem. The majority of errors from translation with explanation are syntactic.
> - **Better explanation leads to better translation, even in smaller model cases**. We compare CodeGen2-1B performance given self-generated explanation vs. ChatGPT generated explanation and see that the better explanation outperforms self-generated explanation in 12/18 Python-to-X directions, with a maximum improvement of 400%+ in Python → JavaScript.
> - In smaller/weaker models, **detailed explanations (exp-lbl or exp-lbl-d) do not improve as much as exp does**. Often this is due to lower quality explanations generated when the model is asked to do something it is not capable of. The line-by-line explanations often lead to repetitive content when source programs contain several repetitive lines (library import in C++ → Python direction, with CodeGen2-1B)
> - we will add these results and results from StarCoder in the revised version of our paper

---

### Official Review · Reviewer_CiwV · 2023-08-06

**Soundness:** 3

**Excitement:**

4: Strong: This paper deepens the understanding of some phenomenon or lowers the barriers to an existing research direction.

**Paper Topic And Main Contributions:**

This paper experimented with chain-of-thought prompting in the case of programming language translation. With the aid of zero-shot and few-shot prompting, the authors came across important insights such as, adding simpler explanations turning out more useful in case of low-resourced ones and using simpler heuristics to perform better explanation selection.

**Questions For The Authors:**

- could you add some comparative analysis comprising multiple LLMs focusing on the key findings?

**Reasons To Accept:**

- Detailed study and adequate explanations regarding the reasonings
- Detailed ablation study

**Reasons To Reject:**

- this study is based on only chatgpt without any room of comparative evaluation. There needs to be some analysis comprising other llm so that the behaviors and finds could be confidently described as general.

**Reproducibility:**

5: Could easily reproduce the results.

**Reviewer Confidence:**

3: Pretty sure, but there's a chance I missed something. Although I have a good feel for this area in general, I did not carefully check the paper's details, e.g., the math, experimental design, or novelty.

---

> ### Author Rebuttal · Authors · 2023-08-29
>
> Thanks for the review and comments! Taking both reviewer#1 and this feedback, we have provided additional tables that we will add to the next version of our paper paper. With the additional data points using opensource models, we can compare model performance over different training dataset and model size, as well as improving reproducibility of our claims with “white-box” models.
>
> In the time available during the rebuttal period, we are able to reproduce most of Table1, Table2 for 0-shot setting with more open source models (CodeGen2-16B, CodeGen2-1B). We plan to finish the experiments with StarCoder as well to compare with the current open source SOTA code model. Additionally, we tested whether smaller models can benefit from better explanations generated by big models (prompting CodeGen2-1B with ChatGPT generated explanations). We hope the tables of results and additional comparison of results across open-source models can be taken into consideration.
>
> ### Python-to-X Translation (Similar to paper table 1)
>
> Translation pass@1 from Python to X. Parenthesis in trial indicates # of shots. Best within same explanation source model is *italicized* and overall best is **bolded**. All parameters are kept the same as table 1 in paper.
>
> ##### Table 1.1 (w/ CodeGen2-1B, self generated explanation and ChatGPT-generated explanation)
>
> | exp  | js | cpp | jv | ts | php | rb | cs | go | pl | r | rs | sc | sw | sh | lua | rkt | jl | d |
> | --- | --- | --- | --- | --- | --- | --- | --- | --- | --- | --- | --- | --- | --- | --- | --- | --- | --- | --- |
> | baseline (0shot) | 4.6 | 9.4 | 5.3 | 7 | 13.4 | 0 | 6.4 | 3.5 | 3.6 | 0.6 | 4.7 | 2.6 | 3.5 | ***5.7*** | 2.7 | 0 | 2.4 | 6.2 |
> | exb (0shot) | *15.4* | *11.7* | ***7.8*** | *13.5* | 14.2 | ***5.9*** | 7.9 | 8.3 | 4.1 | 0.8 | *8.7* | *5.6* | *6.9* | 4.8 | 5.3 | 0 | *3.1* | ***7.9*** |
> | exb-lbl (0shot) | 13.9 | 11.1 | 6.5 | 11.3 | *17.3* |  |  |  |  |  |  |  |  | 3.1 |  |  |  | 7.3 |
> | exb-lbl-d (0shot) | 12.7 | *11.7* | 7.6 | 9.9 | 14.6 | 3.9 | *8.5* | *8.8* | *4.5* | ***0.9*** | 8.2 | 2.4 | *6.9* | 4.7 | ***5.4*** | 0 | 2.8 | 5.9 |
> | --- | --- | --- | --- | --- | --- | --- | --- | --- | --- | --- | --- | --- | --- | --- | --- | --- | --- | --- |
> | exb (0shot) w/ ChatGPT | ***19.3*** | ***15.5*** | 5.8 | 10.8 | 16.5 | 3.9 | 11.2 | 9.8 | *4.2* | 0.4 | ***10.3*** | ***7*** | ***10.4*** | 5.2 | 4.8 | **0.1** | **3.7** | 7.2 |
> | exb-lbl (0shot) w/ ChatGPT | 16.2 | 14.3 |  | ***14.3*** | ***17.6*** | *4.7* | ***11.5*** | ***10.7*** | 3.6 | *0.7* | 9.5 | 5.4 | 10.3 |  |  | **0.1** |  |  |
> | exb-lbl-d (0shot) w/ ChatGPT |  |  |  |  |  |  |  |  |  |  |  |  |  |  |  |  |  |  |
>
>
> ##### Table 1.2 (w/ CodeGen2-16B)
>
> | exp  | js | cpp | jv | ts | php | rb | cs | go | pl | r | rs | sc | sw | sh | lua | rkt | jl | d |
> | --- | --- | --- | --- | --- | --- | --- | --- | --- | --- | --- | --- | --- | --- | --- | --- | --- | --- | --- |
> | baseline (0shot) | **48.4** | **41.2** |  | **44.8** | **38.4** | **8.6** | **46** |  |  |  |  |  |  |  |  |  |  | 11.7 |
> | exp (0shot) | 45.1 | 38.2 |  | 43.7 | 37.5 | 2.8 | 41.8 |  |  |  |  |  |  |  |  |  |  | 14.4 |
> | explain-lbl (0shot) | 45.1 |  |  |  |  |  |  |  |  |  |  |  |  |  |  |  |  | **15.8** |
> | explain-lbl-d (0shot) | 43.2 |  |  |  |  |  |  |  |  |  |  |  |  |  |  |  |  | 15 |
>
>
>
>
> ### X-to-X Translation (Similar to paper table 2)
>
> For the two tables below, we report translation pass@1 between 16 different pairs of languages. **Resource** indicates the language resource levels of the source and target. **Type** indicates the source and target language typing characteristics
> (D/S=dynamically/statically typed). The best runs within the same-shot setting are **bolded**. All parameters are kept the same as table 2 in paper.
>
> ##### Table 2.1 (w/ CodeGen2-1B)
>
> |  |  |  | 0 shot |  |  |  |
> | --- | --- | --- | --- | --- | --- | --- |
> | Resource | Typing | src-tgt | direct | exp | exp-lbl | exp-lbl-d |
> | High-to-High | D-D | Python - JavaScript | 4.6 |  **14.9**  | 13.9 | 13.1 |
> |  | D-T | JavaScript - Java | 7.8 |  **11.2** | 9.6 | 10.1 |
> |  | T-D | C++ - Python | 8.3 | **13.9** | 13.5 | 10.7 |
> |  | T-T | Java - C++ | 14.2 | 15.6 | 14.8 | **17.5** |
> | High-to-ExtLow | D-D | JavaScript - Racket | 0 | 0 | 0 | 0 |
> |  | D-T | Python - D | 6.2 | **7.9** | 7.3 | 5.9 |
> |  | T-D | C++ - Lua | 8.9 | **11.8** | 10 | 9.8 |
> |  | T-T | Java - Julia | 3.2 | **4.9** | 3.9 | 4.4 |
> | ExtLow-to-High | D-D | Lua - Python | 1.9 | **6** | 5.9 | 5.6 |
> |  | D-T | Racket - Java | 7.9 | 7.6 | **8.1** | 7.9 |
> |  | T-D | Julia - JavaScript | 3.7 | **6.9** | 3.9 | 5.1 |
> |  | T-T | D - C++ | 11.7 | 12.7 | 12.6 | **14.1** |
> | ExtLow-toExtLow | D-D | Lua - Racket | 0 | 0 | 0 | 0 |
> |  | D-T | Racket - Julia | 2 | **2.1** | 1.5 | 0.8 |
> |  | T-D | D - Lua | 7.7 | **8.2** | 6.3 | 7.2 |
> |  | T-T | Julia - D | 0 | **0.007** | 0 | 0 |
>
> ##### Table 2.2 (w/ CodeGen2-16B)
>
> |  |  |  | 0 shot |  |  |  |
> | --- | --- | --- | --- | --- | --- | --- |
> | High-to-High | D-D | Python - JavaScript | **48.4** | 45.1 | 45.1 | 43.1 |
> |  | D-T | JavaScript - Java | **47.3** | 44.3 | 42 | 40 |
> |  | T-D | C++ - Python | 38 | **57.2** | 55.1 |  |
> |  | T-T | Java - C++ | 52.7 |  |  |  |
> | High-to-ExtLow | D-D | JavaScript - Racket | 2 |  |  |  |
> |  | D-T | Python - D | 11.7 | 14.4 | **15.8** | 15 |
> |  | T-D | C++ - Lua | 33.6 |  |  |  |
> |  | T-T | Java - Julia | 26.4 | **27.9** | 25.8 | 25.9 |
> | ExtLow-to-High | D-D | Lua - Python | 37.6 | 45.6 | **47.8** | 46.7 |
> |  | D-T | Racket - Java | **28.4** | 27.8 |  |  |
> |  | T-D | Julia - JavaScript | 40.1 | 39.7 | 40 | **42.9** |
> |  | T-T | D - C++ | **53.1** | 47.6 | 47.1 | 50.4 |
> | ExtLow-toExtLow | D-D | Lua - Racket | 0.8 |  |  |  |
> |  | D-T | Racket - Julia | 13.1 | **14.6** | **14.6** | 14.4 |
> |  | T-D | D - Lua | **29.3** | 28.1 | 28.8 | 28.9 |
> |  | T-T | Julia - D | **10.1** | 7.8 | 5.4 | 5.3 |
>
>
>
> From the tables above, we can draw the following additional conclusions, which we will include in the next version of the paper:
> - In weaker models, **we still see improvements with self-generated explanations across most directions**. CodeGen2-1B improves more consistently using self-generated explanations (baseline is the worst in all X-to-X directions, and in 17/18 Python-to-X directions), as much as 300%+ improvement (Table 2.3, Lua→Python, Python→JavaScript). In Python→Ruby, model with explanations obtains a pass rate of 5.9, while baseline does not generate *any* single correct translation (pass rate of 0). CodeGen2-16B shows weaker results, with baseline outperforming in 10/18 directions in Table 1 and 2. Perhaps it has a weaker alignment between natural language and programming language, resulting in worse explanations generated for each problem. The majority of errors from translation with explanation are syntactic.
> - **Better explanation leads to better translation, even in smaller model cases**. We compare CodeGen2-1B performance given self-generated explanation vs. ChatGPT generated explanation and see that the better explanation outperforms self-generated explanation in 12/18 Python-to-X directions, with a maximum improvement of 400%+ in Python → JavaScript.
> - In smaller/weaker models, **detailed explanations (exp-lbl or exp-lbl-d) do not improve as much as exp does**. Often this is due to lower quality explanations generated when the model is asked to do something it is not capable of. The line-by-line explanations often lead to repetitive content when source programs contain several repetitive lines (library import in C++ → Python direction, with CodeGen2-1B)
> - we will add these results and results from StarCoder in the revised version of our paper

---

### Official Review · Reviewer_pfnB · 2023-08-09

**Soundness:** 3

**Excitement:**

3: Ambivalent: It has merits (e.g., it reports state-of-the-art results, the idea is nice), but there are key weaknesses (e.g., it describes incremental work), and it can significantly benefit from another round of revision. However, I won't object to accepting it if my co-reviewers champion it.

**Paper Topic And Main Contributions:**

This paper is concerned with chain-of-thought prompting as applied to program source code translation. More in particular, it applies this method (originally from Chen et al., Teaching large language models to self-debug, 2023 ) extending the number of languages to 19, across low- and high-resource ones.

**Questions For The Authors:**

A The 3.3. Metrics section is very brief; could you elaborate on that comment "n, k" often dependent of sampling temperature k ? What does this dependency look like?

**Reasons To Accept:**

This paper presents a battery of translation pairs and prompt variations. Most importantly, it summarizes the vast experimental evidence into nuggets that hopefully will be valuable to theorists. As such, it's a sound empirical work and well suited for EMNLP.

**Reasons To Reject:**

To me, using a black-box, proprietary model like ChatGPT which periodically changes under the hood is a major scientific liability. This paper as a whole won't likely be reproducible in a few weeks' to months' time.  It is really unfortunate that the authors chose this and not a "weaker" but reproducible model, especially considering the wealth of OSS code LMs we have nowadays (CodeT5, StarCoder, etc.). I get it, ChatGPT is apparently "strong" in a number of areas, very convenient to use etc., but this is not what good science is about.    If the paper used an open-source model instead, or at the very least made a comparison to an OSS LM, I would be much more inclined to recommend it for acceptance.

---

## Rebuttal acknowlegement

I appreciate the authors' response and I gladly increase my score.

**Reproducibility:**

4: Could mostly reproduce the results, but there may be some variation because of sample variance or minor variations in their interpretation of the protocol or method.

**Reviewer Confidence:**

3: Pretty sure, but there's a chance I missed something. Although I have a good feel for this area in general, I did not carefully check the paper's details, e.g., the math, experimental design, or novelty.

**Typos Grammar Style And Presentation Improvements:**

Only in the Conclusions section we learn that the model under test is ChatGPT. This fact should be made clear in the beginning of the work, or at least in the Experiments section.

It is really hard to see a linear fit in Figure 2, seeing the data.I wonder if that correlation could be faceted by controlling for e.g. language dataset size.

---

> ### Author Rebuttal · Authors · 2023-08-29
>
> We thank the reviewer for a thorough review! The goal of our paper is to study how various types of explanations improve code-to-code translation with a strong model. Since ChatGPT is (near)  the SOTA model  we focused on that. However, we completely understand (and agree with) your  point about  ChatGPT-only results. Hence, we conducted additional experiments with an open-source model (CodeGen 1B). The main conclusion is that we can indeed observe similar improvements by providing explanations in smaller open source models. We will make sure to include these new results in the updated version of the paper.
>
> We address some of the  other concerns in below:
>
> > The 3.3. Metrics section is very brief; could you elaborate on that comment "n, k" often dependent of sampling temperature k ? What does this dependency look like?:
>
> The metric of pass@k was first popularized by Chen et. al. 2021 (Codex paper), which was invented so users of the code generation system have a good idea of fractions of successful code generation out of k samples. In the same paper, the authors also perform extensive experiments showing the relationship between sampling temperature and pass@k (Figure 5, 6). To summarize, lower temperature (0.2) is better when n/k is small, but higher temperature can improve when n/k is large due to the variance in the generation (higher recall). In our experiments, we focus on improving mainly pass@1, so we followed Chen et. al. 2021 and Cassano, et al 2023’s advice for using temperature 0.2. We will note this dependency in the future iteration of the paper.
>
> > It is really hard to see a linear fit in Figure 2, seeing the data.I wonder if that correlation could be faceted by controlling for e.g. language dataset size.
>
> As mentioned in text, the correlation is particularly strong ($r^2$
>  = 0.719) if we disregard some outliers (Bash, R, Julia). Bash in particular, has poor test coverage/quality (this is from actual authors of the MultiPL-E paper) and should be analyzed lightly. Nonetheless, our intention with the figure is to bring awareness to the reader that explain-then-translate, similar to few-shot prompting, is just another method to improve performance. Such observation subsequently leads to our discussion on whether natural language or programming language is better as a self-generated context to boost performance. As to the point of whether it is related to dataset size of the programming language, we believe it is not that clear-cut of a story: lower-resource languages such as Racket and D sit on polar ends of the linear fit, and so do higher-resource languages (JavaScript vs. Java).
>
> > If the paper used an open-source model instead, or at the very least made a comparison to an OSS LM, I would be much more inclined to recommend it for acceptance.
>
> In the time available during the rebuttal period, we are able to reproduce most of Table1, Table2 for 0-shot setting with more open source models (CodeGen2-16B, CodeGen2-1B). We plan to finish the experiments with StarCoder as well to compare with the current open source SOTA code model. Additionally, we tested whether smaller models can benefit from better explanations generated by big models (prompting CodeGen2-1B with ChatGPT generated explanations). We hope the tables of results and additional comparison of results across open-source models can be taken into consideration.
>
> ### Python-to-X Translation (Similar to paper table 1)
>
> Translation pass@1 from Python to X. Parenthesis in trial indicates # of shots. Best within same explanation source model is *italicized* and overall best is **bolded**. All parameters are kept the same as table 1 in paper.
>
> ##### Table 1.1 (w/ CodeGen2-1B, self generated explanation and ChatGPT-generated explanation)
>
> | exp  | js | cpp | jv | ts | php | rb | cs | go | pl | r | rs | sc | sw | sh | lua | rkt | jl | d |
> | --- | --- | --- | --- | --- | --- | --- | --- | --- | --- | --- | --- | --- | --- | --- | --- | --- | --- | --- |
> | baseline (0shot) | 4.6 | 9.4 | 5.3 | 7 | 13.4 | 0 | 6.4 | 3.5 | 3.6 | 0.6 | 4.7 | 2.6 | 3.5 | ***5.7*** | 2.7 | 0 | 2.4 | 6.2 |
> | exb (0shot) | *15.4* | *11.7* | ***7.8*** | *13.5* | 14.2 | ***5.9*** | 7.9 | 8.3 | 4.1 | 0.8 | *8.7* | *5.6* | *6.9* | 4.8 | 5.3 | 0 | *3.1* | ***7.9*** |
> | exb-lbl (0shot) | 13.9 | 11.1 | 6.5 | 11.3 | *17.3* |  |  |  |  |  |  |  |  | 3.1 |  |  |  | 7.3 |
> | exb-lbl-d (0shot) | 12.7 | *11.7* | 7.6 | 9.9 | 14.6 | 3.9 | *8.5* | *8.8* | *4.5* | ***0.9*** | 8.2 | 2.4 | *6.9* | 4.7 | ***5.4*** | 0 | 2.8 | 5.9 |
> | --- | --- | --- | --- | --- | --- | --- | --- | --- | --- | --- | --- | --- | --- | --- | --- | --- | --- | --- |
> | exb (0shot) w/ ChatGPT | ***19.3*** | ***15.5*** | 5.8 | 10.8 | 16.5 | 3.9 | 11.2 | 9.8 | *4.2* | 0.4 | ***10.3*** | ***7*** | ***10.4*** | 5.2 | 4.8 | **0.1** | **3.7** | 7.2 |
> | exb-lbl (0shot) w/ ChatGPT | 16.2 | 14.3 |  | ***14.3*** | ***17.6*** | *4.7* | ***11.5*** | ***10.7*** | 3.6 | *0.7* | 9.5 | 5.4 | 10.3 |  |  | **0.1** |  |  |
> | exb-lbl-d (0shot) w/ ChatGPT |  |  |  |  |  |  |  |  |  |  |  |  |  |  |  |  |  |  |
>
> ##### Table 1.2 (w/ CodeGen2-16B)
>
> | exp  | js | cpp | jv | ts | php | rb | cs | go | pl | r | rs | sc | sw | sh | lua | rkt | jl | d |
> | --- | --- | --- | --- | --- | --- | --- | --- | --- | --- | --- | --- | --- | --- | --- | --- | --- | --- | --- |
> | baseline (0shot) | **48.4** | **41.2** |  | **44.8** | **38.4** | **8.6** | **46** |  |  |  |  |  |  |  |  |  |  | 11.7 |
> | exp (0shot) | 45.1 | 38.2 |  | 43.7 | 37.5 | 2.8 | 41.8 |  |  |  |  |  |  |  |  |  |  | 14.4 |
> | explain-lbl (0shot) | 45.1 |  |  |  |  |  |  |  |  |  |  |  |  |  |  |  |  | **15.8** |
> | explain-lbl-d (0shot) | 43.2 |  |  |  |  |  |  |  |  |  |  |  |  |  |  |  |  | 15 |
>
>
>
>
> ### X-to-X Translation (Similar to paper table 2)
>
> For the two tables below, we report translation pass@1 between 16 different pairs of languages. **Resource** indicates the language resource levels of the source and target. **Type** indicates the source and target language typing characteristics
> (D/S=dynamically/statically typed). The best runs within the same-shot setting are **bolded**. All parameters are kept the same as table 2 in paper.
>
> ##### Table 2.1 (w/ CodeGen2-1B)
>
> |  |  |  | 0 shot |  |  |  |
> | --- | --- | --- | --- | --- | --- | --- |
> | Resource | Typing | src-tgt | direct | exp | exp-lbl | exp-lbl-d |
> | High-to-High | D-D | Python - JavaScript | 4.6 |  **14.9**  | 13.9 | 13.1 |
> |  | D-T | JavaScript - Java | 7.8 |  **11.2** | 9.6 | 10.1 |
> |  | T-D | C++ - Python | 8.3 | **13.9** | 13.5 | 10.7 |
> |  | T-T | Java - C++ | 14.2 | 15.6 | 14.8 | **17.5** |
> | High-to-ExtLow | D-D | JavaScript - Racket | 0 | 0 | 0 | 0 |
> |  | D-T | Python - D | 6.2 | **7.9** | 7.3 | 5.9 |
> |  | T-D | C++ - Lua | 8.9 | **11.8** | 10 | 9.8 |
> |  | T-T | Java - Julia | 3.2 | **4.9** | 3.9 | 4.4 |
> | ExtLow-to-High | D-D | Lua - Python | 1.9 | **6** | 5.9 | 5.6 |
> |  | D-T | Racket - Java | 7.9 | 7.6 | **8.1** | 7.9 |
> |  | T-D | Julia - JavaScript | 3.7 | **6.9** | 3.9 | 5.1 |
> |  | T-T | D - C++ | 11.7 | 12.7 | 12.6 | **14.1** |
> | ExtLow-toExtLow | D-D | Lua - Racket | 0 | 0 | 0 | 0 |
> |  | D-T | Racket - Julia | 2 | **2.1** | 1.5 | 0.8 |
> |  | T-D | D - Lua | 7.7 | **8.2** | 6.3 | 7.2 |
> |  | T-T | Julia - D | 0 | **0.007** | 0 | 0 |
>
> ##### Table 2.2 (w/ CodeGen2-16B)
>
> |  |  |  | 0 shot |  |  |  |
> | --- | --- | --- | --- | --- | --- | --- |
> | High-to-High | D-D | Python - JavaScript | **48.4** | 45.1 | 45.1 | 43.1 |
> |  | D-T | JavaScript - Java | **47.3** | 44.3 | 42 | 40 |
> |  | T-D | C++ - Python | 38 | **57.2** | 55.1 |  |
> |  | T-T | Java - C++ | 52.7 |  |  |  |
> | High-to-ExtLow | D-D | JavaScript - Racket | 2 |  |  |  |
> |  | D-T | Python - D | 11.7 | 14.4 | **15.8** | 15 |
> |  | T-D | C++ - Lua | 33.6 |  |  |  |
> |  | T-T | Java - Julia | 26.4 | **27.9** | 25.8 | 25.9 |
> | ExtLow-to-High | D-D | Lua - Python | 37.6 | 45.6 | **47.8** | 46.7 |
> |  | D-T | Racket - Java | **28.4** | 27.8 |  |  |
> |  | T-D | Julia - JavaScript | 40.1 | 39.7 | 40 | **42.9** |
> |  | T-T | D - C++ | **53.1** | 47.6 | 47.1 | 50.4 |
> | ExtLow-toExtLow | D-D | Lua - Racket | 0.8 |  |  |  |
> |  | D-T | Racket - Julia | 13.1 | **14.6** | **14.6** | 14.4 |
> |  | T-D | D - Lua | **29.3** | 28.1 | 28.8 | 28.9 |
> |  | T-T | Julia - D | **10.1** | 7.8 | 5.4 | 5.3 |
>
>
>
> From the tables above, we can draw the following additional conclusions, which we will include in the next version of the paper:
> - In weaker models, **we still see improvements with self-generated explanations across most directions**. CodeGen2-1B improves more consistently using self-generated explanations (baseline is the worst in all X-to-X directions, and in 17/18 Python-to-X directions), as much as 300%+ improvement (Table 2.3, Lua→Python, Python→JavaScript). In Python→Ruby, model with explanations obtains a pass rate of 5.9, while baseline does not generate *any* single correct translation (pass rate of 0). CodeGen2-16B shows weaker results, with baseline outperforming in 10/18 directions in Table 1 and 2. Perhaps it has a weaker alignment between natural language and programming language, resulting in worse explanations generated for each problem. The majority of errors from translation with explanation are syntactic.
> - **Better explanation leads to better translation, even in smaller model cases**. We compare CodeGen2-1B performance given self-generated explanation vs. ChatGPT generated explanation and see that the better explanation outperforms self-generated explanation in 12/18 Python-to-X directions, with a maximum improvement of 400%+ in Python → JavaScript.
> - In smaller/weaker models, **detailed explanations (exp-lbl or exp-lbl-d) do not improve as much as exp does**. Often this is due to lower quality explanations generated when the model is asked to do something it is not capable of. The line-by-line explanations often lead to repetitive content when source programs contain several repetitive lines (library import in C++ → Python direction, with CodeGen2-1B)
> - we will add these results and results from StarCoder in the revised version of our paper

---

### Meta-Review · Area_Chair_xmtM · 2023-09-16

**Recommendation:** 3

**Metareview:**

This study applies CoT to programming translation tasks. Initially, it generates explanations for the source code and then proceeds to generate the programming code. This approach has demonstrated performance improvements in multiple settings. Reviewers have expressed concerns about the reproducibility of experiments conducted solely with ChatGPT, and the authors have presented additional experiments during the rebuttal to address these concerns. On the other hand, the authors plan to compare with the open source SOTA model in the future.

---

### Decision · Program_Chairs · 2023-10-07

**Decision:**

Accept-Findings

**Comment:**

This study applies CoT to programming translation tasks. Initially, it generates explanations for the source code and then proceeds to generate the programming code. This approach has demonstrated performance improvements in multiple settings. Reviewers have expressed concerns about the reproducibility of experiments conducted solely with ChatGPT, and the authors have presented additional experiments during the rebuttal to address these concerns. On the other hand, the authors plan to compare with the open source SOTA model in the future.